# Global net climate effects of anthropogenic reactive nitrogen

Cheng Gong[1✉], Hanqin Tian[2,3], Hong Liao[4], Naiqing Pan[2,5], Shufen Pan[2,6], Akihiko Ito[7,8], Atul K. Jain[9], Sian Kou-Giesbrecht[10], Fortunat Joos[11,12], Qing Sun[11,12], Hao Shi[13], Nicolas Vuichard[14], Qing Zhu[15], Changhui Peng[16,17], Federico Maggi[18], Fiona H. M. Tang[19] & Sönke Zaehle[1]

Anthropogenic activities have substantially enhanced the loadings of reactive nitrogen (Nr) in the Earth system since pre-industrial times[1,2], contributing to widespread eutrophication and air pollution[3–6]. Increased Nr can also influence global climate through a variety of effects on atmospheric and land processes but the cumulative net climate effect is yet to be unravelled. Here we show that anthropogenic Nr causes a net negative direct radiative forcing of −0.34 [−0.20, −0.50] W m$^{-2}$ in the year 2019 relative to the year 1850. This net cooling effect is the result of increased aerosol loading, reduced methane lifetime and increased terrestrial carbon sequestration associated with increases in anthropogenic Nr, which are not offset by the warming effects of enhanced atmospheric nitrous oxide and ozone. Future predictions using three representative scenarios show that this cooling effect may be weakened primarily as a result of reduced aerosol loading and increased lifetime of methane, whereas in particular N$_2$O-induced warming will probably continue to increase under all scenarios. Our results indicate that future reductions in anthropogenic Nr to achieve environmental protection goals need to be accompanied by enhanced efforts to reduce anthropogenic greenhouse gas emissions to achieve climate change mitigation in line with the Paris Agreement.

Reactive nitrogen (Nr) in the Earth system, defined as organic and inorganic forms of nitrogen (N) compounds, except the dinitrogen gas (N$_2$), has increased rapidly since the industrial revolution[1]. This increase can be mainly attributed to emissions associated with anthropogenic fossil fuel combustion and fertilizer application[1,2]. Elevated concentrations of Nr induce detrimental environmental effects[3,4], including air pollution[5], eutrophication of surface and near-coast water[6] and biodiversity loss[7], but can also substantially influence climate. Specifically, the long-lived greenhouse gas nitrous oxide (N$_2$O) contributes to warming of the atmosphere[8], whereas short-lived ammonium (NH$_4^+$) and nitrate (NO$_3^-$) aerosols generated from ammonia (NH$_3$) and nitrogen oxide (NO$_x$) gases can scatter solar radiation and thereby cool the atmosphere[9–11]. NO$_x$ furthermore plays a pivotal role in various atmospheric chemical reactions, regulating the lifetimes and thus mole fractions of other gases, such as the greenhouse gases methane (CH$_4$)[12] and ozone (O$_3$)[13]. Furthermore, fertilizer application and deposition of atmospheric Nr on land and ocean can alleviate N limitation in terrestrial or marine ecosystems and

facilitate carbon sequestration, thereby reducing atmospheric CO$_2$ concentrations[14,15] and exerting a cooling effect on the atmosphere (Fig. 1).

So far, the net global Nr climate effect remains unclear because of the substantial variation of individual Nr-related processes across geographic regions[16] and the timescale dependence of the climate responses to anthropogenic Nr (ref. 17). An earlier study estimated a global net radiative forcing of anthropogenic Nr of −0.24 [+0.2, −0.5] W m$^{-2}$ based only on literature review[18]. Some studies, focusing on hotspots of anthropogenic Nr, such as the United States[19], Europe[20] and China[21], have also assessed the components of the regional climate effects of anthropogenic Nr based on a literature review of the sensitivities of individual processes to anthropogenic Nr inputs. However, these assessments were constrained by their focus on present-day anthropogenic Nr levels, thus neglecting the cumulative effects of long-lived greenhouse gases since the pre-industrial era. They were also limited by the extent to which they consider spatial heterogeneity and nonlinearities between the coupled biogeochemical cycles and atmospheric lifetime of the different

[1]Max Planck Institute for Biogeochemistry, Jena, Germany. [2]Center for Earth System Science and Global Sustainability, Schiller Institute for Integrated Science and Society, Boston College, Chestnut Hill, MA, USA. [3]Department of Earth and Environmental Sciences, Boston College, Chestnut Hill, MA, USA. [4]School of Environmental Science and Engineering, Nanjing University of Information Science and Technology, Nanjing, China. [5]International Center for Climate and Global Change Research, College of Forestry, Wildlife and Environment, Auburn University, Auburn, AL, USA. [6]Department of Engineering and Environmental Studies Program, Boston College, Chestnut Hill, MA, USA. [7]Graduate School of Agricultural and Life Sciences, University of Tokyo, Tokyo, Japan. [8]Earth System Division, National Institute for Environmental Studies, Tsukuba, Japan. [9]Department of Atmospheric Science, University of Illinois, Urbana-Champaign, Urbana, IL, USA. [10]Department of Earth and Environmental Sciences, Dalhousie University, Halifax, Nova Scotia, Canada. [11]Climate and Environmental Physics, Physics Institute, University of Bern, Bern, Switzerland. [12]Oeschger Centre for Climate Change Research, University of Bern, Bern, Switzerland. [13]State Key Laboratory of Urban and Regional Ecology, Research Center for Eco-Environmental Sciences, Chinese Academy of Sciences, Beijing, China. [14]Laboratoire des Sciences du Climat et de l'Environnement, LSCE-IPSL (CEA-CNRS-UVSQ), Université Paris-Saclay, Gif-sur-Yvette, France. [15]Climate and Ecosystem Sciences Division, Lawrence Berkeley National Lab, Berkeley, CA, USA. [16]Department of Biology Sciences, Institute of Environment Science, University of Quebec at Montreal, Montreal, Quebec, Canada. [17]School of Geographic Sciences, Hunan Normal University, Changsha, China. [18]Environmental Engineering, School of Civil Engineering, The University of Sydney, Sydney, New South Wales, Australia. [19]Department of Civil Engineering, Monash University, Clayton, Victoria, Australia. ✉e-mail: cgong@bgc-jena.mpg.de

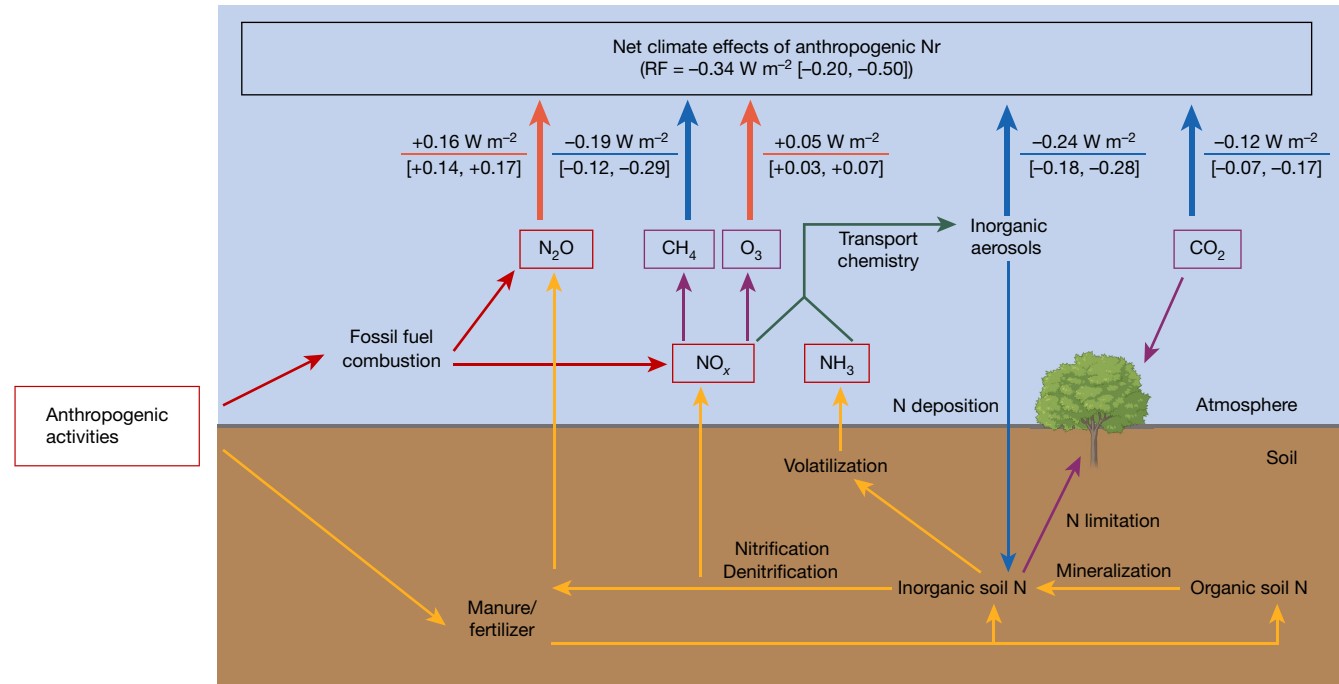

**Fig. 1 | Pathways of anthropogenic Nr effects on global climate.** Solid lines and arrows represent processes included in this study using a combination of terrestrial biosphere and atmospheric chemistry modelling. The direct radiative forcing estimates represent the climatic effects of anthropogenic Nr. The uncertainty range of each pathway is estimated on the basis of the 1 standard deviation across the NMIP2 ensemble members as well as the uncertainties in atmospheric chemistry (Supplementary Information Section 1.2). Orange and dark blue solid arrows on the top indicate the warming or cooling effects, respectively. The image of the tree was created using BioRender.com.

forcers, resulting in significant uncertainties that impede attempts to extrapolate regional estimates to globe scale.

Filling these knowledge gaps requires the integration of terrestrial biogeochemistry and atmospheric chemistry to account for the intricate transformations of Nr compounds and the resulting trade-offs in the climate impacts[22]. Previous studies have demonstrated the effectiveness of both process-based terrestrial biosphere models and global chemical transport models separately in assessing the climate effects of specific anthropogenic Nr compounds or processes[9,16,23–26]. However, most of the studies associated with terrestrial Nr fluxes only relied on a single model, whereas incorporating model uncertainty is essential for a robust assessment[27].

Here we present a comprehensive model framework to estimate the global net direct radiative forcing of anthropogenic Nr as well as the likely changes in radiative forcings in response to future changes of anthropogenic Nr inputs. First, we integrated anthropogenic emission inventories from the community emissions data system (CEDS) and eight terrestrial biosphere model outputs from the global nitrogen/N$_2$O model inter-comparison project phase 2 (NMIP2)[28] to quantify the historical anthropogenic Nr effects on terrestrial carbon sequestration, soil NH$_3$ volatilization and soil NO$_x$ and N$_2$O emissions. Second, we performed a series of model experiments using box models of greenhouse gases and a global chemical transport model (GEOS-Chem) coupled to a radiative transfer module (RRTMG) to estimate the global net direct radiative forcing of anthropogenic Nr associated with each of these emission sources. Finally, we estimated how the net direct radiative forcing may respond to future scenarios of anthropogenic Nr inputs.

## Effects of anthropogenic Nr on emissions

We integrated results from the NMIP2 ensemble with the CEDS inventory (Methods) to comprehensively represent anthropogenic Nr effects on terrestrial carbon fluxes, N$_2$O, NH$_3$ and NO$_x$ emissions (Fig. 2). Here,

anthropogenic Nr sources were defined as the set of anthropogenic activities that directly add Nr into terrestrial ecosystems or the atmosphere, including manure and fertilizer application, N deposition, fossil fuel combustion and livestock NH$_3$ emissions. Other anthropogenic factors in the configuration of NMIP2, for example, irrigation, land-use change (LUC), elevated CO$_2$ and changing climate, also affect the global N cycle indirectly and thereby modify the response of the terrestrial biosphere to Nr additions. However, a robust identification of anthropogenic contributions from these indirect factors is not possible and therefore the influences of these indirect factors were not attributed to anthropogenic Nr effects in this study.

The net biome productivity (NBP) of the NMIP2 ensemble, which corresponds the terrestrial carbon balance, showed similar magnitudes and trends relative to Global Carbon Project 2021[29], with the correlation coefficient of 0.94 (0.98) and mean bias of −0.1 PgC yr$^{-1}$ (0.2 PgC yr$^{-1}$) when excluding (including) the effects of LUC, respectively (Extended Data Fig. 1). Anthropogenic Nr, including fertilizer and manure applications and N deposition, increased terrestrial carbon sinks by 0.55 ± 0.38 PgC yr$^{-1}$ over 2016–2020 (Fig. 2a). The N$_2$O emissions from both soils and fossil fuel combustion over 2016–2020 were 12.6 ± 1.5 TgN yr$^{-1}$, for which anthropogenic Nr contributed about 5.5 ± 0.97 TgN yr$^{-1}$. The NMIP2 ensemble estimates of global soil N$_2$O emissions induced by manure and fertilizer application, N deposition and from natural soil during 2016–2020 were 2.7 ± 0.95, 0.80 ± 0.22 and 6.2 ± 1.6 TgN yr$^{-1}$, respectively (Fig. 2b). These estimates fall well within the uncertainty ranges of 2.5–5.8, 0.4–1.4 and 4.9–6.5 TgN yr$^{-1}$ over 2007–2016 according to the latest N$_2$O budget estimates[30].

Anthropogenic NO$_x$ emissions during 2016–2020 reached 46.5 ± 2.7 TgN yr$^{-1}$, most of which were due to fossil fuel combustion, as derived from CEDS. The NMIP2 ensemble estimated that anthropogenic activities contributed about 3.1 ± 0.77 TgN yr$^{-1}$ of the 12.2 ± 2.7 TgN yr$^{-1}$ global soil NO$_x$ emissions over 2016–2020 (Fig. 2c), the last of which was slightly higher than the recent estimates of about 9.5 ± 0.4 TgN yr$^{-1}$

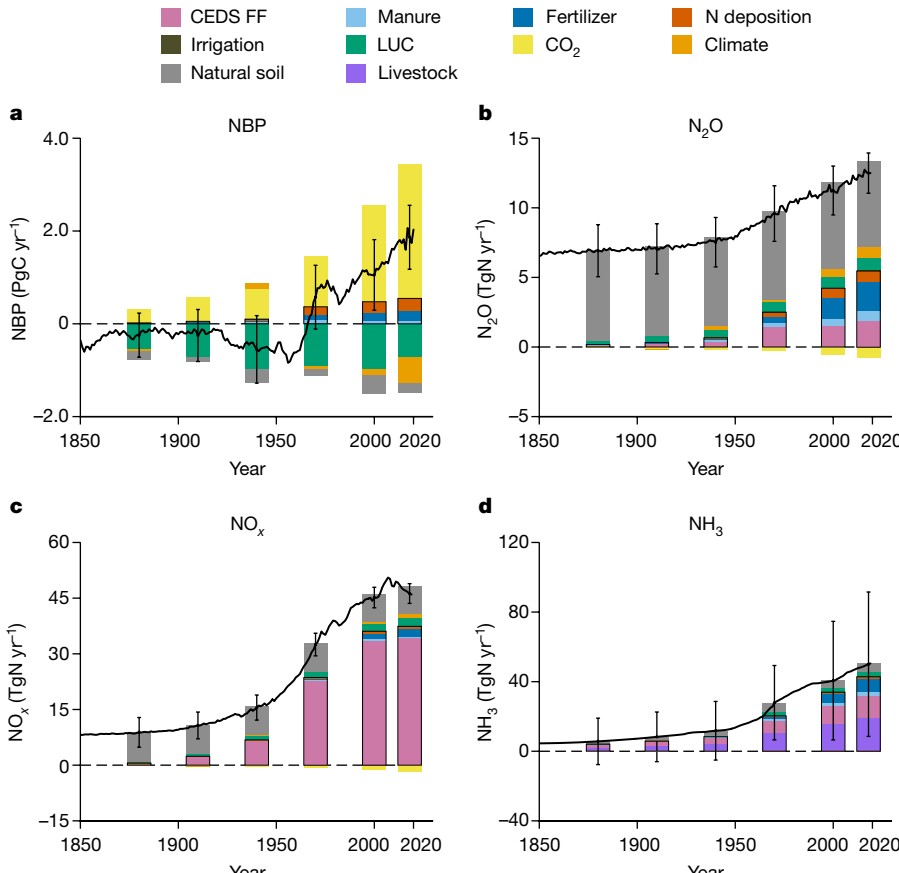

**CEDS FF** **Manure** **Fertilizer** **N deposition**
**Irrigation** **LUC** **CO₂** **Climate**
**Natural soil** **Livestock**

**Fig. 2 | Historical Nr emissions and terrestrial carbon fluxes based on CEDS inventory and NMIP2 ensemble mean. a–d**, The terrestrial NBP (**a**), $N_2O$ (**b**), $NO_x$ (**c**) and $NH_3$ (**d**) emissions, respectively. All of the fossil fuel sources in CEDS are indicated by the pale violet bars, whereas other colours indicate factor contributions based on the NMIP2 ensemble mean. The soil $NH_3$ emissions have been scaled by the CEDS agricultural emissions (Methods). The fire emission of each component is not included in the current NMIP2 configuration. The contributions of each factor were averaged over 1880s, 1910s, 1940s, 1970s, 2000s and 2020s with a 5-year time window, in which the direct anthropogenic Nr effects are shown with a black outline. Black lines indicate the ensemble mean annual flux of each compound and the error bars indicate 1 standard deviation among different NMIP2 members. FF, fossil fuel.

for 1980–2017[31]. Independent estimates of the present-day global $NH_3$ emissions are highly uncertain with a range of 40–163 TgN yr⁻¹ (refs. 32–34). This uncertainty is also reflected in the spread of the NMIP2 ensemble (Fig. 2d and Supplementary Fig. 1), which, however, showed a consistent relative imprint of agricultural fertilizer and manure applications on the trend of global $NH_3$ emissions from 1850 to 2019. To derive a globally consistent time evolution of the anthropogenic Nr effect on $NH_3$ emissions for our climate assessment, we adopted a conservative estimate of total $NH_3$ emissions of 50.5 TgN yr⁻¹ in 2019 based on CEDS inventory and applied relative contribution of anthropogenic Nr to $NH_3$ emissions simulated by the NMIP2 ensemble (Methods).

## Radiative forcing from anthropogenic Nr

We next examined the net climate effects of anthropogenic Nr by combining the box-model simulated atmospheric $CO_2$, $N_2O$ and $CH_4$ concentrations and the emissions of $NH_3$ and $NO_x$, in the GEOS-Chem-RRTMG model with and without accounting for the anthropogenic Nr effect (Methods). Changes in the $CH_4$ lifetime due to the effect of changing atmospheric $NO_x$ burden on hydroxyl radical (OH) were also calculated offline using a box model (Methods). The direct radiative forcing of anthropogenic Nr for each compound was then calculated as the difference in all-sky radiative forcing at the top of the atmosphere between present-day (here defined as 2019) and pre-industrial (here defined as 1850) times. The uncertainties were estimated on the basis of the spread across the NMIP2 ensemble members as well as in atmospheric chemistry (Supplementary Information Section 1.2).

The net global direct radiative forcing associated with the cumulative effect of historical emissions in 2019 was estimated as −0.34 [−0.20, −0.50] W m⁻² (Fig. 3), for which anthropogenic Nr effects on $CO_2$, $N_2O$, $CH_4$, aerosols (including ammonium, nitrate and sulfate; Methods) and tropospheric $O_3$ contributed −0.12 [−0.07, −0.17], +0.16 [+0.14, +0.17], −0.19 [−0.12, −0.29], −0.24 [−0.18, −0.28] and +0.05 [+0.03, +0.07] W m⁻², respectively. Each component generally falls within the expected uncertainty ranges relative to previous studies focusing on individual Nr components or processes (Supplementary Information Section 3 and Supplementary Table 3). The anthropogenic Nr-induced $N_2O$ warming slightly outweighed the cooling effects by N-induced increases in terrestrial carbon sequestration, consistent with a previous study using one terrestrial biosphere model[9]. The enhanced $NO_x$ emissions led to a significant cooling effect through decreasing $CH_4$ lifetime and increasing aerosol burdens, whereas the negative direct radiative forcing of aerosols was unevenly distributed and prevalent in air-polluted regions such as Northern America, Western Europe and Eastern and Southern Asia. In response to the substantial $NO_x$ increases since pre-industrial times, present-day tropospheric $O_3$ was found to be enhanced across the entire simulated global grid, resulting in significant increases in global tropospheric $O_3$ burden from 280.1 to 325.0 Tg (Extended Data Fig. 2). This $O_3$ enhancement partly offsets the cooling climate effects from reduced $CH_4$ lifetime and increased aerosol burden considering the greenhouse gas effect of $O_3$.

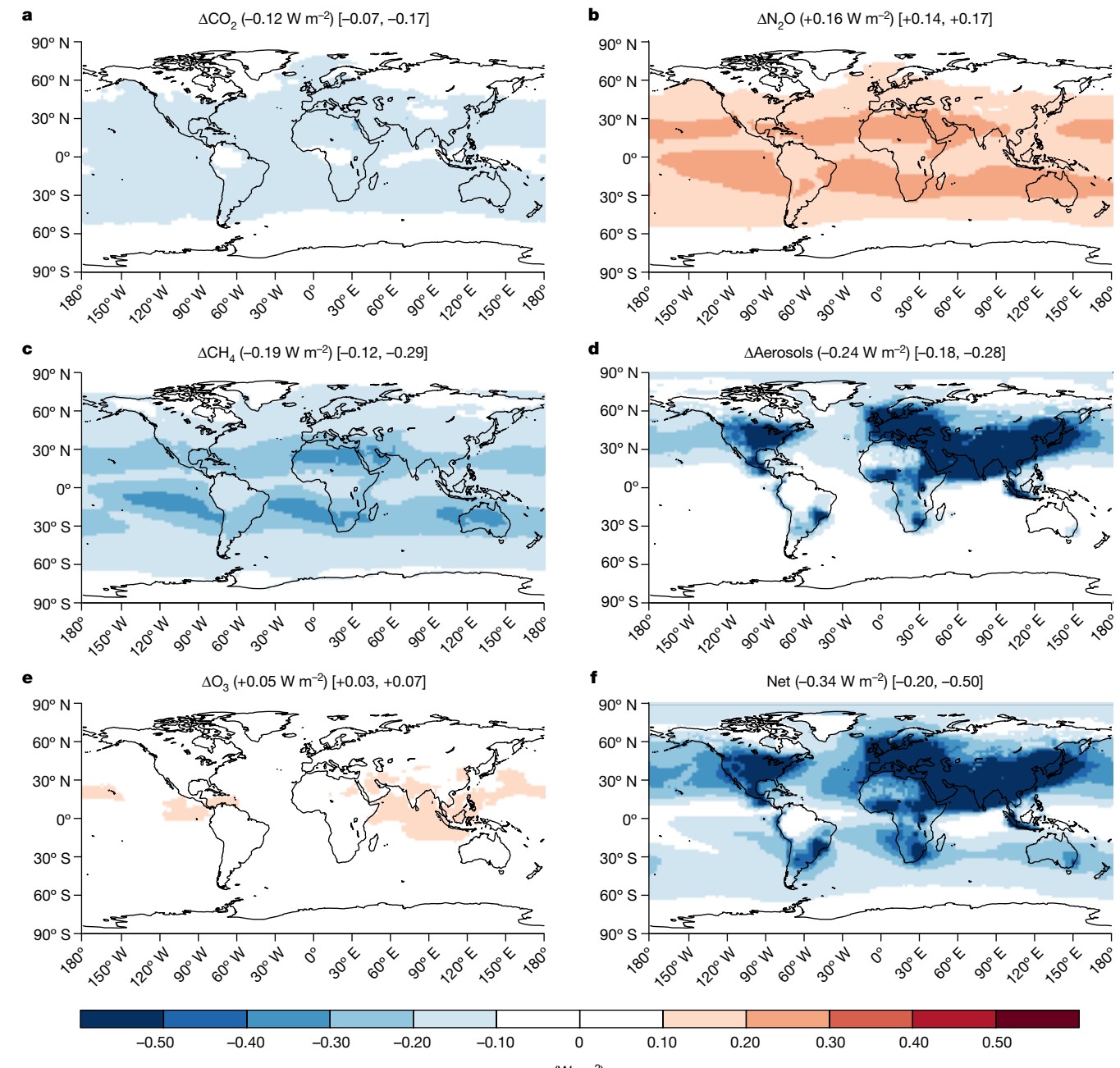

**Fig. 3 | Global direct radiative forcing in 2019 induced by anthropogenic Nr.**
**a–f**, The contributions of $CO_2$ (**a**), $N_2O$ (**b**), $CH_4$ (**c**), aerosols (**d**), $O_3$ (**e**) and the net effect (**f**) (that is, sum of **a**–**e**) were derived in the GEOS-Chem-RRTMG model by calculating differences in all-sky top-of-atmosphere radiative forcing between CTRL_2019 and No_allNr experiments. The radiative forcing of aerosols is the sum of the direct radiative forcing contributed by ammonium, nitrate and sulfate aerosols. Numbers in parentheses represent the global area-weighted averages, whereas numbers in the brackets indicate the uncertainty ranges based on sensitivity experiments with GEOS-Chem-RRTMG using ±1 standard deviation among NMIP2 ensembles as well as ±30% uncertainty in OH and $O_3$ concentrations (Supplementary Information Section 1.2). Note the Nr effects on global $CO_2$, $N_2O$ and $CH_4$ are assumed to be evenly distributed, so that the patterns of these three greenhouse gases are mostly determined by other forcing agents, including the distribution of clouds.

## Splitting agricultural and other sources

To better understand the anthropogenic Nr climate effect, we further isolated the effects of agricultural and non-agricultural activities (Methods). Here, the soil emissions attributed to fertilizer and manure application were considered as agricultural sources whereas fossil fuel combustion and soil emissions attributed to changes in N deposition were regarded as non-agricultural sources. Attributing all the N deposition as non-agricultural sources omits the effect of N deposition on agricultural fluxes[35]

but this effect will be comparatively small given the much lower N deposition rates compared to agricultural fertilizer application (Fig. 2).

Figure 4a showed that the net climate effects derived from agricultural and non-agricultural sources were comparable (−0.19 [−0.03, −0.38] and −0.19 [−0.11, −0.31] W m$^{-2}$, respectively). For the agricultural sources, the net cooling effect was dominated by the direct aerosol effect, which could be attributed to the agricultural $NH_3$ emissions, whereas the Nr effects of $CO_2$ uptake and $N_2O$ emissions on the global radiative forcing compensated each other, in agreement with

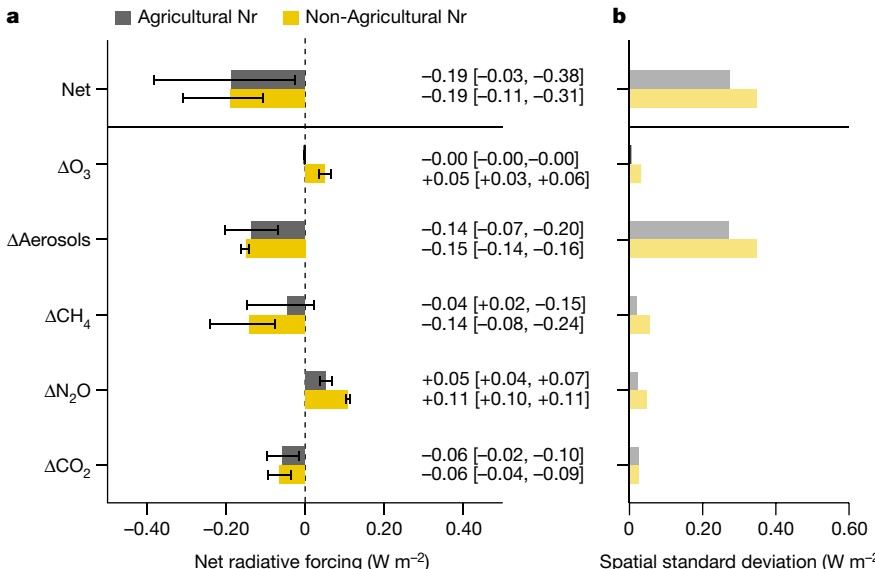

**Fig. 4 | Global direct radiative forcing associated with anthropogenic Nr from agricultural and non-agricultural sources. a**, The direct radiative forcing values are based on differences of sensitivity experiments between CTRL_2019 and No_agriNr or No_nonagriNr, respectively. The radiative forcing of aerosols is the sum of the direct radiative forcing contributed by ammonium, nitrate and sulfate aerosols. Uncertainty bars were derived from GEOS-Chem runs forced with ensemble mean ± 1 standard deviation in the NMIP2 ensemble and the associated sensitivities of radiative forcing to Nr changes (Supplementary Information Section 1.3). **b**, Spatial variation of the direct forcing effect, estimated as 1 standard deviation of direct radiative forcing across the global simulated grid.

previous studies[9]. Conversely, $NO_x$ emissions emitted from fossil fuel combustion dominated the net cooling effects of non-agricultural sources. Higher atmospheric $NO_x$ burden not only induced a higher nitrate aerosol burden but also significantly decreased the atmospheric mole fraction of $CH_4$ through increasing atmospheric OH. The warming effect of non-agricultural $N_2O$ was amplified by the synchronous decline in atmospheric $CH_4$ because of their interactions in the radiative transfer[36]. We quantified this unmasking effect on the $N_2O$ radiative forcing by decreasing $CH_4$ using a sensitivity experiment with GEOS-Chem-RRTMG (Extended Data Table 1) as a decrease in the non-agricultural $N_2O$ radiative forcing from +0.11 to + 0.07 W m$^{-2}$.

Because of the large difference in lifetimes between long-lived greenhouse gases (such as $CO_2$ and $N_2O$) and short-lived reactive gases (such as $NH_3$ and $NO_x$), regional differences in the emissions matter more for the short-lived gases. As a result, the strength of the regional anthropogenic Nr climate effects shows large spatial variations (Figs. 3 and 4b). In particular, the aerosol effects tend to be strong in regions with high levels of air pollution, such as India and eastern China but show negligible radiative forcing on open ocean because of their short lifetime and limited atmospheric transport.

It should be noted that the radiative forcings attributed to agricultural and non-agricultural Nr are affected by the nonlinearity in the chemistry of aerosol formation, which results in a somewhat stronger net cooling effect from the sum of the individual effects (−0.38 W m$^{-2}$) compared to the combined estimate (−0.34 W m$^{-2}$ in Fig. 3f). The direct radiative forcing of nitrate aerosol is not only weakened with substantial $NO_x$ reductions in the no_nonagriNr experiment but also reduced by the decline in the ammonium nitrate aerosol associated with significant $NH_3$ emission reduction in the no_agriNr experiment (Methods; Extended Data Fig. 3 and Supplementary Table 5). The $NO_x$ reduction further affects the concentrations of atmospheric oxidants such as $O_3$ and OH and reduces the formation of sulfate aerosol in no_nonagriNr experiment (Supplementary Table 5; Methods). Nevertheless, this nonlinearity in aerosol chemistry does not influence the ranking or overall magnitude of the factors by which Nr influences radiative forcing.

## Scenarios of future Nr climate effects

To illustrate the likely consequences of potential future changes in anthropogenic Nr, we next use the understanding gained in the previous section in a simplified analysis using three representative scenarios from the shared socioeconomic pathways (SSPs; Methods). The SSP 1-2.6 assumes an 'Nr cleaner' scenario with strong reduction in fossil-fuel-based $NO_x$ emission but relatively unchanged magnitudes of fertilizer and manure application to meet global food demands (Extended Data Fig. 4). These Nr-related emissions changes lead to a net warming effect of +0.09 W m$^{-2}$ by the 2050s relative to 2019 dominated by the increased $CH_4$ lifetime and a decreased direct aerosol effect (Fig. 5a). In the SSP 3-7.0 scenario, the future global total fossil fuel sources of Nr remain close to the 2019 level, resulting in similar magnitude of global aerosol forcing but potentially various trends among different regions. Enhanced fertilizer and manure applications increase $N_2O$ emissions and lead to a stronger $N_2O$ warming effect of +0.06 W m$^{-2}$ in the 2050s relative to 2019, which is compensated by the cooling effects of increased aerosol loadings (−0.03 W m$^{-2}$ enhancement in 2050s relative to 2019) and enhanced terrestrial carbon sequestration (−0.04 W m$^{-2}$ enhancement in the 2050s relative to 2019). However, bounding assumptions on the magnitude of N saturation (Supplementary Information Section 2.1 and Supplementary Fig. 4) suggest that carbon sequestration effect might be overestimated by about 0.02 W m$^{-2}$. Finally, the SSP 5-8.5 scenario predicts a generally unchanged level of anthropogenic Nr compared to 2019, thus compensating changes in climate forcing. These results imply that stronger reductions in greenhouse gases emissions are required accompanied by the 'clean-Nr' scenario to achieve both environmental benefits and climate change mitigation.

The magnitude of the estimated radiative forcing is associated with uncertainties in each individual compound or process (Supplementary Information Section 2) but also the unavoidable ambiguity in defining the scope of anthropogenic impacts. Here we adapted a straightforward but conservative definition with only direct Nr inputs by anthropogenic activities. However, other human-induced factors, such as elevated $CO_2$ and LUC, as well as the climate change and associated impacts (for

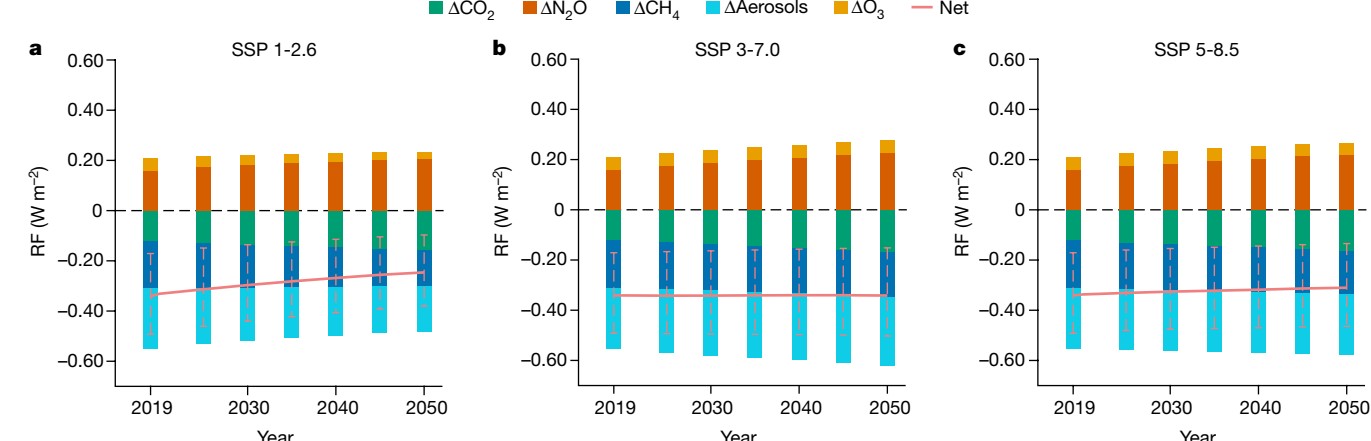

**Fig. 5 | Global direct radiative forcings induced by future scenarios of anthropogenic Nr. a–c,** Global direct radiative forcing relative to pre-industrial concentrations (1850) in response to changes in anthropogenic Nr inputs following SSP 1-2.6 (**a**), SSP 3-7.0 (**b**) and SSP 5-8.5 (**c**) scenarios. The net changes in radiative forcing are shown as solid orange lines. The dashed purple lines indicate no direct radiative forcing change relative to pre-industrial times. The error bars were calculated by the percentage ranges in direct radiative forcing derived from the historical estimates (Supplementary Information Section 1.3).

example, wildfire) can have substantial impacts on biogeochemical cycling, including C[29], water[37] and N cycles, thus making it challenging to unambiguously identify the contributions from anthropogenic Nr. Although these indirect effects might amplify the overall climate effects of anthropogenic Nr, the NMIP2-ensemble simulations suggest that these effects on the C or N cycle are not as significant as the overall direct anthropogenic Nr effect.

In this study, several processes, including the influences of aerosols or $O_3$ on terrestrial carbon fluxes, aerosol–cloud interactions, N addition effects on soil $CH_4$ uptakes and N fertilization on marine biogeochemistry were not included because of the likely small effect on climate or uncertainty to quantify the global effect (Supplementary Information Section 2.5). For the effects we examined in this study, on the one hand, the future $CO_2$ cooling due to $CO_2$ uptake on land may be overestimated in our study because we omit the contribution of fossil-fuel-based $CO_2$ emissions from N fertilizer production by the Haber–Bosch method and, more importantly, terrestrial ecosystems exposed to high chronic N additions may become N saturated within the next few decades and contribute less to terrestrial C storage (Supplementary Information Section 2.1). Uncertainties also remain in quantifying soil $N_2O$, $NO_x$ and $NH_3$ emissions (Supplementary Information Sections 2.2 to 2.4). On the other hand, the negative radiative forcing of nitrate aerosol may be overestimated, as the GEOS-Chem model tends to overestimate nitrate aerosol concentrations[38–40]. Furthermore, changes in $NO_x$ can further influence the formation of organic aerosols by altering atmospheric oxidation capacity and aerosol yields[41–43], which are not examined in this study given the large uncertainty in simulating corresponding chemical processes. To reduce uncertainties and gain a more comprehensive understanding of potential feedbacks, the development of more integrative Earth system models including key interactions among processes of terrestrial and marine biogeochemistry, atmospheric chemistry, climate dynamics and radiative processes would be required.

Comprehensively assessing the global climate effects of anthropogenic Nr has been challenging for decades considering the complexity in atmospheric physical and chemical processes as well as the terrestrial biogeochemical cycles. Bringing together biosphere and atmospheric chemistry modelling, our results contribute to a clearer picture that at present the combined effects from short-lived and long-term Nr-related climate forcers is a global net cooling with strong regional variations. The enhanced consistency allows us to estimate the net radiative forcing of anthropogenic Nr at −0.34 [−0.20, −0.50] W m$^{-2}$, which improves the robustness relative to the only other available estimate based on literature review alone (−0.24 [ + 0.2, −0.50] W m$^{-2}$) (ref. 18). Future reductions in anthropogenic Nr will likely weaken this net cooling effect mainly through a reducing atmospheric aerosol burden and an increased $CH_4$ lifetime, whereas the future effect of warming from fertilizer-induced $N_2O$ emissions will remain or even increase. Our findings thus imply that to alleviate the negative environmental effects of Nr without larger rates of climate change, stronger reductions in the emission of greenhouse gases $CO_2$ and $CH_4$ need to be implemented concurrently with Nr reductions.

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

## Methods

A summary for the data and methods in this study is given in Extended Data Fig. 6. Here we introduce each part in detail.

### NMIP2 multimodel dataset

The NMIP2 ensemble included eight terrestrial biosphere models with comprehensive descriptions of terrestrial carbon and nitrogen cycles, driven by harmonized climate, land use and nitrogen cycle drivers. Each NMIP2 member provided data at a spatial resolution of $0.5° \times 0.5°$ from 11 transient, factorial simulations (Extended Data Table 2) to disentangle the contributions of N fertilizer use, manure application, N deposition, irrigation, LUC, $CO_2$ elevation and climate changes from pre-industrial times (1850) to present day (2020). Climate data were generated from CRU-JRA55 6-h forcing[44]; historical $CO_2$ concentrations were derived from ice core $CO_2$ data and NOAA annual observations. Anthropogenic Nr deposition was generated by international global atmospheric chemistry/stratospheric processes and their role in climate chemistry–climate model initiative (https://www.sparc-climate.org/activities/ccm-initiative/). Nitrogen fertilizer and manure application data were specially generated for NMIP2 based on high-resolution (5 arcmin) harmonized data on the history of anthropogenic nitrogen inputs[45]. Land use changes were generated from land-use harmonization 2 project[46,47], surveys by the International Fertilizer Industry Association and the Food and Agricultural Organization and the Global Livestock Impact Mapping System. For more details on NMIP2 configuration and input data, refer to refs. 28,48. To calculate the ensemble mean, we used output from eight models for NBP and $N_2O$ but could only rely on six models for soil $NH_3$ and three for soil $NO_x$ emissions, respectively (Extended Data Table 3).

### CEDS inventory

The CEDS inventory was generated by integrating existing global, regional and country-specific inventories with a consistent and reproducible methodology, representing monthly grid-level anthropogenic emissions of chemically reactive gases (for example, carbon monoxide (CO), $NH_3$, $NO_x$, sulfur dioxide and non-methane volatile organic compounds), carbonaceous aerosols (black carbon and organic carbon) and greenhouse gases ($CO_2$, $CH_4$ and $N_2O$) from 1750 to present day (updating to the latest year)[32]. For each gas or aerosol, the anthropogenic emissions were divided into eight sectors, including non-combustion agricultural, energy transformation and extraction, industrial combustion, residential, international shipping, solvents, transportation and waste disposal. Here we accessed the CEDS data from a postprocessed version by GEOS-Chem support team, which made several modifications to fit the GEOS-Chem configurations (http://wiki.seas.harvard.edu/geos-chem/index.php/CEDS_anthropogenic_emissions).

### Integration of Nr emission data

The effect of anthropogenic fertilization, manure application, N deposition, irrigation, LUC, $CO_2$ elevation and climate changes on simulations of NBP, $N_2O$ and $NO_x$ in the NMIP2 ensemble are quantified on the basis of the differences among a series of sensitivity experiments (Extended Data Table 2). The contribution of LUC is quantified by the difference between the SH12 and SH11 experiment (rather than differences between SH1 and SH6) to avoid the confounding effects from changes in fertilizer and manure application. $N_2O$ and NBP fluxes are accessible for all of the eight NMIP2 members, whereas the $NO_x$ flux is only available with CLASSIC, OCN and ORCHIDEE (Extended Data Table 3).

The $NH_3$ emission estimate of 39.0 TgN yr$^{-1}$ by the NMIP2 ensemble, which accounts for agricultural $NH_3$ soil emissions but not those emissions from livestock manure, is close to the CEDS agricultural $NH_3$ emissions (38.2 TgN yr$^{-1}$) for the year 2019. However, the large

intermodel variability (Supplementary Fig. 1d) makes the direct use of these simulations to quantify the anthropogenic effect susceptible to biases in individual models. Therefore, we retained the original CEDS agricultural $NH_3$ emission in this study and attribute soil $NH_3$ emission changes by first applying a fixed ratio (48%) on the total agricultural $NH_3$ emissions in 2019, whereas the rest (52%) is led by livestock according to ref. 35 and then scaling the anthropogenic Nr influence on soil $NH_3$ emissions according to the temporal evolution of soil $NH_3$ emissions in the NMIP2 ensemble.

Finally, we integrated the CEDS anthropogenic inventory and NMIP2 multimodel data to represent the anthropogenic Nr emissions. Anthropogenic emissions from fossil fuel combustion were taken as the sum of all sectors in CEDS inventory, except for the agricultural emissions. Besides fossil fuel combustion and soil Nr emissions, we examined the biomass burning emissions of Nr based on ref. 49. This dataset was used to provide the historical biomass burning emissions in CMIP6 from 1850 to 2015 and showed similar magnitude and variabilities as GFED4.1 inventory in the past decades. The historical annual biomass burning $N_2O$ emissions were used to establish $N_2O$ box model (see below). However, because the biomass burning emissions of $NO_x$ and $NH_3$ showed little differences between the present day and pre-industrial period, here we neglected such differences and used the same present-day biomass burning emissions in all the GEOS-Chem experiments.

### $CO_2$, $N_2O$ and $CH_4$ box models

To estimate the effects of anthropogenic Nr on atmospheric $CO_2$ concentrations, we used atmospheric box models based on the framework of ref. 9. The changes in atmospheric $CO_2$ concentrations induced by anthropogenic Nr effects on terrestrial carbon fluxes were represented by:

$$\Delta CO_2 = -\sum_{yr=1850}^{2019} (NBP_{fertilizer,yr} + NBP_{manure,yr} + NBP_{Ndep,yr}) \times \frac{\alpha}{\delta_{CO_2}} \quad (1)$$

where $\Delta CO_2$ indicates changes in atmospheric $CO_2$ concentrations (ppmv) from 1850 to 2019 because of anthropogenic Nr. The accumulated NBP induced by fertilizer and manure applications and N deposition during 1850–2019 was calculated from NMIP2 ensemble mean (Extended Data Table 2). The $\delta_{CO_2}$ was 2.12 PgC ppmv$^{-1}$ following ref. 50. The partitioning constant $\alpha$ accounting for the ocean-borne fraction of atmospheric $CO_2$ increase was determined to be 0.61 given the historical (1850–2019) increases in the atmosphere (235 PgC) and ocean (150 PgC) carbon estimated from the global carbon budget[29].

The $N_2O$ box model was also based on ref. 9:

$$\frac{d[N_2O]_{yr}}{dt} = \frac{N_2O_{FF} + N_2O_{soil} + N_2O_{BB} + N_2O_{AREC} + N_2O_{chem} + N_2O_{NREC} + N_2O_{ocean}}{\delta_{N_2O}}$$
$$- \frac{[N_2O]_{yr}}{\tau} \quad (2)$$

where the $\frac{d[N_2O]_{yr}}{dt}$ was the annual increasing rate of atmospheric $N_2O$ concentrations at the yr year. $N_2O$ sources from fossil fuel (FF) combustion, soil, biomass burning (BB), anthropogenic emissions from river, estuaries and coastal zones (AREC), atmospheric chemistry, natural emissions from river, estuaries and coastal zones (NREC) as well as open ocean were summarized in Extended Data Table 4 (refs. 9,49,51,52). The $\delta_{N_2O}$ was set as 4.8 TgN ppbv$^{-1}$ following ref. 9. The $[N_2O]_{yr}$ indicated the surface atmospheric $N_2O$ concentrations at the yr year and $\tau$ was the perturbation lifetime of atmospheric $N_2O$, taken as 116 years (ref. 52). The simulated global surface $N_2O$ concentrations were shown in Extended Data Fig. 5a.

We used a $CH_4$ box model described by ref. 53 to examine the effects of changes in $NO_x$ emissions on $CH_4$ concentrations due to changed atmospheric OH concentrations:

$$\frac{d\,[CH_4]_{yr}}{dt} = -\frac{1}{\tau_{CH_4}}[CH_4]_{yr} + \frac{E_{CH_4,yr}}{\delta_{CH_4}} \tag{3}$$

where $[CH_4]_{yr}$ indicated the global mean $CH_4$ concentrations at the yr year. $E_{CH_4,yr}$ was the total $CH_4$ emissions at the yr year, which was calculated by summing $CH_4$ emissions by anthropogenic activities based on CEDS inventory, biomass burning emissions based on ref. 49 and natural sources with an estimate of 230 Tg yr$^{-1}$. The $\delta_{CH_4}$ was set as 2.78 Tg ppb$^{-1}$ (ref. 54). The $CH_4$ lifetime $\tau_{CH_4}$ was estimated by:

$$\frac{1}{\tau_{CH_4}} = \frac{1}{\tau_{OH}} + \frac{1}{\tau_{strat}} + \frac{1}{\tau_{soil}} + \frac{1}{\tau_{trop-cl}} \tag{4}$$

where $\tau_{strat}$, $\tau_{soil}$ and $\tau_{trop-cl}$ were set as constant numbers of 120, 150 and 200 years, respectively, to represent $CH_4$ lifetime led by stratospheric loss, soil uptake and tropospheric chlorine reactions. Parameter $\tau_{OH}$ was the $CH_4$ lifetime due to the OH oxidation, which was calculated by:

$$\frac{1}{\tau_{OH}} = \frac{1}{\tau_{OH}^0} \times \left( \left( \frac{[CH_4]_{yr}}{[CH_4]_0} \right)^{S_{OH}} \times e^{(S_{NO_x} \times \Delta E_{NO_x} + S_{CO} \times \Delta E_{CO} + S_{VOC} \times \Delta E_{VOC})} + S_T \times \Delta T \right) \tag{5}$$

where $\tau_{OH}^0$ and $[CH_4]_0$ were the references of $CH_4$ lifetime and concentrations. Here we selected the year of 2005 as the reference year with $\tau_{OH}^0$ of 11.17 years and $[CH_4]_0$ of 1,783.36 ppb. The sensitivity factor $S_{OH}$ was −0.31 following ref. 55. $S_{NO_x}$, $S_{CO}$ and $S_{VOC}$ were set as 0.0042 (Tg[N] yr$^{-1}$)$^{-1}$, −0.000105 (Tg[CO] yr$^{-1}$)$^{-1}$ and −0.000315 (Tg[VOC] yr$^{-1}$)$^{-1}$, respectively, following Table 4.11 of ref. 56. The emission changes in $NO_x$ ($\Delta E_{NO_x}$), CO ($\Delta E_{CO}$) and volatile organic compounds (VOC) ($\Delta E_{VOC}$), which included changes in anthropogenic emissions from CEDS and biomass burning emissions from ref. 49, were calculated by the differences between the yr year and the reference year (2005), respectively. The temperature effects on atmospheric $CH_4$ loss rates were expressed by multiplying factor $S_T$ of 0.0316 K$^{-1}$ (ref. 56) and changes in global surface mean temperature $\Delta T$ relative to the reference year (2005). The simulated global mean surface $CH_4$ concentrations were shown in Extended Data Fig. 5b.

### The GEOS-Chem-RRTMG model and sensitivity experiments

We used the state-of-art global three-dimensional chemical transport model GEOS-Chem (v.12.0.0) with a fully coupled $NO_x$–$O_x$–hydrocarbon–aerosol chemistry mechanism[57–60] to simulate $NH_3$ and $NO_x$ concentration and associated aerosol loadings and $O_3$ at a horizontal resolution of 2° latitude × 2.5° longitude and a vertical resolution of 47 layers from surface to 0.1 hPa level. The photolysis rates were computed by Fast-JX scheme[58]. Aerosol concentrations were calculated online by the ISORROPIA II package[61]. Version two of modern era retrospective-analysis for research and application (MERRA2) assimilated meteorological data was used to drive the GEOS-Chem model. Atmospheric concentrations of the long-lived greenhouse gases $CO_2$, $CH_4$ and $N_2O$ were derived from simple atmospheric box models (see above). On the basis of the simulated concentrations of tracers, we diagnosed direct radiative forcing of Nr-related compounds using the offline RRTMG in GEOS-Chem[62]. The annual-mean direct radiative forcing in the year 2019 was estimated from a year-long simulation after a 6 month spin-up period. In particular, GEOS-Chem fully considers the nonlinearity of inorganic aerosol chemistry, in which sulfate aerosol has higher priority than nitrate aerosol in aerosol formation when ammonia gas is limited in the atmosphere. Changes in the atmospheric $NO_x$ loading can also affect oxidation of sulfur dioxide by perturbating atmospheric oxidants, such as $O_3$ and OH. As a result, the sulfate aerosol loadings could also be perturbed by changes in $NO_x$ emissions, despite the fact that the sulfur dioxide emissions are identical in all our experiments. We thus use the sum of direct radiative forcing of

ammonium ($NH_4^+$), nitrate ($NO_3^-$) and sulfate ($SO_4^{2-}$) aerosols to represent the aerosol climate effects induced by anthropogenic Nr.

We designed four sensitivity experiments to isolate the anthropogenic Nr effects on climate, in which each experiment was driven by the same meteorological forcing but with different $NH_3$ and $NO_x$ emissions as well as $CO_2$, $N_2O$ and $CH_4$ concentrations. The $NH_3$ and $NO_x$ emissions in each experiment are given in Extended Data Fig. 3, whereas $CO_2$, $N_2O$ and $CH_4$ concentrations were summarized in Extended Data Table 1. An extra sensitivity experiment, which followed No_nonagriNr run but assumed $CH_4$ concentrations as in the CTRL run, was designed to quantify the effect of changes in $CH_4$ concentrations on the radiative forcing of $N_2O$ due to non-agricultural emission changes. We estimated the uncertainty in the radiative forcing estimates by propagating the variation across NMIP2 ensemble projections into atmospheric concentrations and thus radiative forcing. The full uncertainty analysis and uncertainty discussions are detailed in the Supplementary Information and rely on refs. 9–11,18,28,32–35,52,62–95.

### Linear extrapolation of climate effects under the SSP scenarios

We extrapolated the future climate effects due to changes in anthropogenic Nr under three representative SSP scenarios (SSP 1-2.6, 3-7.0 and 5-8.5). The future fossil fuel emissions and N deposition were from the input4MIPs dataset (https://esgf-data.dkrz.de/projects/input-4mips-dkrz/). Future fertilizer and manure applications were based on the IMAGE predictions until 2050[96]. To maintain consistency in this study, the future Nr-related sources were scaled to 2019 levels for each dataset (Extended Data Fig. 4). Because the future fossil-fuel-based emission of $N_2O$ is not included in input4MIPs, the future development of this source of $N_2O$ was scaled to the future development of fossil-fuel-based $NO_x$.

To estimate the magnitude of climate effects of anthropogenic Nr under the SSP scenarios, we built a simple linear framework based on the following assumptions. (1) The change in radiative forcing of atmospheric greenhouse gas attributable to Nr-related changes was linearly related to their change in atmospheric concentrations, whereas the direct radiative forcing of short-lived gases or aerosols was linearly related to the total emissions of precursors[11,97,98] at the corresponding year. (2) The effects of anthropogenic Nr on soil–gas fluxes were linearly determined by anthropogenic Nr addition, including both fertilizer/manure application and N deposition. Then a simple model was established based on the GEOS-Chem diagnosed direct radiative forcing of individual compound to calculate the radiative forcing relative to 1850:

$$
\begin{aligned}
RF\_Nr\_CO_{2\,yr} = {}& RF\_Nr\_CO_2\,2019 \\
& + \sum_{yr=2020}^{t} (NBP_{fertilizer,yr} + NBP_{manure,yr} + NBP_{Ndep,yr}) \\
& \times \frac{\alpha}{\delta_{CO_2}} \times S_{CO_2}
\end{aligned} \tag{6}
$$

$$RF\_Nr\_N_2O_{yr} = RF\_Nr\_N_2O_{2019} + ([N_2O]_{yr} - [N_2O]_{2019}) \times S_{N_2O} \tag{7}$$

$$RF\_Nr\_CH_{4\,yr} = RF\_Nr\_CH_{4\,2019} + ([CH_4]_{yr} - [CH_4]_{2019}) \times S_{CH_4} \tag{8}$$

$$RF\_Nr\_aerosol_{yr} = \frac{NO_{x\,yr} + NH_{3\,yr}}{NO_{x\,2019} + NH_{3\,2019}} \times RF\_Nr\_aerosol_{2019} \tag{9}$$

$$RF\_Nr\_O_{3\,yr} = \frac{NO_{x\,yr}}{NO_{x\,2019}} \times RF\_Nr\_O_{3\,2019} \tag{10}$$

Where the $RF\_Nr\_CO_{2yr}$, $RF\_Nr\_N_2O_{yr}$, $RF\_Nr\_CH_{4yr}$, $RF\_Nr\_aerosol_{yr}$ and $RF\_Nr\_O_{3yr}$ represent the direct radiative forcing associated with anthropogenic Nr of each gas at the yr year relative to 1850. The values in 2019 were derived from the differences between CTRL_2019 and No_allNr

experiments (−0.12 W m⁻², +0.16 W m⁻², −0.19 W m⁻², −0.24 W m⁻² and +0.05 W m⁻², respectively; Fig. 3). The sensitivities ($S_{CO_2}$, $S_{N_2O}$ and $S_{CH_4}$) of radiative forcing to greenhouse gas concentrations were derived from the other eight GEOS-Chem sensitivity experiments (Supplementary Information Section 1.3 and Supplementary Table 2).

In particular, we calculated the reduction effect as follows:

1. $NBP_{fertilizer,yr}$, $NBP_{manure,yr}$ and $NBP_{Ndep,yr}$ represented the NBP contributed by fertilizer, manure and N deposition in the yr year, which is calculated by multiplying the NMIP2 ensemble mean present-day (average of 2015–2019) contributions and the corresponding scaling factors in Extended Data Fig. 4.
2. The $N_2O$ and $CH_4$ concentrations in the yr year ($[N_2O]_{yr}$ and $[CH_4]_{yr}$) were derived by the simple $N_2O$ and $CH_4$ box models (equation (2) and equations (3)–(5)) starting from $[N_2O]_{2019}$ ($N_2O$ concentrations in CTRL_2019 experiments) and $[CH_4]_{2019}$ ($CH_4$ concentrations in CTRL_2019 experiments), respectively. $N_2O$ (in $N_2O$ box model) and $NO_x$ (in $CH_4$ box model) emissions from both fossil fuel combustion and anthropogenic Nr-induced soil emissions were reduced relative to emissions in 2019 with the scaling factors accordingly (Extended Data Fig. 4), whereas the other sources were kept the same as 2019.
3. For short-lived compounds (aerosols and $O_3$), $NO_{x,yr}$ (or $NH_{3,yr}$) indicated the $NO_x$ (or $NH_3$) emissions from both fossil fuel and soil by applying the scaling factors on each sector (Extended Data Fig. 4) in the yr year.

## Data availability

The CEDS inventory used in GEOS-Chem can be downloaded at https://ftp.as.harvard.edu/gcgrid/data/ExtData/HEMCO/CEDS/. The NMIP2 model outputs and the GEOS-Chem outputs in this study are available at Zenodo (https://doi.org/10.5281/zenodo.10032973)[99]. The base maps in all figures are based on the default global map in the NCAR Command Language (NCL).

## Code availability

The GEOS-Chem-RRTMG source codes can be accessed at https://github.com/geoschem/geos-chem. Data analysis and visualization are conducted by NCL. Scripts are available at Zenodo (https://doi.org/10.5281/zenodo.11179126)[100].

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

**Acknowledgements** C.G. and S.Z. acknowledge support from the European Commission H2020 programme (grant no. 101003536; ESM2025). H.T. and N.P. acknowledge funding support from the US National Science Foundation (grant no. 1903722) and USDA CBG project no. TENX12899. S.P. acknowledges funding support from the US National Science Foundation (grant no. 1922687) and the US Department of the Treasury (grant no. DISL-MESC-ALCOE-06). A.I. acknowledges support from the JSPS KAKENHI (grant no. 21H05318). Q.Z. is supported by Reducing Uncertainties in Biogeochemical Interactions through Synthesis and Computation (RUBISCO) Scientific Focus Area, Office of Biological and Environmental Research of the US Department of Energy Office of Science. C.G. and S.Z. thank D. Olivié and colleagues for help in building the $CH_4$ box model (https://zenodo.org/records/5293940)[53].

**Author contributions** C.G. and S.Z. designed the study. C.G. performed the GEOS-Chem simulations and data analysis. H.L. assisted the GEOS-Chem simulation. H.T., N.P. and S.P. led the NMIP2 projects. H.T., N.P., S.P., A.I., A.K.J., S.K.-G., F.J., Q.S., H.S., N.V., Q.Z., C.P., F.M., F.H.M.T. and S.Z. together contributed to the simulation of terrestrial biosphere models in NMIP2. C.G. and S.Z. wrote the manuscript. All authors contributed to reviewing or editing the manuscript.

**Funding** Open access funding provided by Max Planck Society.

**Competing interests** The authors declare no competing interests.

**Additional information**
**Correspondence and requests for materials** should be addressed to Cheng Gong.

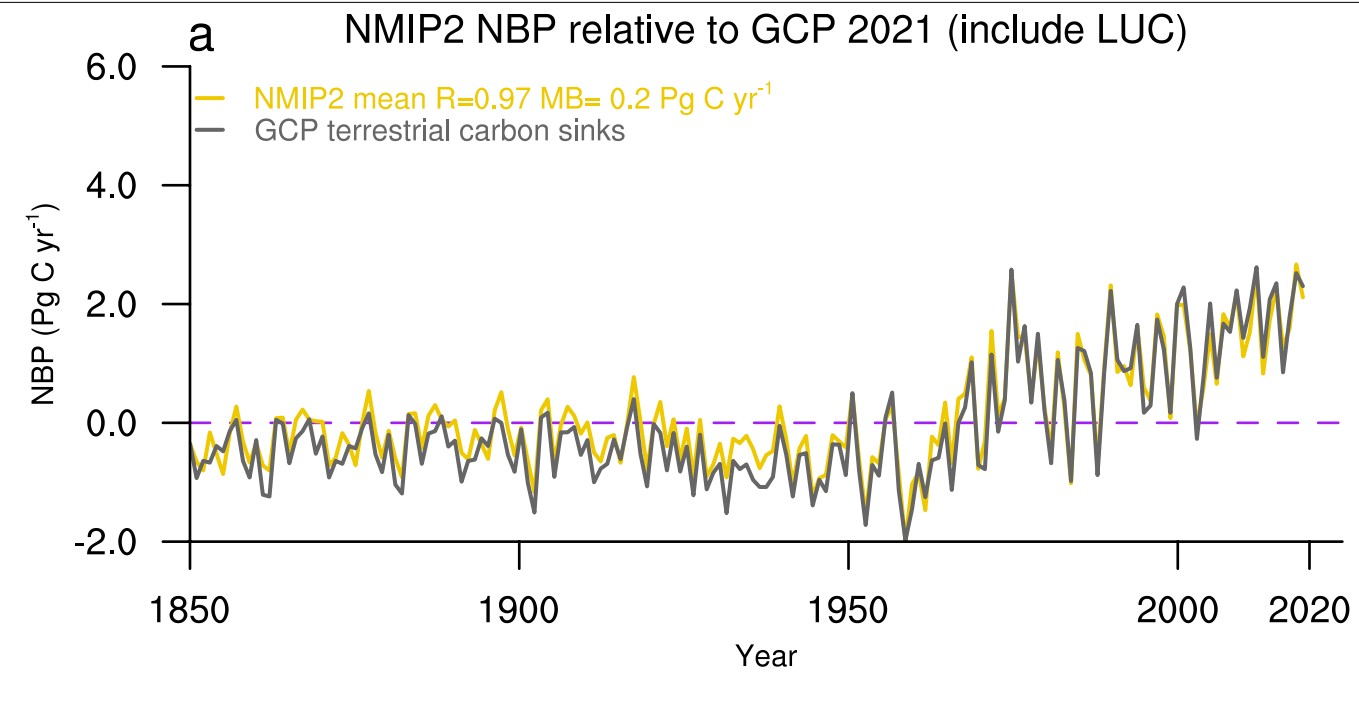

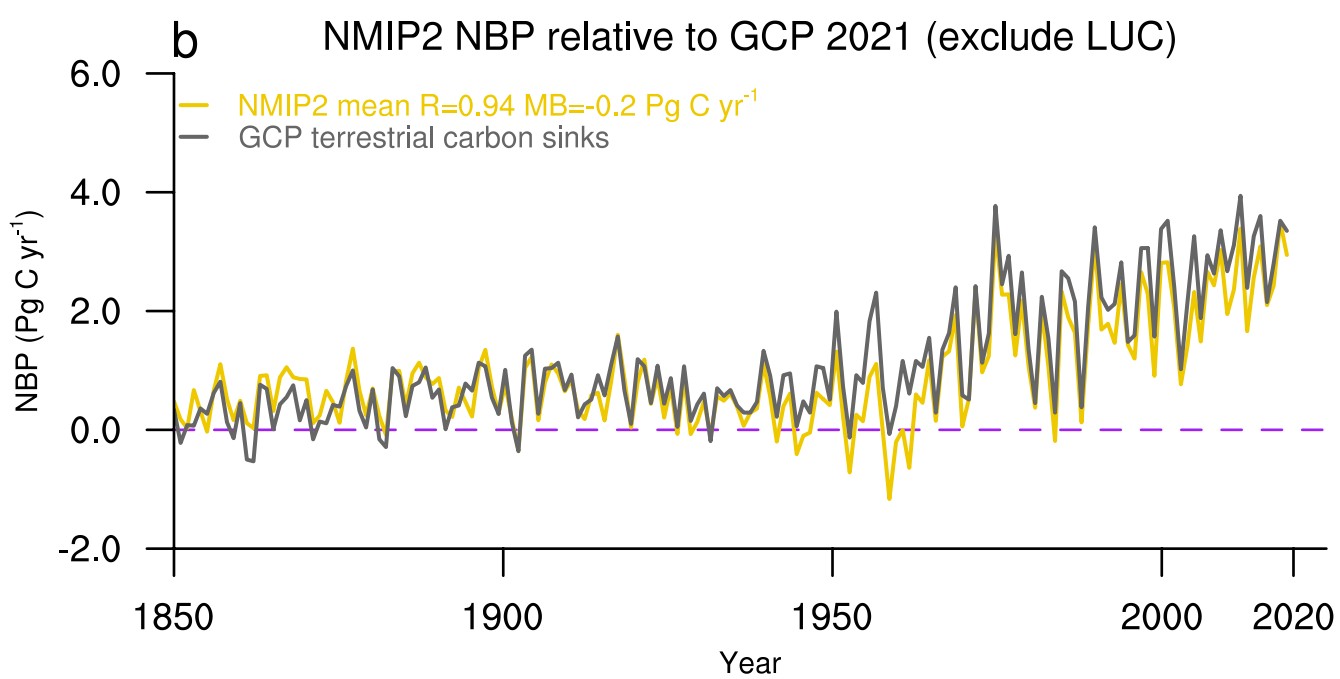

**Extended Data Fig. 1 | Comparison of net biome productivity (NBP) estimates from the NMIP2 ensemble and the Global Carbon Project (GCP2021).** The yellow and gray lines represent the NBP of NMIP2 ensemble and GCP2021[29] estimate, respectively. **a** and **b** show time series with and without the effect of land use change (LUC), respectively. The correlation coefficient (R) and mean bias (MB) of annual NBP between NMIP2 ensemble and GCP2021 are also given.

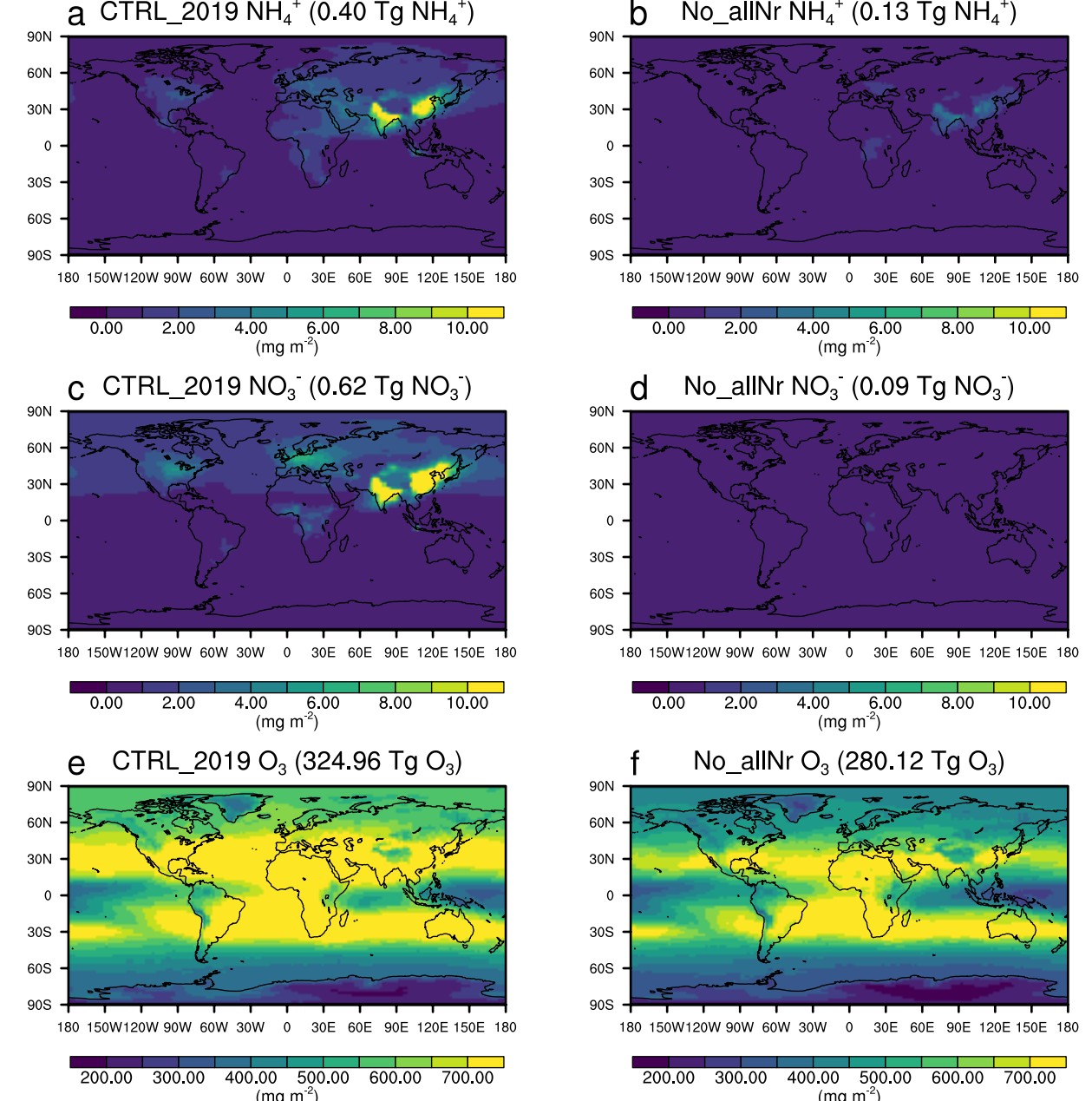

**Extended Data Fig. 2 | Simulated global burden of short-lived atmospheric compounds.** The column burden of **a-b** ammonium aerosol, **c-d** nitrate aerosol and **e-f** $O_3$ in CTRL_2019 and No_allNr experiments are given, respectively. The column burden of each compound was accumulated over the whole atmospheric column in GEOS-Chem based on the annual mean concentrations.

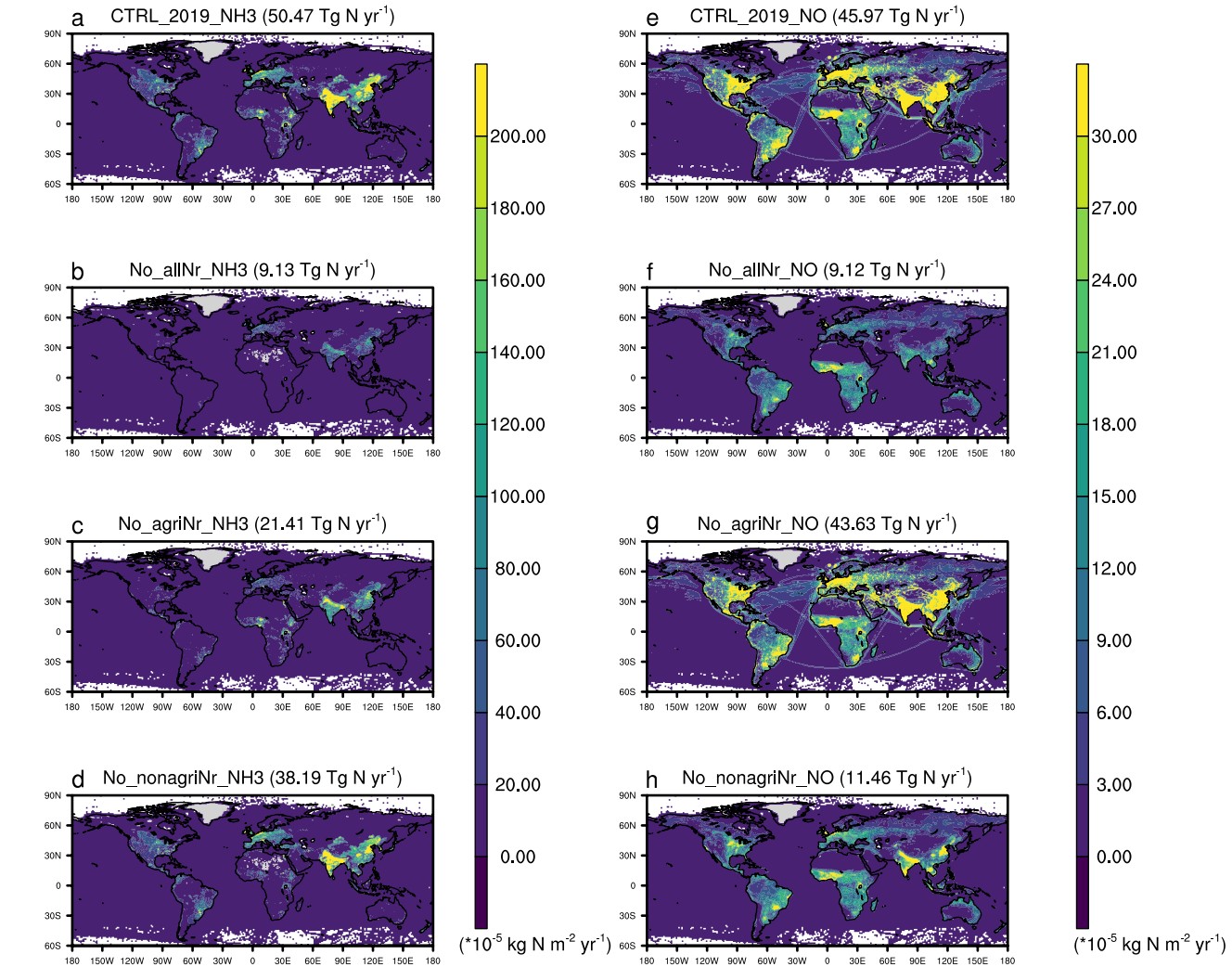

**Extended Data Fig. 3 | Global patterns of anthropogenic NH$_3$ and NO$_x$ emissions. a-d** The NH$_3$ emissions and **e-h** NO$_x$ emissions in the CTRL_2019, No_allNr, No_agriNr, and No_nonagriNr sensitivity experiments, respectively, are shown. The sub-title of each panel gives the global total emissions in the year of 2019.

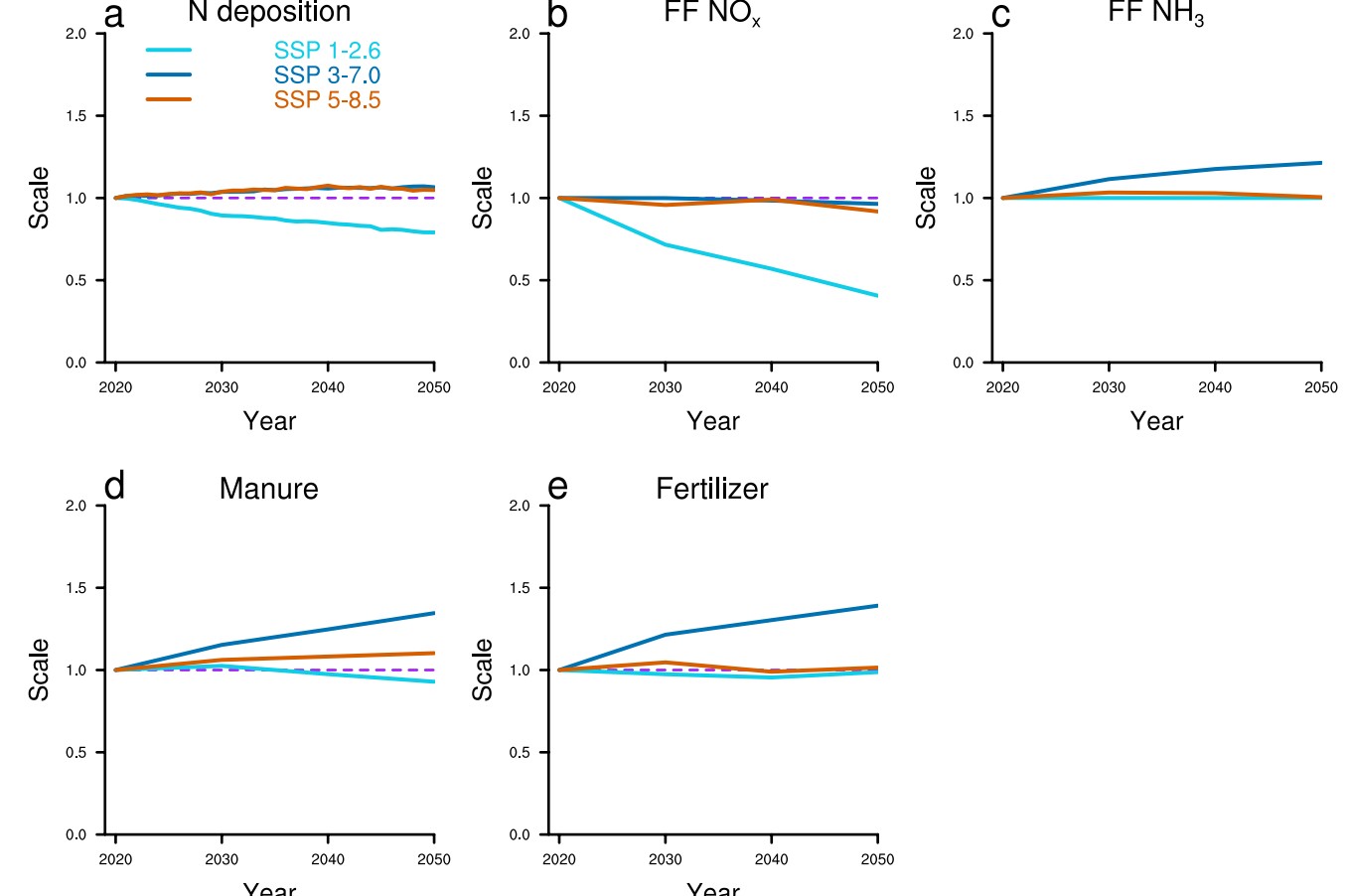

**Extended Data Fig. 4 | Development of global anthropogenic Nr inputs in future SSP scenarios. a-e** indicate the changes of anthropogenic Nr inputs from N deposition, fossil fuel $NO_x$, fossil fuel $NH_3$, manure and fertilizer applications, respectively. For each SSP scenario, the annual scale factors were calculated by the ratios of future anthropogenic Nr to the 2019 levels, which is indicated by the dashed purple lines. The N deposition and fossil fuel data are from the input4MIPs (https://esgf-data.dkrz.de/projects/input4mips-dkrz/). Manure and fertilizer predictions are obtained from Mogollon, et al.[96].

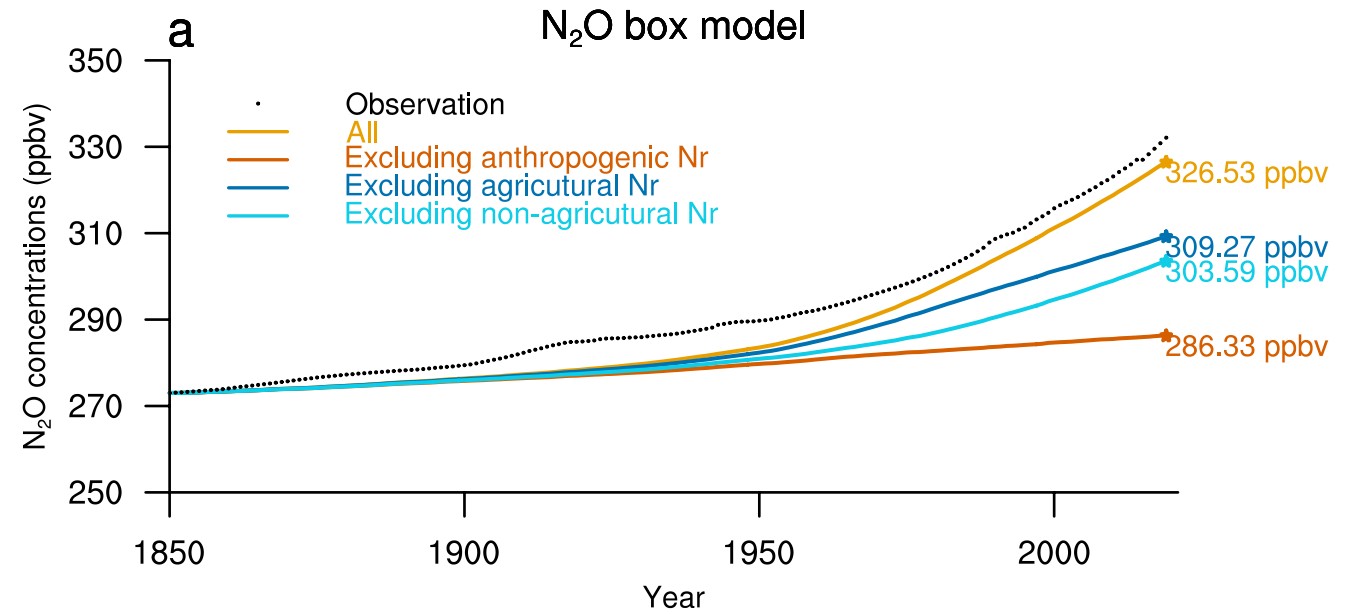

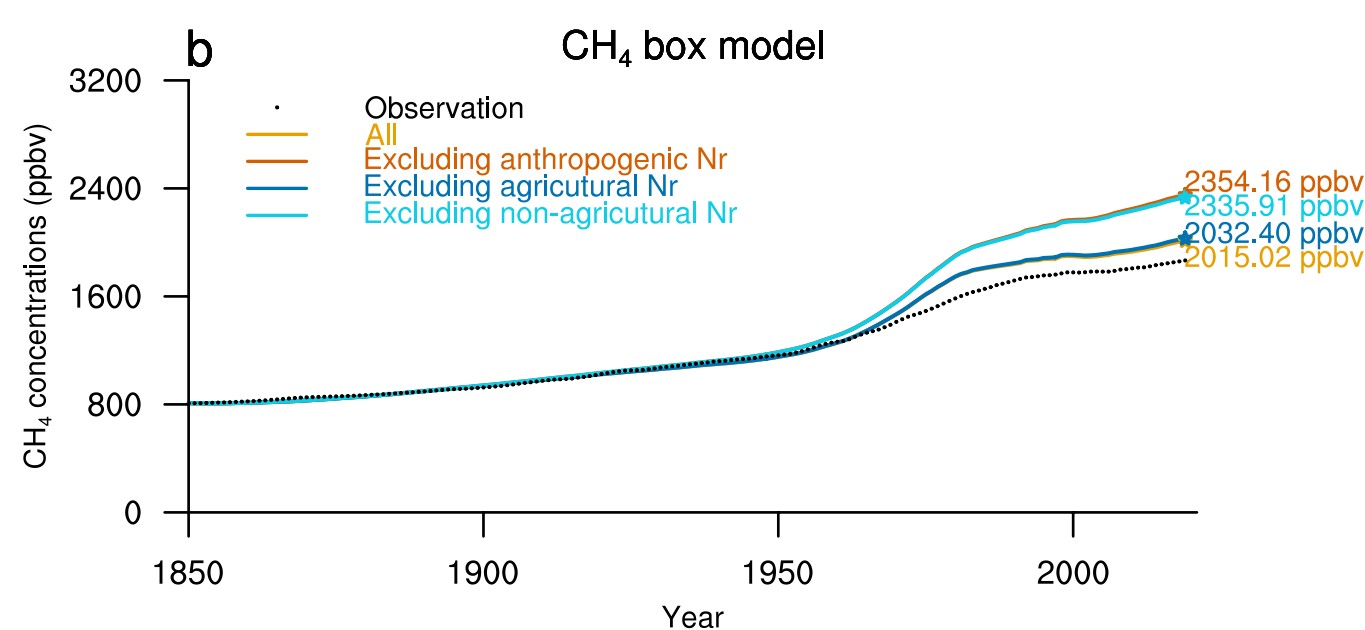

**Extended Data Fig. 5 | Global N₂O and CH₄ concentrations during 1850–2019 based on the box models. a** N₂O concentration; and **b** CH₄ concentration. The black dots indicate the historical annual N₂O and CH₄ concentrations, as observed from ice core, firn and atmospheric measurements[101]. Lines with different colors represent the different SSP scenarios. Concentration values indicate the global mean concentration in the year of 2019 derived from the box models.

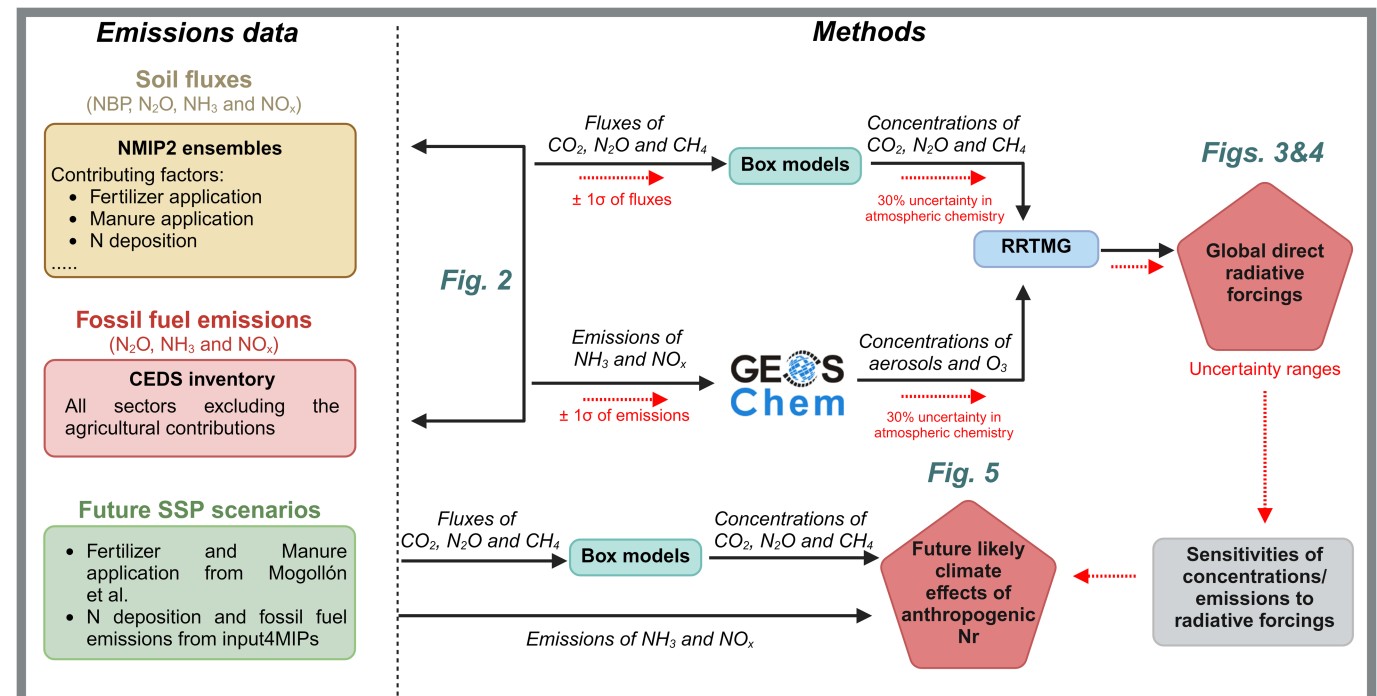

**Extended Data Fig. 6 | Schematic workflow summary of this study.** Solid black arrows indicate main methods or data, as described in Methods. Dashed red arrows indicate the uncertainty analysis and the associated sensitivities to radiative forcing (see SI text S1). The main figures in this study are highlighted accordingly with the figure indexes. This figure is created with BioRender.com.

**Extended Data Table 1 | Summary of the GEOS-Chem sensitivity experiments**

| Experiment name | $NO_x$ emissions | $NH_3$ emissions | $N_2O$ concentrations | $CO_2$ concentrations | $CH_4$ concentrations |
|---|---|---|---|---|---|
| CTRL_2019 | 45.97 Tg N $yr^{-1}$ | 50.47 Tg N $yr^{-1}$ | 326.53 ppbv | 409.9 ppmv | 2015.02 ppbv |
| No_allNr | 9.12 Tg N $yr^{-1}$ | 9.13 Tg N $yr^{-1}$ | 286.33 ppbv | 420.38 ppmv (409.9+10.48) | 2354.16 ppbv |
| No_agriNr | 43.63 Tg N $yr^{-1}$ | 21.41 Tg N $yr^{-1}$ | 309.27 ppbv | 415.19 ppmv (409.9+5.29) | 2032.40 ppbv |
| No_nonagriNr | 11.46 Tg N $yr^{-1}$ | 38.19 Tg N $yr^{-1}$ | 303.59 ppbv | 415.09 ppmv (409.9+5.19) | 2335.91 ppbv |
| Supplement experiment | | | | | |
| No_nonagriNr_lowCH₄ | 11.46 Tg N $yr^{-1}$ | 38.19 Tg N $yr^{-1}$ | 303.59 ppbv | 415.09 ppmv (409.9+5.19) | 2015.02 ppbv |

The global $NO_x$ and $NH_3$ emissions are derived by integrating CEDS and NMIP2 ensembles. Atmospheric concentrations of $N_2O$, $CO_2$ and $CH_4$ are retrieved by the box models. The CTRL_2019 experiment includes all anthropogenic Nr sources; The No_allNr experiment excludes anthropogenic Nr sources of fossil fuel, fertilizer and manure application and N deposition; The No_agriNr experiment excludes anthropogenic Nr sources of fertilizer and manure application; The No_agriNr experiment excludes anthropogenic Nr sources of fossil fuel and N deposition. The livestock $NH_3$ emission is attributed as agricultural sources.

**Extended Data Table 2 | Experiment configurations in NMIP2**

| Experiment name | Fertilizer | Manure | N deposition | Irrigation | Land use changes | CO$_2$ | Climate |
|---|---|---|---|---|---|---|---|
| SH0 | 1850 | 1850 | 1850 | 1850 | 1850 | 1850 | 1901-1920 |
| SH1 | T | T | T | T | T | T | T |
| SH2 | 1850 | T | T | T | T | T | T |
| SH3 | T | 1850 | T | T | T | T | T |
| SH4 | T | T | 1850 | T | T | T | T |
| SH5 | T | T | T | 1850 | T | T | T |
| SH6 | T | T | T | T | 1850 | T | T |
| SH7 | T | T | T | T | T | 1850 | T |
| SH8 | T | T | T | T | T | T | 1901-1920 |
| SH11 | 1850 | 1850 | T | 1850 | 1850 | T | T |
| SH12 | 1850 | 1850 | T | 1850 | T | T | T |
| Factor attribution in this study | SH1-SH2 | SH1-SH3 | SH1-SH4 | SH1-SH5 | SH12-SH11 | SH1-SH7 | SH1-SH8 |

The letter 'T' indicates a transient change as the forcing data from 1850 to 2020. The 1850 or 1901–1920 indicates that the corresponding forcing was fixed in this year or time periods. Climate over 1850–1900 were repeated by 1901–1920 for all experiments due to the missing of data. The last row of the table indicates how the factorial contribution for each factor in this study is calculated from the differences between corresponding experiments.

**Extended Data Table 3 | Summary of the eight terrestrial biosphere models in NMIP2 used in this study**

| | NO$_x$ (3) | NH$_3$ (6) | N$_2$O (8) | NBP (8) | Missing experiments |
|---|---|---|---|---|---|
| CLASSIC | √ | √ | √ | √ | No SH2,4,5 |
| DLEM | × | √ | √ | √ | |
| ELM | × | × | √ | √ | No SH2 |
| ISAM | √ (Excluded) | √ | √ | √ | No SH5 |
| LPX-Bern | × | √ (Excluded) | √ | √ | No SH2,5,12 |
| OCN | √ | √ | √ | √ | No SH5,11,12 |
| ORCHIDEE | √ | √ | √ | √ | No SH5 |
| VISIT | × | √ | √ | √ | |

Some of the models failed to finish all experiments, which were indicated in the last column. Soil NH$_3$ volatilization of LPX-Bern was excluded from the integration of NMIP2 and CEDS since it failed to show factor contributions. Soil NO$_x$ emissions simulated by ISAM was also excluded due to the unreasonably high magnitude. The number of models that we finally used to drive GEOS-Chem model for the corresponding variable was indicated in the top row.

**Extended Data Table 4 | Summary of N₂O sources applied in the box model**

| Sources name | Description |
|---|---|
| $N_2O_{FF}$ | $N_2O$ emissions from fossil fuel combustion. Sum of all-sector $N_2O$ emissions in CEDS but except for agricultural $N_2O$ |
| $N_2O_{soil}$ | Soil $N_2O$ emissions derived from NMIP2 SH1 ensemble mean |
| $N_2O_{BB}$ | $N_2O$ emissions from biomass burning based on van Marle, et al. [49] |
| $N_2O_{AREC}$ | Anthropogenic $N_2O$ emissions from river, estuaries, and coastal zones. Assumed linear increase from 0.1 Tg N yr$^{-1}$ to 0.5 Tg N yr$^{-1}$ based on Yao, et al. [51] and Canadell.J.G, et al. [52] |
| $N_2O_{chem}$ | $N_2O$ generated by atmospheric chemistry. Set as constant 0.6 Tg N yr$^{-1}$ following Zaehle, et al. [9] |
| $N_2O_{NREC}$ | Natural $N_2O$ emissions from river, estuaries, and coastal zones. Set as constant 0.3 Tg N yr$^{-1}$ following Canadell.J.G, et al. [52] |
| $N_2O_{ocean}$ | $N_2O$ emissions from natural open ocean, which was calculated by the differences between the total $N_2O$ sources required to reach equilibrium at 273.2 ppbv in box model (Eq. 2) and the sum of the rest sources in 1850. Set as 3.3 Tg N yr$^{-1}$ in this study |

All N₂O fluxes represented an annual time series from 1850 to 2019.