## [Peer Review File · Nature]

Manuscript Title: Global net climate effects of anthropogenic reactive nitrogen

Reviewer Comments & Author Rebuttals

Reviewer Reports on the Initial Version:

Referees' comments:

Referee #1 (Remarks to the Author):

Excellent paper. Key results are summarised nicely. However, I would add a bit of context to the paper. I know it isn't exactly reactive N, but I feel that the CO₂ produced from the fabrication of synthetic N fertilizers should be mentioned when discussing the climate effects of reducing Nr. I understand that the focus of the paper is the warming potential of existing Nr, however the authors mention that future reductions in anthropogenic Nr will need to be accompanied with enhanced efforts to reduce anthropogenic GHG emissions from other sources (e.g. see abstract lines 44-46). According to a recent study (Galloway et al., 2021. doi.org/10.1146/annurev-environ-012420-045120), about 2/3rds of anthropogenic Nr is produced via Haber Bosch (and this production accounts for about 1.5% of total anthropogenic GHG emissions). So all of the scenarios shown in Figure 5 (with the exception of the 10% reduction in anthropogenic Nr) would almost certainly include some reduction in Haber Bosch, and therefore a reduction in CO₂ emissions from the industrial production of ammonia. I'm not sure if this needs to be modeled in the analysis, but I strongly suggest that there be at least some mention of these potential reductions in the abstract and/or the conclusion.

The manuscript is original as I am unaware of another paper estimating this on a global scale. It is also highly significant as both the climate and Nr are well beyond what are considered safe planetary boundaries. So, it is critical that we reduce both the emissions of GHG and the amount of Nr within the biosphere, and understanding any trade-offs between climate forcing and Nr concentrations is important.

The modeling ensemble used is appropriate and the quality of the data seems excellent. The presentation of the manuscript is very good, however, the English in the Supplementary Information is sometimes a bit awkward. I have noted some examples below.

I would say that "cultivation of N-fixing crops" is more of an "anthropogenic Nr contribution" and that manure application is "recycling" of existing Nr rather than a new source of Nr.

Please mention what is meant by "NBP". I assume that it means "Net biological productivity" or something like that, but it should be defined somewhere in the manuscript.

Also, how reasonable is it to assume that we can achieve reductions in Nr if we enhance N fertilizer and manure application (see lines 236-237)? According to a recent study (Galloway et al., 2021. doi.org/10.1146/annurev-environ-012420-045120), about 2/3rds of the reactive N is from fertilizer + cultivation-induced biological N fixation. Only 15% of reactive N is from fossil fuels, so even reducing fossil fuel use to 0 will have only minor effects on Nr.

The references also appear to be complete, with a lot of good, new literature cited.

There are also a few small "editorial" type suggestions below:

Line 165: Delete "the"

Figure 5: it seems like the Figure caption is cut off in my version of the draft manuscript.

Line 501: ocean-borne, not bonre.

Extended data figure 1: I'm not sure why, but the lines in my pdf version of the draft are yellow and grey, not red and blue as mentioned in the caption. Also, in ED Fig.1b, "terrestrial" is spelled incorrectly.

ED Figure 2a needs a measure of area. I assume this is kg C m⁻² yr⁻¹.

ED Figure 4: again, my pdf has different colours (grey and yellowish) rather than blue and red.

ED Figure 6: can you fix the y-axes? They are difficult to read.

Supplementary Information.

Some of the writing in the Supplementary Information is a bit awkward and needs revision.

SI line 99 "fluxes" not "flues"

SI line 107 to 111: These two sentences need to be revised, it is very difficult to understand exactly what the authors intend here.

SI line 114: any ideas why the CEDS inventory was so much higher than the EDGAR inventory?

SI line 130 "source" not "sources".

SI line 161: "technical" rather than "technique"?

SI line 176: "parallel"?

SI line 179: "negligible" rather than "neglectable"

Referee #1 (Remarks on code availability):

na

Referee #2 (Remarks to the Author):

Human activities, in particular the exponential increase in the use of mineral fertilizers after World War II, increased emissions associated with energy production and the promotion of legumes in crop rotations, have led to an unprecedented perturbation of the global nitrogen cycle by a factor of about three. The increased release of reactive N (Nr) compounds to the environment is a common thread running through the global environment, as it affects atmospheric N₂O concentrations and aerosol loading, as well as atmospheric chemistry (e.g., O₃ and CH₄ concentrations), biodiversity, or ecosystem and human health.

In particular, Gong et al. assessed how changes in anthropogenic N₂O production and availability affect the climate system. The study partly builds on earlier regional studies for Europe, North America and China and a global study, but goes significantly further in terms of the underlying methodological complexity (model ensembles to estimate global anthropogenic Nr effects on terrestrial ecosystems, use of a detailed global atmospheric chemical transport model to assess the effects of changes in NO_x and NH₃ emission deposition scenarios on short-lived atmospheric compounds such as O₃ and aerosols, and for improved assessment of atmospheric chemistry effects). Overall, the study concludes that the current production and use of Nr has resulted in a net cooling effect, mainly driven by three factors: increased C sequestration by the terrestrial biosphere, reduced radiation input to the Earth's surface due to high tropospheric aerosol loading, and shortened atmospheric lifetime of atmospheric CH₄ as a result of increased O₃ concentrations.

These cooling effects were only partially offset by increased atmospheric concentrations of the greenhouse gases N₂O and O₃. While this mechanism and the conclusion (including estimates of the potential magnitude of the cooling effect of anthropogenic N₂O use) have been reported previously, the study successfully goes beyond previous assessments by using improved global databases on N₂O use and sources, and by coupling ensemble outputs of biosphere models directly to the atmospheric chemistry transport model and a radiative transfer module, which allows better consideration of the spatial heterogeneity of effects and feedbacks. In addition, scenarios are explored on how changes in NR use and NR emissions/deposits may feed back to the future climate. In general, the study is well written and structured, which, given the complexity of the processes and mechanisms involved, makes it easy to follow the different steps applied in this study. Nevertheless, the study also requires some understanding of the previous work, in particular the work done in the context of the NMIP2 model comparison, as the results have been used for the estimation of Nr effects on terrestrial C and N₂O fluxes, including their uncertainties. A more comprehensive explanation of NMIP2 in the SI, as well as its underlying assumptions and uncertainties, may be helpful here. The NMIP2 results were further used to drive the atmospheric chemistry model and to estimate changes in radiative forcing (Figure 3), but I failed to understand, for example, how this ultimately led to the shown global patterns of changes in radiative forcing of e.g. N₂O (not necessarily related to source regions) or CO₂ (seems to be rather uniform). I assume it has to do with global atmospheric circulation patterns, but this is not explained. A critical point in the scenario studies is the assumption that persistent Nr additions continue over extended time scales to support increased C sequestration by terrestrial ecosystems (line 224). In contrast to this statement, and based on N addition experiments in forest ecosystems and stoichiometric considerations, Du and de Vries (2023) question whether the transient increase in C fluxes induced by N deposition can actually be transformed into long-term C storage by ecosystems, as assumed in this study. The assumption of a continuous stimulating effect of N deposition on C sequestration in the biosphere may also affect the conclusion that possible future reductions in anthropogenic Nr could lead to a strong warming effect, although this seems to be mainly related to reduced NO_x emissions (Figure 5). Some additional lines to explain this important conclusion, and also to mention that reducing NO_x and NH₃ emissions and the associated atmospheric aerosol load would almost immediately lead to additional global warming (Fig. 5), would be helpful.

Additional clarification is also needed on some of the underlying assumptions (lines 566 following and Table 4):

- a) Linearity of changes in atmospheric concentrations of aerosols to precursors. Citation?
- b) Linearity of soil N gas fluxes to N inputs (contrary to Shcherbak et al, 2014 and others)
- c) Constant N₂O emissions from water surfaces and oceans (Table 4), although the loading of water sources with Nr has increased significantly over time.

Du E., de Vries W. Impacts of nitrogen deposition on forest productivity and carbon sequestration (2023) Atmospheric Nitrogen Deposition to Global Forests: Spatial Variation, Impacts, and Management Implications, pp. 59 – 76. DOI: 10.1016/B978-0-323-91140-5.00016-6

I. Shcherbak, N. Millar, G.P. Robertson Global metaanalysis of the nonlinear response of soil nitrous oxide (N₂O) emissions to fertilizer nitrogen. Proc. Natl. Acad. Sci. U. S. A., 111 (2014), pp. 9199-9204

Referee #3 (Remarks to the Author):

The goal of this study is to quantify the global climate forcing of anthropogenic reactive nitrogen. The authors use a combination of 8 terrestrial biospheric models and a chemical transport model to estimate the radiative forcing from each constituent perturbed by anthropogenic nitrogen, including CO₂, N₂O, CH₄, O₃, and aerosols. The topic is compelling, and the authors are to be commended for bringing together the latest generation of modelling tools to address this important question. The main manuscript is generally well written and I appreciated the discussion of some of the indirect effects not included in the study. However, the methods used to obtain the key results of this study were quite complex and opaque, as described in the methods and SI section, and the limitations of these methods are not fully discussed. I also had some questions about the impact of non-linearities in the system. I have detailed below my concerns, questions, and suggestions. While I find the study appropriate in topic and scope for Nature, without a considerable overhaul of the methods section, including some additional acknowledgement of the limitations of these methods in the main text, I do not find the article publishable in its current form.

Major

1. While the methods are not discussed at length in the main text, the current text suggests that the results are based purely on the terrestrial models and CTM. However, box modelling (with assumptions) and an offline radiative transfer model are central to this analysis. This should be included where the method is mentioned (Abstract, line 38-39; lines 85-92; lines 147-148).
 - a. Line 148: CO₂ and N₂O fluxes are not added to GEOS-Chem, but rather a box model is used to translate fluxes to concentrations and an RT scaling applied.
 - b. Line 150: replace “considered” with “calculated offline using a box model”
2. While the study is reasonably comprehensive, as acknowledged in the text it does not include indirect aerosol forcing (or some other feedbacks). The phrase “net radiative forcing” is therefore a bit misleading (line 39). I suggest using “net direct radiative forcing”
3. Figure 2 and text from lines 116-136 need to be harmonized and updated: The text refers to changes over 2007-2016 (line 116) or averages over 2007-2016 (line 117, 120, 121, 124, 126) and refers the reader to Figure 2 where these time intervals are not shown! The bars in Figure 2 are far too small to clearly see the magnitude and breakdown. In most cases the y-axes should be reduced such that the bars take up the entire vertical extent (e.g. NBP axes should go from -2 to +4). The paper does not discuss the specific trend over time, so perhaps this figure should compare only pre-industrial and present-day?
4. The study did not address whether the perturbations are linearly additive. Saturation and non-linear chemistry would suggest that they are not, and the authors should quantify and discuss this in the main text.
 - a. Are the contributions (e.g. fertilizer use, manure application, etc.) estimated using the NMIP2 11 transient, factorial simulations additive? Or are there any important saturation effects?
 - b. Do the simulations removing agricultural and non-agricultural sources give the same as the net simulation when both are simultaneously removed? For example, the sum of the RF (-0.36 W/m²; lines 187-188) does not equal the net RF (-0.34 W/m²; line 156) – which species contribute to this? And why? How does this hamper the analysis?
 - c. Ammonium nitrate formation is non-linear in emissions of NO_x and NH₃ – i.e. both are needed and one is limiting, depending on the environment. Presumably cutting emissions of agricultural

sources drastically diminishes emissions of NH₃, producing little ammonium nitrate. Similarly, cutting non-agricultural emissions would eliminate emissions of NO_x (fossil fuel), also leading to little ammonium nitrate. Thus, adding the impact of these two simulations should underestimate the impact when all emissions are included. How did the authors deal with this?

d. The non-linear O₃ chemistry was also not addressed (Section S2.4 is insufficient). Decreasing NO_x emissions can either decrease or increase ozone production, depending on the local chemical regime. Are these non-linearities important (locally or globally) when isolating source effects or when reducing anthropogenic Nr in the simulations? (impossible to see the trend in O₃ on the bars in Figure 5).

e. What is the implication of neglecting any of these non-linearities on the uncertainty analysis in the SI?

5. Section on future Nr reductions is not particularly informative. The simultaneous and persistent Nr reductions are not realistic, nor are the application of the radiative forcing sensitivities, and thus the results should not be interpreted in a policy context. In particular, NO_x and NH₃ emissions are unlikely to follow the same trajectory, so the aerosol response here does not correspond to any meaningful scenario (it would be more interesting to separate the response to NO_x cuts and NH₃ cuts). I recommend trimming this section down to a few sentences. I also recommend cutting Figure 5 which provides little insight (very difficult to see the differences in timescales and identify which components contribute to this; the differences in the 4 scenarios are also not very useful/apparent).

6. The Methods are complex with many steps; the section should be edited for clarity. Perhaps the authors would consider developing a summary figure that shows exactly what parameters come from what methods?

a. Extended Table 2: I suggest adding another table that identifies which pairs of simulations were used to estimate each N factor. This can be used to help explain equation 1.

b. The use of the equation 1 and the term “factorial configuration” makes this a bit unnecessarily complicated to read. Effectively a series of sensitivity experiments are differenced from the baseline simulation to quantify each factor. It would be helpful to describe this with an example, rather than using equations with non-intuitive indices. I think this is true for all the equations used in the methods. They are difficult to parse and I would therefore suggest that the methods be explained in words and if the authors insist on the equations that these be placed and described in the supplement.

c. Extended Table 1 is VERY difficult to read with the equations (same point as above). I suggest you simplify with words, but also add the global NO_x and NH₃ emissions for each line so that they can be compared with Table S1

d. Line 528: is there a reference for this approach for calculating the methane lifetime to OH? If not, can you show that this equation is a good approximation?

e. Lines 525, 531-532: How uncertain are the assumed and calculated CH₄ lifetimes? What about the sensitivity factors (also: need to define a sensitivity factor in the text)? How does this impact the results?

f. The uncertainties on NO_x and NH₃ emissions are independent. Therefore, these emissions should be scaled separately in the GEOS-Chem sensitivity experiments for the uncertainty analysis.

g. Uncertainty analysis in S2 – please refer to the figures and numbers where relevant. It is unclear how values discussed here relate to Figure 1 and Figure S1.

h. How do values in Table S1 relate to Figure S1? For example, the minimum NH₃ emissions in present-day in Figure S1 appear to be ~10 Tg/yr not 0.0 as in Aerosol_max in Table S1. (maximum in

Figure S1 ~45 not 27 as in Table S1). Similar discrepancies for NO_x.

Minor

1. Figure 2 and text that follows: NBP is never defined in the text.
2. Line 66: replace “The” with “An”
3. Lines 106, 192, 235: fossil fuel combustion is singular, not plural
4. Lines 107-108: air pollution impacts on vegetation should be included in this list
5. Line 134: how similar is the 45 Tg/yr from CEDS to the NMIP2 ensemble mean? It looks similar from Fig 2d so perhaps worth stating in text? If Figure 2d is already re-scaled to CEDS, this should be stated in the caption.
6. Figure 4: The magnitude of the bars is difficult to assess. I suggest either expanding the x-dimension of the figure, or including the values for each bar on the figure itself.
7. Line 254: strange that this is the first mention of fires. Should the fire emissions (even if small, and generally constant historically, as suggested in Extended Data) not be included in Figure 2? This should be addressed when discussing Figure 2.
8. Lines 277-278: It would be useful to explain “a win-win” as many may not be familiar with this expression.
9. Line 423: Is the Nr deposition from CCM1 which is used as an input to the terrestrial models consistent with the Nr deposition simulated for the same year with the GEOS-Chem CTM?
10. Line 451: missing brackets around (NGSH_{1,j,yr} – NGSH_{l,j,yr})
11. Lines 464-465: given this, it seems the authors should emphasize the uncertainty in emissions of NH₃ in their conclusions
12. Line 507: missing brackets on d[N₂O]/dt to be consistent with d[CH₄]/dt in line 518

Author Rebuttals to Initial Comments:

Response to referee #1

Title: Global net climate effects of anthropogenic reactive nitrogen

Nature reference number: 2023-10-19078B

Authors: Cheng Gong, Hanqin Tian, Hong Liao, Naiqing Pan, Shufen Pan, Akihiko Ito, Atul K. Jain, Sian Kou-Giesbrecht, Fortunat Joos, Qing Sun, Hao Shi, Nicolas Vuichard, Qing Zhu, Changhui Peng, Federico Maggi, Fiona H.M. Tang and Sönke Zaehle

Excellent paper. Key results are summarised nicely. However, I would add a bit of context to the paper. I know it isn't exactly reactive N, but I feel that the CO₂ produced from the fabrication of synthetic N fertilizers should be mentioned when discussing the climate effects of reducing Nr. I understand that the focus of the paper is the warming potential of existing Nr, however the authors mention that future reductions in anthropogenic Nr will need to be accompanied with enhanced efforts to reduce anthropogenic GHG emissions from other sources (e.g. see abstract lines 44-46). According to a recent study (Galloway et al., 2021. doi.org/10.1146/annurev-environ-012420-045120), about 2/3rds of anthropogenic Nr is produced via Haber Bosch (and this production accounts for about 1.5% of total anthropogenic GHG emissions). So all of the scenarios shown in Figure 5 (with the exception of the 10% reduction in anthropogenic Nr) would almost certainly include some reduction in Haber Bosch, and therefore a reduction in CO₂ emissions from the industrial production of ammonia. I'm not sure if this needs to be modeled in the analysis, but I strongly suggest that there be at least some mention of these potential reductions in the abstract and/or the conclusion.

Response:

Thank you for the encouraging comments. We agree with the reviewer that CO₂ from Haber Bosch is not exactly Nr, and should be attributed to 'fossil fuel CO₂ emissions'. Nevertheless, it is strongly correlated to anthropogenic Nr production and therefore to some extent relevant to the topic of the manuscript.

The magnitude of CO₂ emissions from the Haber-Bosch process is much lower than the C sequestration fertilized by N addition in terrestrial ecosystem (based on the NBP of the NMIP2 ensemble used in this study). Based on the oversimplifying and likely high-biased assumption that all fertilizer (4.021 Pg N cumulated from 1850 to 2019 according to NMIP2 input data) has been produced by Haber-Bosch and an emission factor of 1.87 ton CO₂ / 1 ton NH₃ produced¹, Haber Bosch would have been responsible for 2.6 Pg C over 1850 -2019, which is approximately 6.6% of the anthropogenic Nr effect on NBP (36.4 Pg C based on NMIP2 ensemble). In consequence, having ignored this effect does not significantly influence our estimates on the global climate effects of anthropogenic Nr.

In the proposed revision of the manuscript (see below) we suggest to replace our simple N reduction scenarios with the more realistic CMIP6 SSP scenarios (See our responses to Reviewer #3). Accounting for the CO₂ emissions from Haber Bosch will not change our conclusion, but may even slightly enhancing the future net warming effects relative to 2019. In SSP 3-7.0 and SSP 5-8.5, the higher Haber Bosch N fixation induces higher CO₂ emissions, and thus compensate the cooling effects led by N fertilization effects on NBP. However, due to its relatively low magnitude, we do not expect this would have significant influence on the results.

To appropriately account for the comment, we have added the following discussion (addition in **bold**) in the main text:

‘In this study, several processes, including the influence of aerosols or O₃ on terrestrial carbon fluxes, aerosol-cloud interactions, N addition effects on soil CH₄ uptake and N fertilization on marine biogeochemistry were not included due to the likely small effect on climate or uncertainty to quantify the global effect (See details in SI text S2.5). **It should be noted that the future CO₂ cooling due to CO₂ uptake on land may be overestimated in our study because we omit the contribution of fossil-fuel based CO₂ emissions from N fertilizer production by the Haber Bosch,**’

Also added the detailed explanations in SI text S2.1:

‘The estimates presented in this study do not account for the CO₂ emissions due to artificial N fixation as part of the Haber Bosch process, because they are not directly related to the anthropogenic Nr effects, and are implicitly accounted as part of fossil-fuel based carbon emissions. Nevertheless, as a first estimate with an emission factor of 1.87 ton CO₂ per 1 ton NH₃ produced¹, the cumulated fossil fuel CO₂ emissions (C_{emiss,HB}) from Haber Bosch are about 2.6 Pg C with ca. 4.021 Pg N cumulative fertilizer production over 1850-2019 (C_{emiss,HB} = 4.021 * (17.0 / 14.0) * 1.87 * (12.0 / 44.0)). In comparison, this is 6.6% of the NMIP2 ensemble estimate of 36.4 Pg C cumulated NBP due to anthropogenic Nr. However, the constant or even slightly increasing fertilizer production embedded in the future SSP scenarios will result in constant or higher CO₂ emissions from Nr generation, and thus slightly weaken the cooling effects led by N fertilization effects on terrestrial carbon sinks.’

The manuscript is original as I am unaware of another paper estimating this on a global scale. It is also highly significant as both the climate and Nr are well beyond what are considered safe planetary boundaries. So, it is critical that we reduce both the emissions of GHG and the amount of Nr within the biosphere, and understanding any trade-offs between climate forcing and Nr concentrations is important.

Response:

Thank you for acknowledging the significance of this work.

The modeling ensemble used is appropriate and the quality of the data seems excellent. The presentation of the manuscript is very good, however, the English in the Supplementary Information is sometimes a bit awkward. I have noted some examples below.

Response:

We have improved the languages in SI thoroughly and have reworded the text where it was unclear.

I would say that “cultivation of N-fixing crops” is more of an “anthropogenic Nr contribution” and that manure application is “recycling” of existing Nr rather than a new source of Nr.

Response:

In line with our main text discussion, this study did not examine the impact resulting from land use changes, such as the effects of cultivation of N-fixing crops, because various indirect biogeochemical effects (e.g. on C cycles) will be together introduced and it is difficult to isolate the ‘pure’ Nr effect.

Here, Nr sources are identified by the categories of different anthropogenic activities, rather than the form of N transforming. It could be very interesting to further examine all anthropogenic Nr generated from N₂, but it is challenging with the current data and far beyond the scope of this study. As we clarified in the beginning of the results, we aim to assess the ‘direct anthropogenic Nr inputs to the Earth system’, where the increasing manure inputs due to enhanced anthropogenic stockbreeding should be included.

Please mention what is meant by “NBP”. I assume that it means “Net biological productivity” or something like that, but it should be defined somewhere in the manuscript.

Response:

In the revised manuscript we now clarify that NBP is the ‘net biome productivity’ and corresponds the carbon balance of the terrestrial biosphere.

Also, how reasonable is it to assume that we can achieve reductions in Nr if we enhance N fertilizer and manure application (see lines 236-237)? According to a recent study (Galloway et al., 2021. doi.org/10.1146/annurev-environ-012420-045120), about 2/3rds of the reactive N is from fertilizer + cultivation-induced biological N fixation. Only 15% of reactive N is from fossil fuels, so even reducing fossil fuel use to 0 will have only minor effects on Nr.

Response:

Following the comments from Reviewer #3, in the revised manuscript we now make use of the SSP scenarios from the CMIP6 project to provide a consistent and more realistic estimate the future climate effects of Nr. In these scenarios, the future fertilizer and manure applications remain similar or even increase in magnitude relative to present day, while fossil-fuel related Nr emissions decrease to varying degrees in the scenarios. Please see our detailed responses to Reviewer #3.

Fossil fuel is not the dominant source of Nr, but considering the climate effects of fossil-fuel derived NO_x on atmospheric methane, aerosols and ozone, the reduction in fossil-fuel based Nr will lead to significant impacts on climate, as demonstrated in the revised Figure 5.

The references also appear to be complete, with a lot of good, new literature cited.

Response:

Thank you.

*There are also a few small “editorial” type suggestions below:
Line 165: Delete “the”*

Response:

Revised.

Figure 5: it seems like the Figure caption is cut off in my version of the draft manuscript.

Response:

Not sure what happened, but we suppose it should be fine in our revised manuscript.

Line 501: ocean-borne, not bonre.

Response:

Revised.

Extended data figure 1: I'm not sure why, but the lines in my pdf version of the draft are yellow and grey, not red and blue as mentioned in the caption. Also, in ED Fig.1b, "terrestrial" is spelled incorrectly.

Response:

We are very sorry for the inconsistency. We have revised the figure caption as 'yellow and grey'. The spelling error is also fixed.

ED Figure 2a needs a measure of area. I assume this is kg C m⁻² yr⁻¹.

Response:

Revised.

ED Figure 4: again, my pdf has different colours (grey and yellowish) rather than blue and red.

Response:

Sorry for the mistake. It has been revised.

ED Figure 6: can you fix the y-axes? They are difficult to read.

Response:

Revised.

Supplementary Information.

Some of the writing in the Supplementary Information is a bit awkward and needs revision.

SI line 99 "fluxes" not "flues"

Response:

Revised.

SI line 107 to 111: These two sentences need to be revised, it is very difficult to understand exactly what the authors intend here.

Response:

Sorry for the confusion. We have revised the whole paragraph to address uncertainties led by N saturation effects on terrestrial carbon sinks (Please see detailed discussions in the response to the Reviewer #2).

‘The extrapolation of the anthropogenic Nr effect into the next three decades relies on the cumulative response of the C cycle to Nr inputs over 1850-2020. This approach does not consider the potential for future N saturation in terrestrial ecosystems, and therefore potentially leads to an overestimate of the future terrestrial carbon uptake and subsequent cooling effects. Ecosystems with high anthropogenic Nr inputs (e.g. croplands; forest and grassland in dense-population regions) may already be saturated with Nr at present, as shown by previous data-based studies²⁻⁴ and the results of the NMIP2 ensemble (Figs. S3 and S4). To estimate the magnitude of the potential N saturation on the future climate forcing from anthropogenic Nr, we repeated the SSP-scenario experiments, but assumed that future fertilizer and manure application would not further enhance NBP. This modification, shown in Figure S5, does not change the patterns reported in Fig. 5, but exhibits a slight tendency towards stronger warming effects (0.02-0.03 W m⁻² increases by 2050s, Table S4) when considering the N saturation effect. This demonstrates that despite uncertainties in the extend of N saturation, the key findings of our studies remain robust.’

Table S4. The accumulated NBP fluxes over 2020 to 2050 as well as the corresponding predicted radiative forcing relative to 1850 under two sensitivity future experiments under three SSP scenarios. The present-day values are also given as a reference.

		Accumulated NBP over 2020- 2050 (Pg C)	RF (CO ₂) induced by anthropogenic Nr relative to 1850 (W m ⁻²)
Present day (2019)		36.40 (1850-2019)	-0.121
SSP 1-2.6	With fertilizer and manure effects on NBP	15.20	-0.156
	Without fertilizer and manure effects on NBP	6.90	-0.136
SSP 3-7.0	With fertilizer and manure effects on NBP	18.62	-0.164
	Without fertilizer and manure effects on NBP	8.18	-0.139
SSP 5-8.5	With fertilizer and manure effects on NBP	16.92	-0.160
	Without fertilizer and manure effects on NBP	8.17	-0.139

SI line 114: any ideas why the CEDS inventory was so much higher than the EDGAR inventory?

Response:

The reason is not clear to us. However, such uncertainties led by different anthropogenic inventories should be addressed and make readers aware.

SI line 130 “source” not “sources”.

Response:

Revised.

SI line 161: “technical” rather than “technique”?

Response:

Revised.

SI line 176: “parallel”?

Response:

Revised.

SI line 179: “negligible” rather than “neglectable”

Response:

Revised.

References:

- 1 Wang, Y. & Meyer, T. J. A Route to Renewable Energy Triggered by the Haber-Bosch Process. *Chem* **5**, 496-497, doi:10.1016/j.chempr.2019.02.021 (2019).
- 2 Peng, Y. F., Chen, H. Y. H. & Yang, Y. H. Global pattern and drivers of nitrogen saturation threshold of grassland productivity. *Functional Ecology* **34**, 1979-1990, doi:10.1111/1365-2435.13622 (2020).
- 3 He, N. P. *et al.* Global patterns of nitrogen saturation in forests. *Pre-print*, doi:<https://doi.org/10.21203/rs.3.rs-3559857/v1> (2023).
- 4 Schulte-Uebbing, L. F., Beusen, A. H. W., Bouwman, A. F. & de Vries, W. From planetary to regional boundaries for agricultural nitrogen pollution. *Nature* **610**, 507-+, doi:10.1038/s41586-022-05158-2 (2022).

Response to referee #2

Title: Global net climate effects of anthropogenic reactive nitrogen

Nature reference number: 2023-10-19078B

Authors: Cheng Gong, Hanqin Tian, Hong Liao, Naiqing Pan, Shufen Pan, Akihiko Ito, Atul K. Jain, Sian Kou-Giesbrecht, Fortunat Joos, Qing Sun, Hao Shi, Nicolas Vuichard, Qing Zhu, Changhui Peng, Federico Maggi, Fiona H.M. Tang and Sönke Zaehle

Human activities, in particular the exponential increase in the use of mineral fertilizers after World War II, increased emissions associated with energy production and the promotion of legumes in crop rotations, have led to an unprecedented perturbation of the global nitrogen cycle by a factor of about three. The increased release of reactive N (Nr) compounds to the environment is a common thread running through the global environment, as it affects atmospheric N₂O concentrations and aerosol loading, as well as atmospheric chemistry (e.g., O₃ and CH₄ concentrations), biodiversity, or ecosystem and human health.

In particular, Gong et al. assessed how changes in anthropogenic N₂O production and availability affect the climate system. The study partly builds on earlier regional studies for Europe, North America and China and a global study, but goes significantly further in terms of the underlying methodological complexity (model ensembles to estimate global anthropogenic Nr effects on terrestrial ecosystems, use of a detailed global atmospheric chemical transport model to assess the effects of changes in NO_x and NH₃ emission deposition scenarios on short-lived atmospheric compounds such as O₃ and aerosols, and for improved assessment of atmospheric chemistry effects

Overall, the study concludes that the current production and use of Nr has resulted in a net cooling effect, mainly driven by three factors: increased C sequestration by the terrestrial biosphere, reduced radiation input to the Earth's surface due to high tropospheric aerosol loading, and shortened atmospheric lifetime of atmospheric CH₄ as a result of increased O₃ concentrations. These cooling effects were only partially offset by increased atmospheric concentrations of the greenhouse gases N₂O and O₃. While this mechanism and the conclusion (including estimates of the potential magnitude of the cooling effect of anthropogenic N₂O use) have been reported previously, the study successfully goes beyond previous assessments by using improved global databases on N₂O use and sources, and by coupling ensemble outputs of biosphere models directly to the atmospheric chemistry transport model and a radiative transfer module, which allows better consideration of the spatial heterogeneity of effects and feedbacks. In addition, scenarios are explored on how changes in NR use and NR emissions/deposits may feed back to the future climate.

Response:

Thank you for the comprehensive and thoroughly summary for the main results of this study. We appreciate your acknowledgment of the significance of our work.

For the comments below, we noticed that many aspects were concentrated in one paragraph. To make our responses clearer, we have splitted them into several points.

In general, the study is well written and structured, which, given the complexity of the processes and mechanisms involved, makes it easy to follow the different steps applied in this study. Nevertheless,

the study also requires some understanding of the previous work, in particular the work done in the context of the NMIP2 model comparison, as the results have been used for the estimation of Nr effects on terrestrial C and N₂O fluxes, including their uncertainties. A more comprehensive explanation of NMIP2 in the SI, as well as its underlying assumptions and uncertainties, may be helpful here.

Response:

NMIP2 was launched based on the previous NMIP project⁵ with improved input data, increased sensitive experiments and model members. NMIP2 was described as part of the global N₂O budget assessment⁶. The uncertainties in the NMIP2 ensemble can be summarized as differences in model configuration, process parameterization and scope of processes representations, with more details and assessment provided by Tian, et al. ⁷. We have added the Tian, et al. ⁷ to the Methods section to provide the reader with a reference to relevant context of the NMIP ensemble. We also added a new Section in the SI (S1.1) to summarize the uncertainties in NMIP2 model ensembles:

‘S1.1 Uncertainties in the NMIP2 model ensemble

The NMIP2 ensemble consists of eight terrestrial biosphere models with fully-coupled C and N cycles. While the models represent a comprehensive budget of the terrestrial N cycle and generally simulate global fluxes within the range of published estimates, the models are based on diverging representations of key processes, including biological N fixation, N mineralization, nitrification and denitrification, which contributes to inter-model variability as revealed by the substantial standard deviation of the mean. The models represent the individual sensitivities of simulated N₂O to elevated CO₂, warming and changes in wetness broadly in agreement with observations, but given the lack of suitable observations, the interactions among different environmental forcing remains insufficiently evaluated. Tian, et al. ⁶ highlight the representation of the human nitrogen management practices in agriculture and the N effects of seasonal freezing-thaw in permafrost as key weaknesses of these models. However, even with these limitations, terrestrial biosphere models are still the most straightforward and powerful tools to isolate different sectors of anthropogenic Nr and understand their substantial influences on the Nr-related gas fluxes.’

The NMIP2 results were further used to drive the atmospheric chemistry model and to estimate changes in radiative forcing (Figure 3), but I failed to understand, for example, how this ultimately led to the shown global patterns of changes in radiative forcing of e.g. N₂O (not necessarily related to source regions) or CO₂ (seems to be rather uniform). I assume it has to do with global atmospheric circulation patterns, but this is not explained.

Response:

CO₂, N₂O and CH₄ are well-mixed gases within the atmosphere, as they have much longer lifetimes than the average interhemispheric mixing time of the troposphere. Even though slight interhemispheric gradients are known to exist for CO₂ and CH₄, we have assumed here that the Nr effects at the decadal time-scale considered here is uniformly distributed in the global atmosphere. We thus assumed that the estimated changes in the mean global atmospheric CO₂, N₂O and CH₄ mole fractions derived from the box models can be directly applied as uniformly distributed globally into the RRTMG model in GEOS-Chem.

The global patterns of radiative forcing attributed to CO₂, CH₄ and N₂O changes are not induced by the spatial arrangement of emission changes, but correspond to the interaction of the individual GHG's radiative forcing with other gases and, in particular, the global climatological cloud patterns. Figure R1 shows the MERRA2 annual-mean global pattern of total cloud fractions in 2019, which shares the similar pattern as Figs. 3a-3c. The radiative forcing of long-lived greenhouse gases will be slightly higher with low cloud fraction given the overlap with the radiative forcing from water vapour.

Figure R1. The global pattern of total cloud fraction averaged over 2019 derived from MERRA2 reanalyzed dataset.

To make the description accurate, we added the corresponding description to explain the global patterns of the radiative forcing in the figure caption:

Figure 3. Global direct radiative forcing induced by anthropogenic Nr derived by GEOS-Chem-RRTMG model for (a) CO₂, (b) N₂O, (c) CH₄, (d) Aerosols, (e) O₃, and (f) the net effect (i.e. sum of a-e). Results were derived by differences in all-sky top-of-atmosphere radiative forcing between CTRL_2019 and No_allNr experiments (see Methods and Extended Data Table 1). The radiative forcing of aerosols is the sum of the direct radiative forcing contributed by ammonium, nitrate and sulphate aerosols. Numbers in parentheses represent the global area-weighted averages, while numbers in the brackets indicate the uncertainty ranges based on sensitivity experiments with GEOS-Chem-RRTMG using \pm one standard deviation among NMIP2 ensembles as well as $\pm 30\%$ uncertainty in OH and O₃ concentrations (SI text S1.2). **Note the Nr effects on global CO₂, N₂O and CH₄ are assumed to be evenly distributed, so that the patterns of these three greenhouse gases are mostly determined by other forcing agents, including the distribution of clouds.**

A critical point in the scenario studies is the assumption that persistent Nr additions continue over extended time scales to support increased C sequestration by terrestrial ecosystems (line 224). In contrast to this statement, and based on N addition experiments in forest ecosystems and stoichiometric considerations, Du and de Vries (2023) question whether the transient increase in C fluxes induced by N deposition can actually be transformed into long-term C storage by ecosystems, as assumed in this study.

Response:

The reviewer raises an important point, which we already considered in the previous version of the manuscript, since the terrestrial biosphere models generally have the capability to simulate the N saturation effects on C sequestration. Therefore, non-linear effects between N addition and NBP have been already considered in the historical assessment within the limits of the realism of processes included in the models. Nevertheless, as we detail in the following paragraphs, our assumption that future Nr addition will continuously increase C sequestration can be considered optimistic, especially considering that many studies^{2-4,8} have reported the N saturation in population-dense regions. We find, as we explain below, that this does not change our conclusions about the future trends in climate forcing from anthropogenic Nr.

A robust, data-driven understanding of regions of N saturation is still lacking, scientifically very challenging and therefore beyond the scope of this work. Nevertheless, we explore the potential impact on our analysis by discussing 1) the N-saturated thresholds provided by Du and de Vries (2023); and 2) the performance of NMIP2 models. Finally, we test the sensitivity of our results for the future scenario analysis to the N saturation effect.

1) From a data-based perspective, Du and de Vries (2023) assessed the N-saturated thresholds of N deposition as around 5-10 kg ha⁻¹ yr⁻¹ for boreal forest and 5-46 kg ha⁻¹ yr⁻¹ for temperate forest, while the data were insufficient to derive N saturation thresholds for tropical forests and grasslands. Using these thresholds together with current rates of N deposition (values for 2019, as used to drive the NMIP2 ensemble) (Fig. R2a) and the present-day PFT distribution (Fig. R3), Figure R2b shows the N deposition with exceedance of the N-saturated thresholds, where the N saturates in regions with high Nr loadings and dense population, especially in Southern and Eastern Asia. Such pattern is similar with a recent pre-print data-driven study (under review at one Nature portfolio journal, <https://doi.org/10.21203/rs.3.rs-3559857/v1>).

Figure R2. (a) The global pattern of N deposition in 2019 from IGAC/CCMI, which is used to drive NMIP2 models. (b) N deposition that exceeding the N-saturated thresholds. We adopt the thresholds of $8 \text{ kg ha}^{-1} \text{ yr}^{-1}$ for boreal forests, and $20 \text{ kg ha}^{-1} \text{ yr}^{-1}$ for temperate and tropical forest, and $5 \text{ kg ha}^{-1} \text{ yr}^{-1}$ grassland and crops (considering already high fertilizer/manure loading).

Figure R3. The global PFT distributions that is used to calculate N exceedance in Fig. R2b

2) We analyze the relationships between N addition and its fertilization effects on global NBP based on the NMIP2 dataset (Figs. R4 and R5). N deposition in recent decades and many global regions have stopped rapid increases and consequently hindered the amplification of N deposition effects on global NBP, the saturation effect of N deposition is not significant across models (see Fig. R4). However, in agreement with the data-based analysis by Du and de Vries (2023), models that show a large effect of N addition on carbon storage in managed ecosystems (e.g. DLEM, ELM, LPX-Bern and ORCHIDEE) show signs of N saturation from high-levels of fertilizer and manure application, which are typically concentrated at the population-dense regions.

Figure R4. The relationships of N fertilization effects of N deposition on NBP, which is derived from the differences among corresponding NMIP2 sensitivity experiments. Each dot indicates the annual value of global N deposition (X axis) and global NBP enhancement (Y axis) for each model member as well as the ensemble mean from 1850 to 2019.

Figure R5. The same as Fig. R4 but for the N fertilization effects of fertilizer and manure applications on NBP.

In conclusion of both 1) and 2), we can reasonably assume that N saturation effects will likely be significant in areas of high loading of fertilizer and manure application, but not (at least at global scales) for the N deposition effect in the next few decades. As the future SSP scenarios used in our revised manuscript for future projects suggest across scenarios with stable or even increasing fertilizer and manure application but decreased N deposition (see also the responses to Reviewer #3), we add a sensitivity test for the future assessment of the N_r climate effects. In this model simulation, we assume that future fertilizer and manure additions will not increase future NBP due to the N saturation effect, while every other forcing varies as identical as the revised Fig. 5. Figure R6 shows that our main results will not be significantly changed with the N saturation effect in cropland and grasslands considered. Conversely, with the N saturation on NBP, our analysis suggests an even stronger warming effect due to future anthropogenic N_r , because in these projections the terrestrial carbon sinks do not increase as expected from the simple linear assumption between anthropogenic N_r and NBP.

Figure R6. Global climate responses to the future SSP scenarios of anthropogenic Nr with exclusion (silver lines) and inclusion (brown lines) of the effects of fertilizer and manure application on NBP. Methods of this future extrapolation are detailed in the response to Review #3.

The assumption of a continuous stimulating effect of N deposition on C sequestration in the biosphere may also affect the conclusion that possible future reductions in anthropogenic Nr could lead to a strong warming effect, although this seems to be mainly related to reduced NOx emissions (Figure 5). Some additional lines to explain this important conclusion, and also to mention that reducing NOx and NH3 emissions and the associated atmospheric aerosol load would almost immediately lead to additional global warming (Fig. 5), would be helpful.

Response:

Thank you for the comment. We have added the following description to the main text:

‘In the SSP 3-7.0 scenario, the future fossil fuel sources of Nr remain close to the 2019 level. Enhanced fertilizer and manure applications increase N_2O emissions and lead to a stronger N_2O warming effect of $+0.06 W m^{-2}$ in 2050s relative to 2019, which is compensated by the cooling effects of increased aerosol loadings ($-0.03 W m^{-2}$ enhancement in 2050s relative to 2019) and enhanced terrestrial carbon sequestration ($-0.04 W m^{-2}$ enhancement in 2050s relative to 2019). However, bounding assumptions on the magnitude of N saturation (See SI text S2.1 and Fig. S4) suggest that carbon sequestration effect might be overestimated by about $0.02 W m^{-2}$...’

We have also revised the discussion of the N saturation effect in the SI text S2.1:

‘The extrapolation of the anthropogenic Nr effect into the next three decades relies on the cumulative response of the C cycle to Nr inputs over 1850-2020. This approach does not consider the potential for future N saturation in terrestrial ecosystems, and therefore potentially leads to an overestimate of the future terrestrial carbon uptake and subsequent cooling effects. Ecosystems with high anthropogenic Nr inputs (e.g. croplands; forest and grassland in dense-population regions) may already be saturated with Nr at present, as shown by previous data-based studies²⁻⁴ and the results of the NMIP2 ensemble (Figs. S3 and S4). To estimate the magnitude of the potential N saturation on the future climate forcing from anthropogenic Nr, we repeated the SSP-scenario experiments, but assumed that future fertilizer and manure application would not further enhance NBP. This modification, shown in Figure S5, does not change the patterns reported in Fig. 5, but exhibits a slight tendency towards stronger warming effects (0.02-0.03 W m⁻² increases by 2050s, Table S4) when considering the N saturation effect. This demonstrates that despite uncertainties in the extend of N saturation, the key findings of our studies remain robust.’

Additional clarification is also needed on some of the underlying assumptions (lines 566 following and Table 4):

a) Linearity of changes in atmospheric concentrations of aerosols to precursors. Citation?

Response:

We added citations as:

‘... while the radiative forcing of short-lived gases or aerosols was linearly related to total precursors’ emissions⁹⁻¹¹ at the corresponding year;’

b) Linearity of soil N gas fluxes to N inputs (contrary to Shcherbak et al, 2014 and others)

Response:

Based on the synthesis of site-level manipulating experiments, Shcherbak et al. (2014) revealed a non-linear relationship between N inputs and soil N₂O emissions, which is depending on varying soil N contents, plant N requirements, plant N use efficiency and local climate condition. Such local non-linear relationships between N inputs and N₂O emissions are involved by the NMIP2 models. However, the NMIP2 ensemble shows that at the global scale, which integrates over ecosystems in various stages of N availability, the relationship between N input and soil N₂O emission is approximately linear within the range of the future Nr inputs (around 260-320 Tg N yr⁻¹) in the SSP scenarios for the next three decades. (Fig. R7).

Figure R7. The same as Fig. R4 but for the soil N₂O emissions contributed by the total anthropogenic Nr inputs, including fertilizer and manure application and N deposition.

c) Constant N₂O emissions from water surfaces and oceans (Table 4), although the loading of water sources with Nr has increased significantly over time.

Response:

Firstly, the water surface N₂O emissions are not constant. For N₂O emissions from natural river, estuaries, and coastal zones, we divide them into anthropogenic and natural contributions following the numbers of IPCC AR6¹². The anthropogenic contribution increased from 0.1 Tg N yr⁻¹ to 0.5 Tg N yr⁻¹ from 1850 to 2019, while the natural contribution was fixed as 0.3 Tg N yr⁻¹. To avoid misleading, we have added ‘Natural N₂O emissions’ on the description of N₂O_{NREC} in Extended Data Table 4.

According to IPCC AR6, the oceanic N₂O emission itself is quite uncertain (2.5-4.3 Tg N yr⁻¹), while the contribution led by anthropogenic N inputs is estimated around 0.1-0.2 Tg N yr⁻¹¹², which is much lower than the uncertainty range. In this case, the anthropogenic Nr effect on oceanic N₂O emissions is neglectable.

Du E., de Vries W. Impacts of nitrogen deposition on forest productivity and carbon sequestration (2023) Atmospheric Nitrogen Deposition to Global Forests: Spatial Variation, Impacts, and Management Implications, pp. 59 – 76. DOI: 10.1016/B978-0-323-91140-5.00016-6

I. Shcherbak, N. Millar, G.P. Robertson Global metaanalysis of the nonlinear response of soil nitrous oxide (N₂O) emissions to fertilizer nitrogen. Proc. Natl. Acad. Sci. U. S. A., 111 (2014), pp. 9199-9204

Reference

- 2 Peng, Y. F., Chen, H. Y. H. & Yang, Y. H. Global pattern and drivers of nitrogen saturation threshold of grassland productivity. *Functional Ecology* **34**, 1979-1990, doi:10.1111/1365-2435.13622 (2020).
- 3 He, N. P. *et al.* Global patterns of nitrogen saturation in forests. *Pre-print*, doi:<https://doi.org/10.21203/rs.3.rs-3559857/v1> (2023).
- 4 Schulte-Uebbing, L. F., Beusen, A. H. W., Bouwman, A. F. & de Vries, W. From planetary to regional boundaries for agricultural nitrogen pollution. *Nature* **610**, 507-+, doi:10.1038/s41586-022-05158-2 (2022).
- 5 Tian, H. Q. *et al.* THE GLOBAL N₂O MODEL INTERCOMPARISON PROJECT. *Bulletin of the American Meteorological Society* **99**, 1231-1252, doi:10.1175/bams-d-17-0212.1 (2018).
- 6 Tian, H. *et al.* Global Nitrous Oxide Budget 1980-2020. *Earth Syst. Sci. Data Discuss.* **2023**, 1-98, doi:10.5194/essd-2023-401 (2023).
- 7 Tian, H. Q. *et al.* Global soil nitrous oxide emissions since the preindustrial era estimated by an ensemble of terrestrial biosphere models: Magnitude, attribution, and uncertainty. *Global Change Biology* **25**, 640-659, doi:10.1111/gcb.14514 (2019).
- 8 Lu, X. H. *et al.* Simulated effects of nitrogen saturation on the global carbon budget using the IBIS model. *Scientific Reports* **6**, doi:10.1038/srep39173 (2016).
- 9 Unger, N., Shindell, D. T., Koch, D. M. & Streets, D. G. Air pollution radiative forcing from specific emissions sectors at 2030. *Journal of Geophysical Research-Atmospheres* **113**, doi:10.1029/2007jd008683 (2008).
- 10 Chen, Y. J. *et al.* Investigating the Linear Dependence of Direct and Indirect Radiative Forcing on Emission of Carbonaceous Aerosols in a Global Climate Model. *Journal of Geophysical Research-Atmospheres* **123**, 1657-1672, doi:10.1002/2017jd027244 (2018).
- 11 Thornhill, G. D. *et al.* Effective radiative forcing from emissions of reactive gases and aerosols - a multi-model comparison. *Atmospheric Chemistry and Physics* **21**, 853-874, doi:10.5194/acp-21-853-2021 (2021).
- 12 Canadell, J. G. *et al.* Global Carbon and other Biogeochemical Cycles and Feedbacks. In *Climate Change 2021: The Physical Science Basis. Contribution of Working Group I to the Sixth Assessment Report of the Intergovernmental Panel on Climate Change*. 673–816 (2021).

Response to referee #3

Title: Global net climate effects of anthropogenic reactive nitrogen

Nature reference number: 2023-10-19078B

Authors: Cheng Gong, Hanqin Tian, Hong Liao, Naiqing Pan, Shufen Pan, Akihiko Ito, Atul K. Jain, Sian Kou-Giesbrecht, Fortunat Joos, Qing Sun, Hao Shi, Nicolas Vuichard, Qing Zhu, Changhui Peng, Federico Maggi, Fiona H.M. Tang and Sönke Zaehle

The goal of this study is to quantify the global climate forcing of anthropogenic reactive nitrogen. The authors use a combination of 8 terrestrial biospheric models and a chemical transport model to estimate the radiative forcing from each constituent perturbed by anthropogenic nitrogen, including CO₂, N₂O, CH₄, O₃, and aerosols. The topic is compelling, and the authors are to be commended for bringing together the latest generation of modelling tools to address this important question. The main manuscript is generally well written and I appreciated the discussion of some of the indirect effects not included in the study. However, the methods used to obtain the key results of this study were quite complex and opaque, as described in the methods and SI section, and the limitations of these methods are not fully discussed. I also had some questions about the impact of non-linearities in the system. I have detailed below my concerns, questions, and suggestions. While I find the study appropriate in topic and scope for Nature, without a considerable overhaul of the methods section, including some additional acknowledgement of the limitations of these methods in the main text, I do not find the article publishable in its current form.

Response:

Thank you for acknowledging the importance of the topic of this work. We appreciate your constructive and helpful comments on further improving methods descriptions. We have improved the methods section and explained the non-linearity impacts in this work accordingly. Please find our point-to-point responses below.

Major

1. While the methods are not discussed at length in the main text, the current text suggests that the results are based purely on the terrestrial models and CTM. However, box modelling (with assumptions) and f are central to this analysis. This should be included where the method is mentioned (Abstract, line 38-39; lines 85-92; lines 147-148).

Response:

Thank you for the comments. In the revised manuscript the description of the Methods has been modified and the box modelling approach has been mentioned accordingly.

a. Line 148: CO₂ and N₂O fluxes are not added to GEOS-Chem, but rather a box model is used to translate fluxes to concentrations and an RT scaling applied.

Response:

Revised.

b. Line 150: replace “considered” with “calculated offline using a box model”

Response:

Revised.

2. While the study is reasonably comprehensive, as acknowledged in the text it does not include indirect aerosol forcing (or some other feedbacks). The phrase “net radiative forcing” is therefore a bit misleading (line 39). I suggest using “net direct radiative forcing”

Response:

Thank you for the correction. We have replaced all ‘net radiative forcing’ as ‘net direct radiative forcing’ throughout the manuscript.

3. Figure 2 and text from lines 116-136 need to be harmonized and updated: The text refers to changes over 2007-2016 (line 116) or averages over 2007-2016 (line 117, 120, 121, 124, 126) and refers the reader to Figure 2 where these time intervals are not shown! The bars in Figure 2 are far too small to clearly see the magnitude and breakdown. In most cases the y-axes should be reduced such that the bars take up the entire vertical extent (e.g. NBP axes should go from -2 to +4). The paper does not discuss the specific trend over time, so perhaps this figure should compare only pre-industrial and present-day?

Response:

The reason we use the averages over 2007-2016 is to make the numbers comparable to the IPCC report, which leads to the the inconsistency with the Fig. 2. To make it clear, we have updated the values in the main text as the averages over 2016-2020, which is consistent with the final bars in Fig. 2.

The Figure 2 has been re-plotted:

While we do not specifically focus decadal differences between 1850 and present-day, the development is still important to show the historical trend and thus help readers to 1) better understand the future potential trends of each individual Nr components and 2) to build the box models of CO₂, N₂O and CH₄ and. We therefore prefer to keep the time-series character of the Figure 2.

4. The study did not address whether the perturbations are linearly additive. Saturation and non-linear chemistry would suggest that they are not, and the authors should quantify and discuss this in the main text.

Response:

Complex systems dynamics and to some extent non-linear chemistry imply that is impossible to achieve a strict, 100% linearly additive form regarding the anthropogenic Nr effects at local levels. However, with a global perspective across the interannual to decadal timescale, most of the effects are linear enough to allow for a reasonably accurate attribution. Below we provide evidence for why the nonlinear effects (e.g. saturation and chemistry) exist but will not significantly influence our main results. We also add more details and discussions where the non-linearity arises up, in particular the chemistry of nitrate aerosols. Please see details in the point-to-point response below.

a. Are the contributions (e.g. fertilizer use, manure application, etc.) estimated using the NMIP2 11 transient, factorial simulations additive? Or are there any important saturation effects?

Response:

For the historical analysis, the assumption of additive factorial effects in the NMIP2 ensemble are adopted in the global N₂O budget estimates¹³ as well as the IPCC report¹². The NMIP2 ensemble offers the possibility to examine the additivity by calculating the factorial effects from different combinations of

simulations (Extended Data Tables 2-3). For example, the effect of fertilizer and manure application can be estimated separately from SH1-SH2 and SH1-SH3, but also by using the difference between SH1 and SH12. SH12 (fixing fertilizer and manure application as well as irrigation at the 1850 level) can be used to assess the combined effect and compare it with the sum of individual contributors (Fig. R1). As this analysis reveals, the differences of the effects are small, therefore allowing a robust attribution of the relative effect sizes.

Figure R1. The cumulated contributions led by manure, fertilizer application and irrigation on global (a) NBP, (b) soil N₂O, (c) soil NH₃ and (d) soil NO_x among NMIP2 model members over 1850-2019. The cyan, blue, gold and pink bars indicate values derived by the differences between SH1 and SH2, SH3, SH5 or SH12, respectively, following the NMIP2 protocol (Extended Data Tables 2). Note there are some model members that cannot finish all simulations as well as cover all N-related variables, as we already clarified in Extended Data Table 3.

Saturation effects exist mainly on the N fertilization effects on NBP, which are comprehensively considered in each terrestrial biosphere models in NMIP2 during the historical analysis but not in our future extrapolation. We have added discussions in the main text accordingly. Please refer to the details in the response to Reviewer #2.

b. Do the simulations removing agricultural and non-agricultural sources give the same as the net

simulation when both are simultaneously removed? For example, the sum of the RF (-0.36 W/m²; lines 187-188) does not equal the net RF (-0.34 W/m²; line 156) – which species contribute to this? And why? How does this hamper the analysis?

Response:

The nonlinearity could arise because 1) the atmospheric chemistry, e.g. the relationship between Nr emissions and aerosol concentrations, is not strictly linear; 2) the radiative transfer process is non-linear, as we already discussed with an example in the main text: ‘*The warming effect of non-agricultural N₂O was amplified by the synchronous decline in atmospheric CH₄ because of their interactions in the radiative transfer*³⁵. We quantified this unmasking effect on the N₂O radiative forcing by decreasing CH₄ using a sensitivity experiment with GEOS-Chem-RRTMG (Extended Data Table 1) as a decrease in the non-agricultural N₂O radiative forcing from +0.11 W m⁻² to +0.07 W m⁻².’

Table R1 shows the comparison of RF between the total Nr effects and the sum of agricultural and non-agricultural Nr effects. For most forcing agents, the RFs from each effect are additive, with the exception of NO₃⁻ where we find a notable attenuating effect of about 0.06 W m⁻². This attenuating effect can be explained by the non-linear aerosol chemistry. We have clarified this phenomenon in the main text (as detailed in our following response) but point out that this does not affect our conclusions on the global perspective at decadal timescale. Note that the results are slightly different to the previous version of the manuscript, as we had not added sulphate in the calculation of the sum of contributions in Fig. 4 (also detailed in the following response).

Table R1. The global direct radiative forcing (W m⁻²) induced by all anthropogenic Nr and the sum of agricultural and non-agricultural Nr sources.

Radiative forcing	All anthropogenic Nr (from Fig. 3)	Sum of agricultural and non-agricultural Nr	Agricultural Nr	Non-agricultural Nr
CO ₂	-0.12	-0.12	-0.06	-0.06
N ₂ O	0.16	0.16	0.05	0.11
CH ₄	-0.19	-0.18	-0.04	-0.14
NH ₄ ⁺	-0.08	-0.08	-0.05	-0.03
Aerosols				
NO ₃ ⁻	-0.13	-0.19	-0.09	-0.10
SO ₄ ²⁻	-0.03	-0.02	0.00	-0.02
O ₃	0.05	0.05	0.00	0.05
Total	-0.34	-0.38	-0.19	-0.19

c. Ammonium nitrate formation is non-linear in emissions of NO_x and NH₃ – i.e. both are needed and one is limiting, depending on the environment. Presumably cutting emissions of agricultural sources drastically diminishes emissions of NH₃, producing little ammonium nitrate. Similarly, cutting non-agricultural emissions would eliminate emissions of NO_x (fossil fuel), also leading to little ammonium nitrate. Thus, adding the impact of these two simulations should underestimate the impact when all emissions are included. How did the authors deal with this?

Response:

We appreciate reviewer's comment to help us explain more of the non-linearity in aerosol chemistry. Here, we explain firstly the technical details in GEOS-Chem model as well as our analysis, and then the accordingly revised figure and main text to indicate such non-linearity.

In GEOS-Chem, ammonium and nitrate aerosols are diagnosed independently, so there is actually no 'ammonium nitrate (NH₄NO₃) aerosol' but only the 'ammonium aerosol (NH₄⁺)' and 'nitrate aerosol (NO₃⁻)'. The non-linearity of inorganic aerosol chemistry is fully considered in GEOS-Chem, where the sulphate aerosol (SO₄²⁻) has a higher priority to be generated than nitrate when sufficient ammonia gas exists. As a result, emission changes in NO_x and NH₃ will not only influence the concentrations of ammonium (NH₄⁺) and nitrate (NO₃⁻), but also sulphate (SO₄²⁻) through formation of ammonium sulphate ((NH₄)₂SO₄). Therefore, we believe it is more appropriate for our study to consider the RF changes in NH₄⁺, NO₃⁻ and SO₄²⁻ as the aerosol effects due to by anthropogenic Nr, even if SO₂ emissions are absolutely identical in all simulation experiment and the contribution of SO₄²⁻ as shown in Table R1 is small. As a result, we labelled Fig. 3d as 'Aerosol' rather than 'ammonium nitrate aerosol'.

The change in SO₄²⁻ has already been considered in Fig. 3 (the effects of total anthropogenic Nr), however, not in the original Fig. 4, which distinguishes agricultural and non-agricultural impacts. We have now updated the Fig. 4 with the sulphate contribution, which slightly altered the non-agricultural aerosol effects from -0.13 to -0.15 W m⁻², but did not change the overall directions or magnitude of our previous results.

All of our sensitivity experiments were conducted with present-day anthropogenic emissions of non-Nr related species, which implies that sulphur dioxide (SO₂) emissions are generally high in regions of higher anthropogenic Nr emissions. The existence of SO₂ in the atmosphere buffers the non-linearity of ammonium (NH₄⁺) chemistry, which is why the RF changes in NH₄⁺ and SO₄²⁻ are linearly additive, but the NO₃⁻ effect is not fully additive. In particular, the NO₃⁻ forcing is notably weaker in the agricultural emissions only experiment (Table R1), where NO_x emissions remain high (the new Extended Data Figure 3). We extended the main text as follows:

‘It should be noted that the radiative forcings attributed to agricultural and non-agricultural Nr are affected by the nonlinearity arising in the nitrate aerosol formation, which results in a somewhat stronger net cooling effect by the summing the partitioned effects (-0.38 W m⁻²) compared to the combined estimate (-0.34 W m⁻² in Fig. 3f). The direct radiative forcing of nitrate aerosol is not only weakened with substantial NO_x reductions in the no_nonagriNr experiment, but also reduces due to the associated reduction in the ammonium nitrate aerosol with significant NH₃ emission reduction in the no_agriNr experiment (See Methods, Extended Data Figure 3 and Table S5). Nevertheless, this non-linearity in aerosol chemistry does not significantly influence the additivity of the effects from the ammonium aerosol due to the buffering effect of sulphate (See Table S5 and Methods), as well as the ranking or overall magnitude of the factors by which Nr influences radiative forcing.’

And we also added more demonstration about meanings of the ‘aerosol RF’ in the main text, method as well as figure captions of Figs. 3-4:

In main text:

‘..., where anthropogenic Nr effects on CO₂, N₂O, CH₄, aerosols (including ammonium, nitrate and sulphate, see Methods) and tropospheric O₃ contributed -0.12 [-0.07, -0.17], +0.16 [+0.14, +0.17], -0.19 [-0.12, -0.29], -0.24 [-0.18, -0.28] and +0.05[+0.03, +0.07] W m⁻², respectively.’

In Method:

‘The GEOS-Chem-RRTMG model and sensitivity experiments

...

In particular, GEOS-Chem fully considers the non-linearity of inorganic aerosol chemistry, where sulphate aerosol has higher priority to be generated than nitrate aerosol when sufficient ammonia gas exists. As a result, the sulphate aerosol loadings could also be perturbed by changes in NH₃ and NO_x emissions, despite the fact that the sulfur dioxide emissions are identical in all our experiments. We thus use the sum of direct radiative forcing of ammonium (NH₄⁺), nitrate (NO₃⁻) and sulphate (SO₄²⁻) aerosols to represent the aerosol climate effects induced by anthropogenic Nr.’

Figure 3 caption:

‘**Figure 3. Global direct radiative forcing induced by anthropogenic Nr derived by GEOS-Chem-RRTMG model for (a) CO₂, (b) N₂O, (c) CH₄, (d) Aerosols, (e) O₃, and (f) the net effect (i.e. sum of a-e).** Results were derived by differences in all-sky top-of-atmosphere radiative forcing between CTRL_2019 and No_allNr experiments (see Methods and Extended Data Table 1). **The radiative forcing of aerosols is the sum of the direct radiative forcing contributed by ammonium, nitrate and sulphate aerosols.** Numbers in parentheses represent the global area-weighted averages, while numbers in the brackets indicate the uncertainty ranges based on sensitivity experiments with GEOS-Chem-RRTMG using ± one standard deviation among NMIP2 ensembles as well as ±30% uncertainty in OH and O₃ concentrations (SI text S1.2). Note the Nr effects on global CO₂, N₂O and CH₄ are assumed to be evenly distributed, so that the patterns of these three greenhouse gases are mostly determined by other forcing agents, including the distribution of clouds.

,

Figure 4 caption:

‘**Figure 4. Global direct radiative forcing associated with anthropogenic Nr from agricultural and non-agricultural sources.** (a) The direct radiative forcing values are based on differences of sensitivity experiments between CTRL_2019 and No_agriNr or No_nonagriNr, respectively. **The radiative forcing of aerosols is the sum of the direct radiative forcing contributed by ammonium, nitrate and sulphate aerosols.** Uncertainty bars were diagnosed based on the standard deviation of NMIP2 ensembles of agricultural sources and non-agricultural sources (SI text S1.3). (b) One standard deviation of radiative forcing among all of the global simulated grids.’

We also revised the text of ‘Ammonia (NH₄⁺) and Nitrate (NO₃⁻) aerosols’ in Fig. 1 as ‘Inorganic aerosols’ to avoid confusion.

d. The non-linear O₃ chemistry was also not addressed (Section S2.4 is insufficient). Decreasing NO_x emissions can either decrease or increase ozone production, depending on the local chemical regime. Are these non-linearities important (locally or globally) when isolating source effects or when reducing anthropogenic Nr in the simulations? (impossible to see the trend in O₃ on the bars in Figure 5).

Response:

GEOS-Chem has been widely used to simulate the non-linear O₃ chemistry with good capability¹⁴⁻¹⁶. As a result, the effects of non-linear O₃ chemistry on NO_x and O₃ abundance is addressed by our historical analysis. The surface O₃ concentration is likely to increase with a slight NO_x reduction in urban regions (VOC-limited). It should be noted, however, that the radiative forcing from tropospheric O₃ is induced by the total O₃ column rather than only the surface O₃, while most of the enhanced O₃ with NO_x decrease are observed on the ground level. With substantial NO_x reductions, our simulations do not show significant chemical regime effects on the global column O₃ (Fig. R3a) and the O₃ radiative forcing (Fig. 3). The O₃ differences aligned with the similar pattern with anthropogenic NO_x sources (Fig. R3b) although the global atmospheric transport spreads O₃ more evenly around the north hemisphere. Meanwhile, the non-agricultural sources dominate the NO_x emissions (the new Extended Data Fig. 3) as well as the radiative forcing of O₃ (Fig. 4). As a result, the non-linearity will have no significant effects on isolating the source effects.

Figure R3. The global patterns of the differences in (a) tropospheric O₃ burden and (b) NO_x emissions between CTRL_2019 and no_allNr experiments.

For the future extrapolation, the linear assumption between NO_x emissions and O₃ concentrations is parsimonious, and the implications are addressed in the uncertainty discussions in Sect. S2.4. Non-linear effects in atmospheric chemistry have already been assigned an uncertainty of 30% on the radiative forcing of O₃, similar to the uncertainty in atmospheric OH in the CH₄ box model, and are reflected in the uncertainty estimates of this study.

e. What is the implication of neglecting any of these non-linearities on the uncertainty analysis in the SI?

The uncertainty analysis conducted to estimate variance in the RF estimates has accounted for the non-linearities. We assess the effect of uncertainty in Nr-related fluxes by varying the ensemble mean based on

the NMIP2 standard deviation, which implies that the non-linearity in soil biogeochemistry response to anthropogenic Nr as represented in the terrestrial biosphere models is accounted for. We then investigate the effects of these variations using the complete chemistry of GEOS-Chem, and therefore accounting for the non-linearities in the aerosol and ozone chemistry using GEOS-Chem.

We only adopt a linear additive assumption of effect for the future extrapolation, where most of the assumptions are reasonable as discussed above. We have added additional analysis and discussions regarding the possible impact of N saturation on NBP, see the responses to Reviewer #2.

5. Section on future Nr reductions is not particularly informative. The simultaneous and persistent Nr reductions are not realistic, nor are the application of the radiative forcing sensitivities, and thus the results should not be interpreted in a policy context. In particular, NO_x and NH₃ emissions are unlikely to follow the same trajectory, so the aerosol response here does not correspond to any meaningful scenario (it would be more interesting to separate the response to NO_x cuts and NH₃ cuts). I recommend trimming this section down to a few sentences. I also recommend cutting Figure 5 which provides little insight (very difficult to see the differences in timescales and identify which components contribute to this; the differences in the 4 scenarios are also not very useful/apparent).

Response:

We acknowledge this instructive comment, which pushed us to investigate the effects of more realistic scenarios. In the revised manuscript, three SSP scenarios (SSP 1-2.6, SSP 3-7.0 and SSP 5-8.5) are used to estimate the future climate effects of anthropogenic Nr. The future fossil fuel emissions as well as N deposition are from the input4MIPs dataset (<https://esgf-data.dkrz.de/projects/input4mips-dkrz/>). Future fertilizer and manure applications are based on the IMAGE prediction¹⁷, which only extends data until 2050. To keep the data consistency of this study, the ratio of future Nr-related sources to the 2019 level for each dataset is firstly calculated (Fig. R4), and then the ratios are used to scale present-day (2019) emission levels in this study. A technical note is that the future fossil fuel emission of N₂O is not included in input4MIPs, which is replaced by the scaling factor of NO_x. In this way, the future scenarios become more realistic, e.g. increases in fertilizer application to meet the food demands and decreases in fossil fuel emissions.

Figure R4. Future scale factors of global anthropogenic Nr inputs on (a) N deposition, (b) fossil fuel NO_x and (c) NH_3 , (d) manure and (e) fertilizer applications. For each SSP scenario, the annual scale factors were calculated by the ratios of future anthropogenic Nr to the 2019 levels, which is indicated by the dashed purple lines. The N deposition and fossil fuel data are from the input4MIPs (<https://esgf-data.dkrz.de/projects/input4mips-dkrz/>). Manure and fertilizer predictions are obtained from Mogollon, et al. ¹⁷

Figure R5 (the new Fig. 5 in the main text) shows the future climate effects of anthropogenic Nr under the SSP scenarios. Despite changes in the absolute trends and timing of the climate effect, the overall conclusion and the implications of our manuscript remain similar to our previous extrapolation with fixed N reduction (Details in Response to Reviewer #2). The strong reduction in fossil fuel NO_x under the SSP 1-2.6 scenario dominated the warming effect relative to 2019, with the increases in the CH_4 lifetime as well as the decreases in aerosol solar diffusion. In the SSP 3-7.0 scenario, future manure and fertilizer application substantially increase, resulting in a stronger warming effect due to increased N_2O , which is offset by the stronger cooling effects from aerosols due to the increase in agricultural and fossil fuel sources of NH_3 . In the SSP 5-8.5 scenario Nr emissions remain approximately at present-day levels, so the climate effects of anthropogenic Nr remains similar until 2050. With the purpose to alleviate the negative environmental effects of anthropogenic Nr, these results still implicate that we need stronger reduction in other greenhouse gases accompanied with the ‘clean-Nr’ scenario to achieve both environmental and climatic benefits.

Figure R5. Global climate responses to future changes in anthropogenic Nr. (a)-(c) Global direct radiative forcing relative to pre-industrial levels (1850) in response to anthropogenic Nr following SSP 1-2.6, SSP 3-7.0 and SSP 5-8.5 scenarios. The net changes in radiative forcing were indicated by the orange lines, while dashed purple lines indicated the zero reference. The uncertainty bars were calculated by the percentage ranges in direct radiative forcing derived from Fig. 3

In light of these updated results, we have re-written the final section of the manuscript as follows:

‘To illustrate the likely consequences of potential future changes in anthropogenic Nr, we next use the understanding gained in the previous section in a simplified analysis using three representative scenarios from the Shared Socioeconomic Pathways (SSPs; see Methods). The SSP 1-2.6 assumes a ‘Nr cleaner’ scenario with strong reduction in fossil-fuel-based NO_x emission but relatively unchanged magnitudes of fertilizer and manure application to meet global food demands (Extended Data Fig. 4). These emissions changes lead to a net warming effect of +0.09 $W m^{-2}$ by 2050s relative to 2019 dominated by the increased CH₄ lifetime and a decreased direct aerosol effect (Fig. 5a). In the SSP 3-7.0 scenario, the future fossil fuel sources of Nr remain close to the 2019 level. Enhanced fertilizer and manure applications increase N₂O emissions and lead to a stronger N₂O warming effect of +0.06 $W m^{-2}$ in 2050s relative to 2019, which is compensated by the cooling effects of increased aerosol loadings (-0.03 $W m^{-2}$ enhancement in 2050s relative to 2019) and enhanced terrestrial carbon sequestration (-0.04 $W m^{-2}$ enhancement in 2050s relative to 2019). However, bounding assumptions on the magnitude of N saturation (See SI text S2.1 and Fig. S4) suggest that carbon sequestration effect might be overestimated by about 0.02 $W m^{-2}$. Finally, the SSP 5-

8.5 scenario predicts a generally unchanged level of anthropogenic Nr compared to 2019, thus compensating changes in climate forcing. These results implicate that stronger reductions in green-house gases emissions are required accompanying with the ‘clean-Nr’ scenario to achieve both environmental benefits and climate change mitigation.’

We also revised the description of future SSP analysis in the Methods:

‘Linear extrapolation of climate effects under the SSP scenarios

We extrapolated the future climate effects due to changes in anthropogenic Nr under three representative SSP scenarios (SSP 1-2.6, SSP 3-7.0 and SSP 5-8.5). The future fossil fuel emissions and N deposition were from the input4MIPs dataset (<https://esgf-data.dkrz.de/projects/input4mips-dkrz/>). Future fertilizer and manure applications were based on the IMAGE predictions until 2050¹⁷. To maintain consistency in this study, the future Nr-related sources were scaled to 2019 levels for each dataset (Extended Data Fig. 4). Since the future fossil-fuel based emission of N₂O is not included in input4MIPs, the future development of this source of N₂O was scaled to the future development of fossil-fuel based NO_x.

To estimate the magnitude of climate effects of anthropogenic Nr under the SSP scenarios, we built a simple linear framework based on the following assumptions: 1). The change in radiative forcing of atmospheric green-house gas attributable to Nr-related changers was linearly related to their change in atmospheric concentrations, while the direct radiative forcing of short-lived gases or aerosols was linearly related to total precursors’ emissions⁹⁻¹¹ at the corresponding year; 2). The effects of anthropogenic Nr on soil-gas fluxes were linearly determined by anthropogenic Nr addition, including both fertilizer/manure application and N deposition; Then a simple model was established based on the GEOS-Chem diagnosed direct radiative forcing of individual compound to calculate the radiative forcing relative to 1850:

$$RF_Nr_CO_2_{yr} = RF_Nr_CO_2_{2019} + \sum_{yr=2020}^t (NBP_{fertilizer, yr} + NBP_{manure, yr} + NBP_{Ndep, yr}) \times \frac{\alpha}{\delta_{CO_2}} \times S_{CO_2} \quad (6)$$

$$RF_Nr_N_2O_{yr} = RF_Nr_N_2O_{2019} + ([N_2O]_{yr} - [N_2O]_{2019}) \times S_{N_2O} \quad (7)$$

$$RF_Nr_CH_4_{yr} = RF_Nr_CH_4_{2019} + ([CH_4]_{yr} - [CH_4]_{2019}) \times S_{CH_4} \quad (8)$$

$$RF_Nr_aerosol_{yr} = \frac{NO_x_{yr} + NH_3_{yr}}{NO_x_{2019} + NH_3_{2019}} \times RF_Nr_aerosol_{2019} \quad (9)$$

$$RF_Nr_O_3_{yr} = \frac{NO_x_{yr}}{NO_x_{2019}} \times RF_Nr_O_3_{2019} \quad (10)$$

Where the $RF_Nr_CO_2_{yr}$, $RF_Nr_N_2O_{yr}$, $RF_Nr_CH_4_{yr}$, $RF_Nr_aerosol_{yr}$ and $RF_Nr_O_3_{yr}$ represent the direct radiative forcing associated with anthropogenic Nr of each gas at the yr year relative to 1850. The values in 2019 were derived from the differences between CTRL_2019 and No_allNr experiments (-0.12 W m⁻², +0.16 W m⁻², -0.19 W m⁻², -0.24 W m⁻² and +0.05 W m⁻², respectively, Fig. 3). The sensitivities (S_{CO_2} , S_{N_2O} , and S_{CH_4}) of radiative forcing to greenhouse gas concentrations were derived from the other eight GEOS-Chem sensitive experiments (See SI text S1.3 and Table S2)

In particular, we calculated the reduction effect as follows:

- $NBP_{fertilizer, yr}$, $NBP_{manure, yr}$ and $NBP_{Ndep, yr}$ represented the NBP contributed by fertilizer, manure and N deposition in the yr year, which is calculated by multiplying the NMIP2 ensemble mean present-day (average of 2015-2019) contributions and the corresponding scaling factors in Extended Data Fig. 4.
- The N_2O and CH_4 concentrations in the yr year ($[N_2O]_{yr}$ and $[CH_4]_{yr}$) were derived by the simple N_2O and CH_4 box models (Eq. 2 and Eqs. 3-5) starting from $[N_2O]_{2019}$ (N_2O concentrations in CTRL_2019 experiments) and $[CH_4]_{2019}$ (CH_4 concentrations in CTRL_2019 experiments), respectively. N_2O (in N_2O box model) and NO_x (in CH_4 box model) emissions from both fossil fuel combustion and anthropogenic-Nr-induced soil emissions were reduced relative to emissions in 2019 with the scaling factors accordingly (Extended Data Fig. 4), while the rest sources kept the same as 2019.
- For short-lived compounds (aerosols and O_3), NO_x yr (or NH_3 yr) indicated the NO_x (or NH_3) emissions from both fossil fuel and soil by applying the scaling factors on each sector (Extended Data Fig. 4) in the yr year.'

6. The Methods are complex with many steps; the section should be edited for clarity. Perhaps the authors would consider developing a summary figure that shows exactly what parameters come from what methods?

Response:

Thank you for the suggestion. We added one summary figure for methods:

Extended Data Figure 6. A summary figure for the data and methods. The solid black arrows indicate the main methods or data, which are introduced in the Methods section. The dashed red arrows indicate the uncertainty analysis and the consequent sensitivities to radiative forcing, which are documented in the SI text. The main figures in this study are highlighted accordingly with the figure indexes.

We added a summary description in the beginning of the Method section:

‘A summary figure for the data and methods in this study is given in Extended Data Fig. 6. Here we introduce each part in details.’

At the same time, we moved the original Extended Data Fig. 2 into the supplementary (related to the ensemble uncertainties) and removed the original Extended Data Fig. 4 (the biomass emissions) due to the limited figure spaces in the Extended Data.

a. Extended Table 2: I suggest adding another table that identifies which pairs of simulations were used to estimate each N factor. This can be used to help explain equation 1.

Response:

Due to the limited space of Extended Data, we think it might be not necessarily to have a new table. Instead, we have added explanation for each sensitivity experiment in ED Table 2:

Extended Data Table 2. Experiment configurations in NMIP2. The letter ‘T’ indicates a transient change as the forcing data from 1850 to 2020. The 1850 or 1901-1920 indicates that the corresponding forcing is fixed in this year or time periods. Climate over 1850-1900 are replaced by randomly selected years from 1901-1920 for all experiments due to missing climate forcing data. The last row of the table indicates how the factorial contribution for each factor in this study is calculated from the differences between corresponding experiments.

Experiment name	Fertilizer	Manure	N deposition	Irrigation	Land use changes	CO ₂	Climate
SH0	1850	1850	1850	1850	1850	1850	1901-1920
SH1	T	T	T	T	T	T	T
SH2	1850	T	T	T	T	T	T
SH3	T	1850	T	T	T	T	T
SH4	T	T	1850	T	T	T	T
SH5	T	T	T	1850	T	T	T
SH6	T	T	T	T	1850	T	T
SH7	T	T	T	T	T	1850	T

SH8	T	T	T	T	T	T	1901-1920
SH11	1850	1850	T	1850	1850	T	T
SH12	1850	1850	T	1850	T	T	T
Factor attribution in this study	SH1-SH2	SH1-SH3	SH1-SH4	SH1-SH5	SH12-SH11	SH1-SH7	SH1-SH8

b. The use of the equation 1 and the term “factorial configuration” makes this a bit unnecessarily complicated to read. Effectively a series of sensitivity experiments are differenced from the baseline simulation to quantify each factor. It would be helpful to describe this with an example, rather than using equations with non-intuitive indices. I think this is true for all the equations used in the methods. They are difficult to parse and I would therefore suggest that the methods be explained in words and if the authors insist on the equations that these be placed and described in the supplement.

Response:

Sorry for the confusion. As suggested above, adding one line in Extended Data Table 2 is very helpful to explain the factor attribution. Therefore, we remove all equations with the form of ‘factorial configuration’ in the method and Extended Data Table 1 (the comment below). However, equations of the box models and the future exploration are still necessary to make it clear and easy to follow.

The according methods section is revised as follow:

‘The effect of anthropogenic fertilization, manure application, N deposition, irrigation, LUC, CO₂ elevation and climate changes on simulations of NBP, N₂O and NO_x in the NMIP2 ensemble are quantified based on the differences among a series of sensitivity experiments (Extended Data Table 2). The contribution of LUC is quantified by the difference between the SH12 and SH11 experiment (rather than differences between SH1 and SH6) to avoid the confounding effects from changes in fertilizer and manure application. N₂O and NBP fluxes are accessible for all of the eight NMIP2 members, while the NO_x flux is only available with CLASSIC, OCN and ORCHIDEE (Extended Data Table 3).

The NH₃ emission estimate of 39.0 Tg N yr⁻¹ by the NMIP2 ensemble, which accounts for agricultural NH₃ soil emissions but not those emissions from livestock manure, is close to the CEDS agricultural NH₃ emissions (38.2 Tg N yr⁻¹) for the year 2019. However, the large inter-model variability (SI Fig. S1d) makes the direct use of these simulations to quantify the anthropogenic effect susceptible to biases in individual models. Therefore, we retained the original CEDS agricultural NH₃ emission in this study, and attribute soil NH₃ emission changes by first applying a fixed ratio (48%) on the total agricultural NH₃ emissions in 2019, while the rest part (52%) is led by livestock according to Liu, et al. ¹⁸ and then scaling the anthropogenic Nr influence on soil NH₃ emissions according to the temporal evolution of soil NH₃ emissions in the NMIP2 ensemble.’

The CO₂ box model equation is also revised as:

$$\Delta CO_2 = - \sum_{yr=1850}^{2019} (NBP_{fertilizer,yr} + NBP_{manure,yr} + NBP_{Ndep,yr}) \times \frac{\alpha}{\delta_{CO_2}} \quad (1)$$

Where ΔCO_2 indicates changes in atmospheric CO₂ concentrations (ppmv) from 1850 to 2019 due to anthropogenic Nr. The accumulated NBP induced by fertilizer and manure applications and N deposition during 1850-2019 was calculated from NMIP2 ensemble mean (Extended Data Table 2). δ_{CO_2} was 2.12 Pg C ppmv⁻¹ following Ballantyne, et al.¹⁹. The partitioning constant α accounting for the ocean-borne fraction of atmospheric CO₂ increase was determined to be 0.61 given the historical (1850-2019) increases in the atmosphere (235 PgC) and ocean (150 PgC) carbon estimated from the global carbon budget.²⁰

c. Extended Table 1 is VERY difficult to read with the equations (same point as above). I suggest you simplify with words, but also add the global NO_x and NH₃ emissions for each line so that they can be compared with Table S1

Response:

The Extended Table 1 is now revised as:

Extended Data Table 1. Summary of the GEOS-Chem sensitive experiments. The global NO_x and NH₃ emissions are derived by integrating CEDS and NMIP2 ensembles. Atmospheric concentrations of N₂O, CO₂ and CH₄ are retrieved by the box models.

Experiment name *	NO _x emissions	NH ₃ emissions	N ₂ O concentrations	CO ₂ concentrations	CH ₄ concentrations
CTRL_2019	45.97 Tg N yr ⁻¹	50.47 Tg N yr ⁻¹	326.53 ppbv	409.9 ppmv	2015.02 ppbv
No_allNr	9.12 Tg N yr ⁻¹	9.13 Tg N yr ⁻¹	286.33 ppbv	420.38 ppmv (409.9+10.48)	2354.16 ppbv
No_agriNr	43.63 Tg N yr ⁻¹	21.41 Tg N yr ⁻¹	309.27 ppbv	415.19 ppmv (409.9+5.29)	2032.40 ppbv
No_nonagriNr	11.46 Tg N yr ⁻¹	38.19 Tg N yr ⁻¹	303.59 ppbv	415.09 ppmv (409.9+5.19)	2335.91 ppbv
Supplement experiment					
No_nonagriNr_lowCH ₄	11.46 Tg N yr ⁻¹	38.19 Tg N yr ⁻¹	303.59 ppbv	415.09 ppmv (409.9+5.19)	2015.02 ppbv

*The CTRL_2019 experiment includes all anthropogenic Nr sources; The No_allNr experiment excludes anthropogenic Nr sources of fossil fuel, fertilizer and manure application and N deposition; The No_agriNr experiment excludes anthropogenic Nr sources of fertilizer and manure application; The No_agriNr

experiment excludes anthropogenic Nr sources of fossil fuel and N deposition. The livestock NH₃ emission is attributed as agricultural sources.

d. Line 528: is there a reference for this approach for calculating the methane lifetime to OH? If not, can you show that this equation is a good approximation?

Response:

The CH₄ box model is based on the Table 4.11 from IPCC [2001] Chapter 4 and fully described by Olivieri, et al. ²¹, which is a Zenodo document (<https://zenodo.org/records/5293940>) that also provides uncertainty analyses, and which has been the basis of peer-reviewed publications^{22,23}. Both this Zenodo document and our Extended Data Fig. 5b together showed the good capability of this CH₄ box model in simulating historical CH₄ trends.

e. Lines 525, 531-532: How uncertain are the assumed and calculated CH4 lifetimes? What about the sensitivity factors (also: need to define a sensitivity factor in the text)? How does this impact the results?

Response:

Those sensitivity factors that determines CH₄ lifetime are diagnosed from comprehensive atmospheric chemistry transport models, therefore it is difficult for us to quantify their uncertainties. However, in this work, the emissions of CO, VOCs and temperature are identical in both cases with/without anthropogenic NO_x, so we assume that the uncertainties for those factors would be canceled by calculating the differences. To account for uncertainties in the NO_x effect on CH₄ lifetime, we include this implicitly due to the ±30% range associated with the uncertainties in OH concentrations that are closely associated with NO_x concentration changes, where the ±30% range is sufficiently large to cover the whole uncertainty range (SI text S1.2).

f. The uncertainties on NOx and NH3 emissions are independent. Therefore, these emissions should be scaled separately in the GEOS-Chem sensitivity experiments for the uncertainty analysis.

Response:

To address this comment, we added four more sensitivity experiments with the GEOS-Chem model to separately test the upper and bottom boundary of the RF_{aerosol} estimates. As summarized in Table R2, the NH₃ variation dominates the uncertainties in the direct radiative forcing because the uncertainty in the NH₃ emissions is much larger than that of NO_x (SI Fig. S2). Separating emissions will not change the results of uncertainty analysis.

Table R2. The direct radiative forcing of ammonium (NH₄⁺), nitrate (NO₃⁻) and sulphate (SO₄²⁻) aerosols in the central estimates (Fig. 3 in the main text) as well as the uncertainty analysis. The ‘NO_x and NH₃’ results with max and min are from SI Table 1 experiments with ‘Aerosol_max’ and ‘Aerosol_min’, respectively.

The ‘NO_x only’ or ‘NH₃ only’ is the experiment which only perturbs the NO_x or NH₃ emission with one standard deviation in NMIP2 ensembles, but keeps the other the same as the experiment of central estimate.

		RF _{NH4+} (W m ⁻²)	RF _{NO3-} (W m ⁻²)	RF _{SO42-} (W m ⁻²)	RF _{aerosol} (W m ⁻²)
Central estimates		-0.079	-0.127	-0.035	-0.241
	NO _x and NH ₃	-0.110	-0.130	-0.041	-0.281
Max	NO _x only	-0.079	-0.128	-0.037	-0.244
	NH ₃ only	-0.110	-0.130	-0.039	-0.279
	NO _x and NH ₃	-0.047	-0.111	-0.023	-0.181
Min	NO _x only	-0.079	-0.127	-0.033	-0.239
	NH ₃ only	-0.048	-0.112	-0.024	-0.184

g. Uncertainty analysis in S2 – please refer to the figures and numbers where relevant. It is unclear how values discussed here relate to Figure 1 and Figure S1.

Response:

We have added sources of the numbers accordingly. However, we need to clarify that the S2 text is to discuss all of the underlying uncertainties throughout the whole analysis, while the Fig. 1 and Fig. S1 (now is Fig. S2) are mainly used to quantify the uncertainty range in Fig. 3. Therefore, the numbers in the S2 text are not all necessarily from the S1 section.

h. How do values in Table S1 relate to Figure S1? For example, the minimum NH3 emissions in present-day in Figure S1 appear to be ~10 Tg/yr not 0.0 as in Aerosol_max in Table S1. (maximum in Figure S1 ~45 not 27 as in Table S1). Similar discrepancies for NOx.

Response:

The experiments in Table S1 is varied based on the ‘no_allNr’ case rather than ‘2019_CTRL’ case. The upper and bottom boundaries of each component are derived by calculating the differences between ‘2019_CTRL’ and the Table S1 experiments. The global NH₃ emission in the 2019_CTRL experiment is 50.5 Tg N yr⁻¹, and the bottom boundary in Fig. S1 (now is Fig. S2) is 23.4 Tg N yr⁻¹. The difference of 27.1 Tg N yr⁻¹ is what we showed in Fig. S1 (now is Fig. S2). However, when removing the anthropogenic Nr sources (what we showed in Fig. S1), the high estimate of NH₃ emissions (62.2 Tg N yr⁻¹) exceeded our current global NH₃ emissions (50.5 Tg N yr⁻¹). As a result, we can only set it as zero to avoid a ‘negative emission’ in GEOS-Chem, which was already explained in the original Table S1:

‘Due to the large uncertainty in estimating global NH₃ emissions, as we also demonstrated in the main text, the upper boundary of anthropogenic contributions on NH₃ (62.2 Tg N yr⁻¹ in 2019) have already exceed

our current global total estimates (50.5 Tg N yr⁻¹ in 2019). As a result, we assumed the global NH₃ anthropogenic emissions to be zero in the sensitive experiments as the guess of maximum contribution of anthropogenic N_r on NH₃.

Minor

1. Figure 2 and text that follows: NBP is never defined in the text.

Response:

The full name ‘net biome productivity’ has been added

2. Line 66: replace “The” with “An”

Response:

Revised.

3. Lines 106, 192, 235: fossil fuel combustion is singular, not plural

Response:

Revised.

4. Lines 107-108: air pollution impacts on vegetation should be included in this list

Response:

Here we referred to the factors in the NMIP2 configuration. The air pollution impacts on vegetation will only be discussed in the end. To clarify, we revised the sentence as:

‘Other anthropogenic factors in the configuration of NMIP2, e.g. irrigation, land-use change (LUC),’

5. Line 134: how similar is the 45 Tg/yr from CEDS to the NMIP2 ensemble mean? It looks similar from Fig 2d so perhaps worth stating in text? If Figure 2d is already re-scaled to CEDS, this should be stated in the caption.

Response:

As we mentioned in the methods:

‘The NH₃ emission estimates by the NMIP2 ensemble, which accounts for agricultural NH₃ soil emissions but not those emissions from livestock manure, is close to the CEDS agricultural NH₃ emissions (39.0 and 38.2 Tg N yr⁻¹ for the year 2019)’

The Fig. 2d is already re-scaled to CEDS. We have added statement in the figure caption:

‘Figure 2. Historical emissions and terrestrial carbon fluxes based on CEDS inventory and NMIP2 ensemble means. All of the non-agricultural (fossil fuel combustion) sources in CEDS are indicated by the pink bars, while other colors indicate factor contributions based on NMIP2 ensemble means. **The soil NH₃ emissions have been scaled by the CEDS agricultural emissions (See Methods).** The fire emission of each component is not included within the current NMIP2 configuration. The contributions of each factor were averaged over 1880s, 1910s, 1940s, 1970s, 2000s, and 2020s with a 5-yr time window, where the direct anthropogenic Nr effects are highlighted by slashes. Black lines indicate the ensemble mean annual flux of each compound, and the error bars indicate one standard deviation among different NMIP2 members.’

6. Figure 4: The magnitude of the bars is difficult to assess. I suggest either expanding the x-dimension of the figure, or including the values for each bar on the figure itself.

Response:

The Fig. 4 has been revised with values for each bar.

Meanwhile, we revised the main text regarding to the slight changes by involving SO₄²⁻ RF in Fig. 4:

‘Figure 4a showed the net climate effects derived from agricultural and non-agricultural sources were comparable (-0.19 [-0.03, -0.38] W m⁻² and -0.19 [-0.11, -0.31] W m⁻², respectively)’

7. Line 254: strange that this is the first mention of fires. Should the fire emissions (even if small, and generally constant historically, as suggested in Extended Data) not be included in Figure2? This should be addressed when discussing Figure 2.

Response:

Although some of the NMIP2 models account for wildfire-based emissions, the available output does not allow to partition land carbon and nitrogen fluxes into fire-based and undisturbed fluxes. The Fig. 2 caption is stated as ‘Historical emissions and terrestrial carbon fluxes based on CEDS inventory and NMIP2 ensemble means’, where the fire emissions are not included. To make it clear, we added ‘The fire emission of each component is not included within the current NMIP2 configuration.’ in the Fig. 2 caption.

8. Lines 277-278: It would be useful to explain “a win-win” as many may not be familiar with this expression.

Response:

We revised the descriptions as:

‘Our findings thus imply that in order to alleviate the negative environmental effects of Nr without larger rates of climate change, stronger reductions in the emission of green-house gases CO₂ and CH₄ need to be implemented concurrently with Nr reductions.’

9. Line 423: Is the Nr deposition from CCMI which is used as an input to the terrestrial models consistent with the Nr deposition simulated for the same year with the GEOS-Chem CTM?

Response:

Yes, these estimates are broadly consistent with small regional deviations. Fig. R6 shows the global pattern of the annual-mean N depositions from CCMI and GEOS-Chem. In general, they share the similar pattern with closed global total budgets (65.4 Tg N v.s. 63.8 Tg N).

Figure R6. The global patterns of the annual-mean N deposition in 2019 from the (a) CCMI (used as NMIP2 inputs), (b) GEOS-Chem and (c) their differences. The N deposition in GEOS-Chem is summed by both dry and wet depositions of gas-phased NH_3 , HNO_3 , N_2O_5 and NO_2 as well as the particle-phased NH_4^+ and NO_3^- .

10. Line 451: missing brackets around $(\text{NGSHI}_{j,\text{yr}} - \text{NGSHI}_{j,\text{yr}})$

Response:

The equation has been removed.

11. Lines 464-465: given this, it seems the authors should emphasize the uncertainty in emissions of NH_3 in their conclusions

Response:

The large uncertainties of NH_3 (Fig. 2d) has already been considered in our uncertainty analysis and showed in the uncertainty range of aerosols. However, not only NH_3 , but also the soil N_2O as well as the N fertilization effects on NBP have similarly unresolved uncertainties, as we discussed in the SI text S2. Therefore, we would rather mention all uncertainties together rather than only focus on one aspect.

We added discussions as:

‘In this study, several processes, including the influences of aerosols or O₃ on terrestrial carbon fluxes, aerosol-cloud interactions, N addition effects on soil CH₄ uptakes and N fertilization on marine biogeochemistry were not included due to the likely small effect on climate or uncertainty to quantify the global effect (See details in SI text S2.5). **It should be noted that the future CO₂ cooling due to CO₂ uptake on land may be overestimated in our study because we omit the contribution of fossil-fuel based CO₂ emissions from N fertilizer production by the Haber Bosch, and more importantly, terrestrial ecosystems exposed to high chronic N additions may become N saturated within the next few decades and contribute less to terrestrial C storage (See details in SI text S2.1). Uncertainties also remain in quantifying soil N₂O, NO_x and NH₃ emissions (See details in SI text S2.2 to S2.4).** To reduce uncertainties and gain a more comprehensive understanding of potential feedbacks, the development of more integrative Earth system models including key interactions among processes of terrestrial and marine biogeochemistry, atmospheric chemistry, climate dynamics and radiative processes would be required.

12. Line 507: missing brackets on $d[N_2O]/dt$ to be consistent with $d[CH_4]/dt$ in line 518

Response:

Revised.

- 9 Unger, N., Shindell, D. T., Koch, D. M. & Streets, D. G. Air pollution radiative forcing from specific emissions sectors at 2030. *Journal of Geophysical Research-Atmospheres* **113**, doi:10.1029/2007jd008683 (2008).
- 10 Chen, Y. J. *et al.* Investigating the Linear Dependence of Direct and Indirect Radiative Forcing on Emission of Carbonaceous Aerosols in a Global Climate Model. *Journal of Geophysical Research-Atmospheres* **123**, 1657-1672, doi:10.1002/2017jd027244 (2018).
- 11 Thornhill, G. D. *et al.* Effective radiative forcing from emissions of reactive gases and aerosols - a multi-model comparison. *Atmospheric Chemistry and Physics* **21**, 853-874, doi:10.5194/acp-21-853-2021 (2021).
- 12 Canadell, J. G. *et al.* Global Carbon and other Biogeochemical Cycles and Feedbacks. In *Climate Change 2021: The Physical Science Basis. Contribution of Working Group I to the Sixth Assessment Report of the Intergovernmental Panel on Climate Change*. 673–816 (2021).
- 13 Tian, H. Q. *et al.* A comprehensive quantification of global nitrous oxide sources and sinks. *Nature* **586**, 248–+, doi:10.1038/s41586-020-2780-0 (2020).
- 14 Wu, S. L., Duncan, B. N., Jacob, D. J., Fiore, A. M. & Wild, O. Chemical nonlinearities in relating intercontinental ozone pollution to anthropogenic emissions. *Geophysical Research Letters* **36**, doi:10.1029/2008gl036607 (2009).
- 15 Christian, K. E., Brune, W. H. & Mao, J. Q. Global sensitivity analysis of the GEOS-Chem chemical transport model: ozone and hydrogen oxides during ARCTAS (2008). *Atmospheric Chemistry and Physics* **17**, 3769-3784, doi:10.5194/acp-17-3769-2017 (2017).
- 16 Qu, Z., Henze, D. K., Cooper, O. R. & Neu, J. L. Impacts of global NO_x inversions on NO₂ and ozone simulations. *Atmospheric Chemistry and Physics* **20**, 13109-13130, doi:10.5194/acp-20-13109-2020 (2020).

- 17 Mogollon, J. M. *et al.* Assessing future reactive nitrogen inputs into global croplands based on the shared socioeconomic pathways. *Environmental Research Letters* **13**, doi:10.1088/1748-9326/aab212 (2018).
- 18 Liu, L. *et al.* Exploring global changes in agricultural ammonia emissions and their contribution to nitrogen deposition since 1980. *Proceedings of the National Academy of Sciences of the United States of America* **119**, doi:10.1073/pnas.2121998119 (2022).
- 19 Ballantyne, A. P., Alden, C. B., Miller, J. B., Tans, P. P. & White, J. W. C. Increase in observed net carbon dioxide uptake by land and oceans during the past 50 years. *Nature* **488**, 70-+, doi:10.1038/nature11299 (2012).
- 20 Friedlingstein, P. *et al.* Global Carbon Budget 2021. *Earth System Science Data* **14**, 1917-2005, doi:10.5194/essd-14-1917-2022 (2022).
- 21 Olivié, D., Höglund-Isaksson, L., Klimont, Z. & Olivié, a. K. v. S. Boxmodel for calculation of global atmospheric methane concentration. *Zenodo*, doi:10.5281/zenodo.5293940 (2021).
- 22 Whaley, C. H. *et al.* Model evaluation of short-lived climate forcers for the Arctic Monitoring and Assessment Programme: a multi-species, multi-model study. *Atmospheric Chemistry and Physics* **22**, 5775-5828, doi:10.5194/acp-22-5775-2022 (2022).
- 23 Whaley, C. H. *et al.* Arctic tropospheric ozone: assessment of current knowledge and model performance. *Atmospheric Chemistry and Physics* **23**, 637-661, doi:10.5194/acp-23-637-2023 (2023).

Reviewer Reports on the First Revision:

Referees' comments:

Referee #1 (Remarks to the Author):

Very interesting and novel paper examining the net radiative forcing effects of anthropogenically derived reactive nitrogen (Nr). In the end, the net cumulative effect of the anthropogenic Nr is a cooling of 0.34 W m^{-2} [0.20, 0.50] relative to the year 1850. The manuscript is well-written, very thorough and has global implications, so it is certainly appropriate for the journal "Nature". The methodology is appropriate and well described, and the results are clearly shown through a number of nicely organized figures. The authors have run an ensemble of model runs to investigate the uncertainties and this is also presented clearly and concisely in the manuscript. The conclusions of the study are also well supported by the data. I will say though, that it is odd to see CO₂ and CH₄ on the cooling side of the equation (e.g. Figure 5). I understand that it is due to C sequestration and a shortening of the CH₄ lifespan in the atmosphere, but still.

The authors have also done a very nice, and thorough, job of responding to the concerns of the previous reviewers. I also have a few small editorial type revisions (see below).

Line numbers are from "clean" document

Line 67: I would say "An earlier study".

Line 162: delete the space after the "(".

Line 201: Not really a revision, but I find it interesting the NO_x, which is mainly produced via fossil fuel combustion, dominates the cooling effects of non-ag sources. So, reducing fossil fuel combustion will reduce warming from CO₂, but at the same time reduce cooling from NO_x.

Line 243: I think you should clarify here that this is just a net warming effect from the changes to Nr and not an overall net warming from a strong reduction in fossil fuel use.

Line 253: "imply" not "implicate".

Line 456: should that be carbon monoxide?

Line 484 (and 486): I would suggest deleting the "s" from fossil fuel combustions (combustion).

Line 550 (also line 583): Greenhouse, not green-house.

Extended data Figure 3: in e and g, I assume those lines are related to shipping?

Nice work with Extended data figure 6.

Referee #2 (Remarks to the Author):

The authors addressed all my comments appropriately and the explanations given on the long-term non-linearity effects of Nr deposition on C sequestration and soil N₂O emissions are sensible and reflect the current state of knowledge. I appreciate the significant improvement of explanations on the underlying models and data including Extended Data Figure 6 and the attempt to provide realistic uncertainty estimates.

The paper provides a great summary of the current knowledge of global Nr effects on the climate system, including all of its major processes and underlying complexities. Very nice work

Referee #3 (Remarks to the Author):

The authors have made substantive changes to their manuscript to address issues raised in review. In particular, they have clarified their methods, adopted a more realistic future scenario, and made some attempt to address the question of non-linearities. There are a few items that I think should be clarified in the manuscript text. None of these are issues that will substantively change the conclusions of the manuscript, but should be addressed for accuracy/completeness. Once the authors have made these changes, I recommend that the manuscript be accepted for publication.

1. Atmospheric chemistry:

a. Correction: in the response to reviews the authors say that "there is actually no ammonium nitrate" in GEOS-Chem. This is not quite right. Ammonium nitrate is simulated in GEOS-Chem (as is ammonium sulphate, ammonium bisulphate, etc.) in the thermodynamics model ISORROPIA. The ions are then separated as tracers in the model, after the relevant salt formation. No change to the text is needed but I wanted to correct this statement from the response to reviews.

b. Line 224 refers to the "buffering effect of sulphate". What do the authors mean by this? The response to reviews and updated methods seems to suggest that the authors consider this to be a thermodynamic partitioning effect. However, while decreases in sulphur emissions can dramatically impact nitrate formation (by freeing up ammonia to produce ammonium nitrate), the inverse is not true. Sulphate does not need to be neutralized by ammonia to form in the atmosphere (i.e. H₂SO₄ can directly condense). Thus, the response of sulphate to changing NO_x emissions is primarily due to the response of the oxidants (i.e. changes in O₃ and OH will impact the rate of SO₂ oxidation). While this process is presumably included in the GEOS-Chem model, it is noticeably absent from the manuscript discussion. The interaction between nitrate and sulphate should be clarified in the main text and methods.

c. The response of oxidants to changing NO_x would also perturb other secondary constituents, such as organic aerosol, which are not included in this analysis. These responses via oxidants are likely modest, however it would certainly add to the uncertainty and should be mentioned in the text.

d. The authors responded to the question of non-linear ozone chemistry (and changing sign in particular) in the response to reviews, but did not add anything to the manuscript text. They should state somewhere in the main text that from their historical simulations they find that the global O₃ formation is NO_x limited and therefore O₃ responds positively to emissions changes in NO_x. If indeed they have verified that there is a positive correlation between changes in NO_x and O₃ at the column level in *every* gridbox then they should also state that this is true at the local level as well (and if not true, they should indicate that there are local differences in the sign of the response of O₃ to NO_x).

2. The aerosol radiative forcing value itself is likely high, reflecting the long-standing high bias in the aerosol nitrate simulation in GEOS-Chem (Zhang et al., 2012; Travis et al., 2016; Dutta and Heald, 2023).

While this bias would not change the sign of the net effect of Nr in this study, the authors should acknowledge that the aerosol cooling is most likely overestimated.

3. Line 85-86: the authors refer to “climate responses” but only the forcing, not the response, is characterised in this study.

4. Line 245: The numbers given here are global; please specify “the global total fossil fuel sources of Nr”. The trajectory of fossil fuel emissions (and perhaps agricultural emissions) of Nr differ regionally in these future scenarios. Please comment on this in the text and highlight that some regions may experience different trends in aerosol forcing.

5. Figure 5 legend: please re-organize the order of the legend to correspond to the order of the bars – it is difficult to identify which shade of yellow/brown corresponds to which species.

Referee 3's comments to authors:

The authors did a nice job addressing all the remaining comments from the previous reviews. I would suggest three minor modifications to the phrasing of the manuscript text for these responses (detailed below). Otherwise, the paper is, in my view, ready for publication in Nature.

1. Line 539: How would NH₃ emissions perturb sulphate? The authors have described the mechanism by which NO_x can alter sulphate via oxidant loading. Ammonia does not appreciably impact oxidant loading, and therefore I suggest the authors remove NH₃ from this line (or otherwise describe the mechanism by which NH₃ can modulate sulphate).

2. Line 242: NO_x can also alter secondary organic aerosol formation (i.e. yields vary as a function of NO_x (e.g. Kroll et al., 2006; Ng et al. 2007a; 2007b)). Thus for completeness I suggest that this addition be made to the text: “by altering atmospheric oxidation capacity and aerosol yields”

3. Line 243: “expected insignificant magnitude” seems a precise statement (“insignificant” has a specific statistical meaning) that the authors have not verified. More broadly, it’s not clear if this effect (especially when including the yield effect in point #2) is indeed negligible. There is however, quite a bit of uncertainty in SOA formation and its representation in models. I would suggest that the authors modify their stated reasoning for not exploring organic aerosol in this work.

Author Rebuttals to First Revision:

Response to referee #1

Title: Global net climate effects of anthropogenic reactive nitrogen

Nature reference number: 2023-10-19078C

Authors: Cheng Gong, Hanqin Tian, Hong Liao, Naiqing Pan, Shufen Pan, Akihiko Ito, Atul K. Jain, Sian Kou-Giesbrecht, Fortunat Joos, Qing Sun, Hao Shi, Nicolas Vuichard, Qing Zhu, Changhui Peng, Federico Maggi, Fiona H.M. Tang and Sönke Zaehle

Very interesting and novel paper examining the net radiative forcing effects of anthropogenically derived reactive nitrogen (Nr). In the end, the net cumulative effect of the anthropogenic Nr is a cooling of 0.34 W m² [0.20, 0.50] relative to the year 1850. The manuscript is well-written, very thorough and has global implications, so it is certainly appropriate for the journal "Nature". The methodology is appropriate and well described, and the results are clearly shown through a number of nicely organized figures. The authors have run an ensemble of model runs to investigate the uncertainties and this is also presented clearly and concisely in the manuscript. The conclusions of the study are also well supported by the data. I will say though, that it is odd to see CO₂ and CH₄ on the cooling side of the equation (e.g. Figure 5). I understand that it is due to C sequestration and a shortening of the CH₄ lifespan in the atmosphere, but still.

Response:

We appreciate for reviewer's acknowledgment for our work. We have revised all sub-titles in Figs. 3-5 as 'ΔCO₂', 'ΔN₂O', and etc. to indicate they are changes induced by anthropogenic Nr.

The authors have also done a very nice, and thorough, job of responding to the concerns of the previous reviewers. I also have a few small editorial type revisions (see below).

Line numbers are from "clean" document

Line 67: I would say "An earlier study".

Response:

Revised.

Line 162: delete the space after the "(".

Response:

Revised.

Line 201: Not really a revision, but I find it interesting the NO_x, which is mainly produced via fossil fuel combustion, dominates the cooling effects of non-ag sources. So, reducing fossil fuel combustion will reduce warming from CO₂, but at the same time reduce cooling from NO_x.

Response:

Thank you for the discussion. We would like to point out that warming effects of fossil fuel CO₂ are still stronger than the cooling effects due to fossil-fuel derived NO_x, which implies a win-win situation with benefits of both climate change mitigation (reduction in CO₂) and environmental health (alleviation of air pollution by NO_x reduction) by reducing fossil fuel combustion. However, we agree that one conclusion from this study is that the cooling result by reducing fossil-fuel CO₂ emissions will be partly offset by the reduced cooling effect of NO_x.

Line 243: I think you should clarify here that this is just a net warming effect from the changes to Nr and not an overall net warming from a strong reduction in fossil fuel use.

Response:

We revise the statement as:

‘These **Nr-related** emissions changes lead to a net warming effect of +0.09 W m⁻² by 2050s...’

Line 253: “imply” not “implicate”.

Response:

Revised.

Line 456: should that be carbon monoxide?

Response:

Yes. We have revised it.

Line 484 (and 486): I would suggest deleting the “s” from fossil fuel combustions (combustion).

Response:

Revised.

Line 550 (also line 583): Greenhouse, not green-house.

Response:

Revised.

Extended data Figure 3: in e and g, I assume those lines are related to shipping?

Response:

Yes.

Nice work with Extended data figure 6.

Response:

Thank you.

Response to referee #2

Title: Global net climate effects of anthropogenic reactive nitrogen

Nature reference number: 2023-10-19078C

Authors: Cheng Gong, Hanqin Tian, Hong Liao, Naiqing Pan, Shufen Pan, Akihiko Ito, Atul K. Jain, Sian Kou-Giesbrecht, Fortunat Joos, Qing Sun, Hao Shi, Nicolas Vuichard, Qing Zhu, Changhui Peng, Federico Maggi, Fiona H.M. Tang and Sönke Zaehle

The authors addressed all my comments appropriately and the explanations given on the long-term non-linearity effects of Nr deposition on C sequestration and soil N₂O emissions are sensible and reflect the current state of knowledge. I appreciate the significant improvement of explanations on the underlying models and data including Extended Data Figure 6 and the attempt to provide realistic uncertainty estimates.

The paper provides a great summary of the current knowledge of global Nr effects on the climate system, including all of its major processes and underlying complexities. Very nice work

Response:

We appreciate for reviewer's acknowledgment for our work.

Response to referee #3

Title: Global net climate effects of anthropogenic reactive nitrogen

Nature reference number: 2023-10-19078C

Authors: Cheng Gong, Hanqin Tian, Hong Liao, Naiqing Pan, Shufen Pan, Akihiko Ito, Atul K. Jain, Sian Kou-Giesbrecht, Fortunat Joos, Qing Sun, Hao Shi, Nicolas Vuichard, Qing Zhu, Changhui Peng, Federico Maggi, Fiona H.M. Tang and Sönke Zaehle

The authors have made substantive changes to their manuscript to address issues raised in review. In particular, they have clarified their methods, adopted a more realistic future scenario, and made some attempt to address the question of non-linearities. There are a few items that I think should be clarified in the manuscript text. None of these are issues that will substantively change the conclusions of the manuscript, but should be addressed for accuracy/completeness. Once the authors have made these changes, I recommend that the manuscript be accepted for publication.

Response:

We appreciate for the reviewer's advice to further improve this manuscript. We agree with all of the suggestions and have addressed them in the main text. Please see the point-to-point response below.

1. Atmospheric chemistry:

a. Correction: in the response to reviews the authors say that "there is actually no ammonium nitrate" in GEOS-Chem. This is not quite right. Ammonium nitrate is simulated in GEOS-Chem (as is ammonium sulphate, ammonium bisulphate, etc.) in the thermodynamics model ISORROPIA. The ions are then separated as tracers in the model, after the relevant salt formation. No change to the text is needed but I wanted to correct this statement from the response to reviews.

Response:

Sorry for our previous inaccurate statement. Indeed, ammonium nitrate is considered in ISORROPIA but separated as ammonium and nitrate tracers in the diagnosed output of GEOS-Chem. We appreciate the reviewer to point this out and accept this correction.

b. Line 224 refers to the "buffering effect of sulphate". What do the authors mean by this? The response to reviews and updated methods seems to suggest that the authors consider this to be a thermodynamic partitioning effect. However, while decreases in sulphur emissions can dramatically impact nitrate formation (by freeing up ammonia to produce ammonium nitrate), the inverse is not true. Sulphate does not need to be neutralized by ammonia to form in the atmosphere (i.e. H₂SO₄ can directly condense). Thus, the response of sulphate to changing NO_x emissions is primarily due to the response of the oxidants (i.e.

changes in O₃ and OH will impact the rate of SO₂ oxidation). While this process is presumably included in the GEOS-Chem model, it is noticeably absent from the manuscript discussion. The interaction between nitrate and sulphate should be clarified in the main text and methods.

Response:

Following the reviewer's suggestion, we have added explanation about the effects of NO_x on atmospheric oxidants and the subsequent influences on SO₂ oxidation in the main text as well as the method:

In the main text (revision **in bold**):

‘It should be noted that the radiative forcings attributed to agricultural and non-agricultural Nr are affected by the nonlinearity **in the chemistry of aerosol formation**, which results in a somewhat stronger net cooling effect **from the sum of the individual effects** (-0.38 W m⁻²) compared to the combined estimate (-0.34 W m⁻² in Fig. 3f). The direct radiative forcing of nitrate aerosol is not only weakened by substantial NO_x reductions in the no_nonagriNr experiment, but also reduces by the decline in the ammonium nitrate aerosol associated with the significant NH₃ emission reduction in the no_agriNr experiment (See Methods, Extended Data Figure 3 and Table S5). **The NO_x reduction further affects the concentrations of atmospheric oxidants such as O₃ and OH and reduces the formation of sulphate aerosol in no_nonagriNr experiment (See Table S5 and Methods). Nevertheless, this non-linearity in aerosol chemistry does not influence the ranking or overall magnitude of the factors by which Nr influences radiative forcing.**’

In the methods (revision **in bold**):

‘In particular, GEOS-Chem fully considers the non-linearity of inorganic aerosol chemistry, where sulphate aerosol has higher priority than nitrate aerosol in aerosol formation when ammonia gas is limited in the atmosphere. **Changes in the atmospheric NO_x loading can also affect oxidation of sulfur dioxide by perturbing atmospheric oxidants, such as O₃ and OH.** As a result, the sulphate aerosol loadings can also be perturbed by changes in NH₃ and NO_x emissions, despite the fact that the sulfur dioxide emissions are identical in all our experiments.’

c. The response of oxidants to changing NO_x would also perturb other secondary constituents, such as organic aerosol, which are not included in this analysis. These responses via oxidants are likely modest, however it would certainly add to the uncertainty and should be mentioned in the text.

Response:

Combining with the comments of Point 2 below, we added discussions (**in bold**) about the aerosol RF in the main text as:

‘...**For the effects we examined in this study, on the one hand**, the future CO₂ cooling due to CO₂ uptake on land may be overestimated in our study because we omit the contribution of fossil-fuel based CO₂ emissions from N fertilizer production by the Haber Bosch, and more importantly, terrestrial ecosystems exposed to high chronic N additions may become N saturated within the next few decades and contribute less to terrestrial C storage (See details in SI text S2.1). Uncertainties also remain in quantifying soil N₂O, NO_x and NH₃ emissions (See details in SI text S2.2 to S2.4). **On the other hand, the negative radiative**

forcing of nitrate aerosol may be overestimated, as the GEOS-Chem model tends to overestimate nitrate aerosol concentrations^{4,6}. Furthermore, changes in NO_x can further influence the formation of organic aerosols by altering atmospheric oxidation capacity, which are not examined in this study given the expected insignificant magnitude. To reduce uncertainties and gain a more comprehensive understanding of potential feedbacks, the development of more integrative Earth system models including key interactions among processes of terrestrial and marine biogeochemistry, atmospheric chemistry, climate dynamics and radiative processes would be required.'

*d. The authors responded to the question of non-linear ozone chemistry (and changing sign in particular) in the response to reviews, but did not add anything to the manuscript text. They should state somewhere in the main text that from their historical simulations they find that the global O₃ formation is NO_x limited and therefore O₃ responds positively to emissions changes in NO_x. If indeed they have verified that there is a positive correlation between changes in NO_x and O₃ at the column level in *every* gridbox then they should also state that this is true at the local level as well (and if not true, they should indicate that there are local differences in the sign of the response of O₃ to NO_x).*

Response:

We revised the last sentence when describing Fig. 3 as:

'In response to the substantial NO_x increases since pre-industrial times, present-day tropospheric O₃ was found to be enhanced across the entire simulated global grid, resulting in significant increases in global tropospheric O₃ burden from 280.1 Tg to 325.0 Tg (Extended Data Figure 2). This O₃ enhancement partly offsets the cooling climate effects from reduced CH₄ lifetime and increased aerosol burden considering the greenhouse gas effect of O₃.'

2. The aerosol radiative forcing value itself is likely high, reflecting the long-standing high bias in the aerosol nitrate simulation in GEOS-Chem (Zhang et al., 2012; Travis et al., 2016; Dutta and Heald, 2023). While this bias would not change the sign of the net effect of Nr in this study, the authors should acknowledge that the aerosol cooling is most likely overestimated.

Response:

Please see our response to Point 1c.

3. Line 85-86: the authors refer to "climate responses" but only the forcing, not the response, is characterised in this study.

Response:

We have revised it as '...the likely changes in radiative forcings in response to future changes of anthropogenic Nr inputs'.

4. Line 245: The numbers given here are global; please specify "the global total fossil fuel sources of Nr". The trajectory of fossil fuel emissions (and perhaps agricultural emissions) of Nr differ regionally in these future scenarios. Please comment on this in the text and highlight that some regions may experience different trends in aerosol forcing.

Response:

We have revised it as:

‘In the SSP 3-7.0 scenario, the future global total fossil fuel sources of Nr remain close to the 2019 level, resulting in similar magnitude of global aerosol forcing but potentially various trends among different regions.’

5. Figure 5 legend: please re-organize the order of the legend to correspond to the order of the bars – it is difficult to identify which shade of yellow/brown corresponds to which species.

Response:

The Fig. 5 has been revised as:

Reference

- 4 Zhang, L. *et al.* Nitrogen deposition to the United States: distribution, sources, and processes. *Atmospheric Chemistry and Physics* **12**, 4539-4554, doi:10.5194/acp-12-4539-2012 (2012).
- 5 Travis, K. R. *et al.* Why do models overestimate surface ozone in the Southeast United States? *Atmospheric Chemistry and Physics* **16**, 13561-13577, doi:10.5194/acp-16-13561-2016 (2016).
- 6 Dutta, I. & Heald, C. L. Exploring Deposition Observations of Oxidized Sulfur and Nitrogen as a Constraint on Emissions in the United States. *Journal of Geophysical Research-Atmospheres* **128**, doi:10.1029/2023jd039610 (2023).

Response to referee #3

Title: Global net climate effects of anthropogenic reactive nitrogen

Nature reference number: 2023-10-19078D

Authors: Cheng Gong, Hanqin Tian, Hong Liao, Naiqing Pan, Shufen Pan, Akihiko Ito, Atul K. Jain, Sian Kou-Giesbrecht, Fortunat Joos, Qing Sun, Hao Shi, Nicolas Vuichard, Qing Zhu, Changhui Peng, Federico Maggi, Fiona H.M. Tang and Sönke Zaehle

The authors did a nice job addressing all the remaining comments from the previous reviews. I would suggest three minor modifications to the phrasing of the manuscript text for these responses (detailed below). Otherwise, the paper is, in my view, ready for publication in Nature.

Response:

We appreciate reviewer's suggestions to further improve the text. Please see our point-to-point response below.

1. Line 539: How would NH₃ emissions perturb sulphate? The authors have described the mechanism by which NO_x can alter sulphate via oxidant loading. Ammonia does not appreciably impact oxidant loading, and therefore I suggest the authors remove NH₃ from this line (or otherwise describe the mechanism by which NH₃ can modulate sulphate).

Response:

The NH₃ has been removed.

2. Line 242: NO_x can also alter secondary organic aerosol formation (i.e. yields vary as a function of NO_x (e.g. Kröll et al., 2006; Ng et al. 2007a; 2007b)). Thus for completeness I suggest that this addition be made to the text: "by altering atmospheric oxidation capacity and aerosol yields"

Response:

Added.

3. Line 243: "expected insignificant magnitude" seems a precise statement ("insignificant" has a specific statistical meaning) that the authors have not verified. More broadly, it's not clear if this effect (especially when including the yield effect in point #2) is indeed negligible. There is however, quite a bit of uncertainty in SOA formation and its representation in models. I would suggest that the authors modify their stated reasoning for not exploring organic aerosol in this work.

Response:

Thank you for the suggestion. We have revised the statement as:

‘Furthermore, changes in NO_x can further influence the formation of organic aerosols by altering atmospheric oxidation capacity **and aerosol yields**¹⁻³, which are not examined in this study given **the large uncertainty in simulating corresponding chemical processes.**’

Reference:

- 1 Kroll, J. H., Ng, N. L., Murphy, S. M., Flagan, R. C. & Seinfeld, J. H. Secondary organic aerosol formation from isoprene photooxidation. *Environmental Science & Technology* **40**, 1869-1877, doi:10.1021/es0524301 (2006).
- 2 Ng, N. L. *et al.* Effect of NO_x level on secondary organic aerosol (SOA) formation from the photooxidation of terpenes. *Atmospheric Chemistry and Physics* **7**, 5159-5174, doi:10.5194/acp-7-5159-2007 (2007).
- 3 Ng, N. L. *et al.* Secondary organic aerosol formation from m-xylene, toluene, and benzene. *Atmospheric Chemistry and Physics* **7**, 3909-3922, doi:10.5194/acp-7-3909-2007 (2007).